# Network Lasso Bandits

## Abstract

We consider a multi-task contextual bandit setting, where the learner is given a graph encoding relations between the bandit tasks. The tasks' preference vectors are assumed to be piecewise constant over the graph, forming clusters. At every round, we estimate the preference vectors by solving an online network lasso problem with a suitably chosen, time-dependent regularization parameter. We establish a novel oracle inequality relying on a convenient restricted eigenvalue assumption. Our theoretical findings highlight the importance of dense intra-cluster connections and sparse inter-cluster ones. That results in a sublinear regret bound significantly lower than its counterpart in the independent task learning setting. Finally, we support our theoretical findings by experimental evaluation against graph bandit multi-task learning and online clustering of bandits algorithms.

## 1 Introduction

Online commercial websites aim to properly recommend their products to their customers, and the performance of these recommendations depends on the knowledge of users' preferences. Unlike traditional collaborative-filtering-based methods [Su and Khoshgoftaar, 2009], such knowledge is initially unavailable. Therefore, the online recommender systems need to recommend various items to the users and observe their ratings to *explore* their preferences. At the same time, the recommender system should be able to recommend items that attract users' attention and receive high ratings by *exploiting* the learned knowledge. The contextual bandits frameworks [Li et al., 2010] have been popularly used to formalize and address this exploration-exploitation trade-off.

However, the classical form of contextual bandits [Li et al., 2010, Chu et al., 2011, Abbasi-Yadkori et al., 2011] ignores the availability of social networks amongst users and solves the problem for each user separately. Consequently, such algorithms have some drawbacks when applied to problems with a large number of users. First, such a large number hinders the computational efficiency of such algorithms. Second, the partial feedback of the bandit settings exposes the algorithms to have weak estimations and impair their decision-making ability [Yang et al., 2020]. Consequently, to improve bandit algorithms' performance for large-scale applications, structural assumptions that link the different users are usually integrated within bandit algorithms [Cesa-Bianchi et al., 2013, Gentile et al., 2014, Li et al., 2019, Herbster et al., 2021].

The papers of Cesa-Bianchi et al. [2013], Yang et al. [2020] attempt to integrate the prior knowledge of social networks into their contextual bandit algorithms. Both papers proposed UCB-style algorithms and exhibited the importance of using the social network graph to achieve lower regrets using Laplacian regularization. Consequently, both methods promote smoothness among the preference vectors of users in order to transfer the collected information between them. However, the Laplacian regularization does not account for the smoothness heterogeneity introduced by a piecewise constant behavior over the graph [Wang et al., 2016]. On the other hand, algorithms of online clustering of bandits [Gentile et al., 2014, Li et al., 2019] start from a graph and gradually add or remove edges to

form clusters as connected components. However, their clustering can cause overconfidence in the constructed clusters, potentially leading to error accumulation.

In this paper, we assume access to a graph encoding relations between bandit tasks, and that the task parameter vectors are piecewise constant over the graph. That means that tasks form clusters. We propose an algorithm that integrates the prior knowledge of the piecewise constant structure to update tasks rather than finding the clusters explicitly. That way, we mitigate the limitations mentioned above: the piecewise constant smoothness is naturally integrated into our regularizer, and we do not estimate the clusters so our algorithm does not suffer from overconfidence drawbacks.

More precisely, we provide the following contributions

- We analyze an instance of the Network Lasso problem [Hallac et al., 2015], where every vertex's preference vector is estimated using data generated during the interaction between users and the bandit. We provide the first oracle inequality in this setting and link it to fundamental quantities characterizing the relation between the graph and the true preference vectors of the users. Our result relies on our novel restricted eigenvalue (RE) condition, which we assume for our setting. This result is of independent interest and can be applied to independently generated data as a special case.

- We prove how the empirical multi-task Gram matrix of the data inherits the RE condition from its true counterpart. Both this result and the previous one depend on the sparsity of inter-cluster connections and the density of intra-cluster ones.

- We provide a regret upper bound for our setting. Our bound highlights the advantage of our algorithm in high dimensional settings, and for large graphs.

- We support our theoretical findings by extensive numerical experiments on simulated data that prove the advantage of our algorithm compared to other approaches used for online clustering of bandits.

The rest of the paper is organized as follows. Section 2 discusses the relation of our work to the literature. We formulate our problem and state some of our assumptions in Section 3, then we present our bandit algorithm in Section 4. We analyze the problem theoretically in Section 5, and finally, we demonstrate its practical interest via numerical experiments in Section 6.

## 2   Related work

**Lasso contextual bandits**   To address the high dimensional setting for linear bandits, several multi-armed bandit papers solve a LASSO [Tibshirani, 1996] problem under different assumptions [Bastani and Bayati, 2019, Kim and Paik, 2019, Oh et al., 2021, Ariu et al., 2022]. They all rely on a previously established compatibility or RE condition [Bühlmann and van de Geer, 2011], that they adapt to the non-i.i.d case. Such assumptions were also used in the multi-task setting by Cella and Pontil [2021] with a Group Lasso regularization [Yuan and Lin, 2006], and to impose a low rank structure on the task preference vectors in Cella et al. [2023]. In our case, we provide a novel oracle inequality, rather than just generalize an existing one to the non-i.i.d setting, with a newly introduced RE assumption.

**Clustering of bandits**   Sequentially clustering bandit tasks was introduced in Gentile et al. [2014] with CLUB algorithm. In CLUB, starting with a fully connected graph, an iterative graph learning process is performed, where edges between users are deleted if their preference vectors are significantly different. As a result, any connected component is seen as a cluster and only one recommendation per cluster is developed. In another work, Li et al. [2019] generalize the setting of Gentile et al. [2014] and address its limitations via including merging operations in addition to splitting. In contrast to these approaches, the algorithm in Nguyen and Lauw [2014] groups users via K-means clustering, and the algorithm in Cheng et al. [2023] relies on hedonic games for online clustering of bandits. Furthermore, Yang and Toni [2018] make use of community detection techniques on graphs to find user clusters. Gentile et al. [2017] study the clustering of the contextual bandit problem where their proposed algorithm, named CAB, adaptively matches user preferences in the face of constantly evolving items. Our work fundamentally differs from the previous ones on two aspects. First, we assume access to a graph encoding relations between users, which is more informative than a complete graph. Second, we do not keep track of a model for each cluster, but rather we integrate a prior over

the graph via a graph total variation regularizer that enforces a piecewise constant behaviour for the estimated preference vectors.

**Multi-task learning**    Several contributions assume some underlying structure that links the bandit tasks. In Cella and Pontil [2021], task preference vectors are assumed to be sparse and to share their sparsity support, implying that they lie in a low-dimensional subspace with dimensions aligning with the canonical basis vectors. This idea is further generalized in Cella et al. [2023], where the tasks are assumed to be confined to an arbitrary unknown low-dimensional subspace. That work improves upon Hu et al. [2021] by not requiring the knowledge of the small dimenson of the task space. The underlying structure linking tasks can also be a graph encoding relations between them [Cesa-Bianchi et al., 2013, Yang and Toni, 2018], which is our case. However, while they assume smoothness as a prior, we assume piecewise constant behavior.

# 3    Problem setting

We consider a linear bandit setting, with a finite number of tasks representing users in a recommendation system for example. For each task the agent has to choose among $K$ arms, each associated to a $d$-dimensional context vector. All interactions over a horizon of $T$ time steps. We further assume that we have access to an undirected graph $\mathcal{G} = (\mathcal{V}, \mathcal{E})$, with vertex set $\mathcal{V}$ representing the tasks and edge set $\mathcal{E}$ encoding the relationships between them. We identify the vertex set $\mathcal{V}$ with the set of vertex indices $[|\mathcal{V}|]$. Thus, we consider $\mathcal{E}$ to be a subset of $\mathcal{V}^2$, where every edge $(m, n) \in \mathcal{E}$ has weight $w_{mn} > 0$, with $m < n$. The tasks' preference vectors are denoted by $\{\boldsymbol{\theta}_m\}_{m \in \mathcal{V}} \subset \mathbb{R}^d$ verifying $\|\boldsymbol{\theta}_m\| \leq 1 \ \forall m \in \mathcal{V}$, which we concatenate as row vectors into matrix $\boldsymbol{\Theta} \in \mathbb{R}^{|\mathcal{V}| \times d}$. The latter represents a graph vector signal, assumed to be piecewise constant over $\mathcal{G}$.

At a round $t \in \mathbb{N}^\star$, a user $m(t) \in \mathcal{V}$ is selected uniformly at random and served an arm with context vector $\mathbf{x}(t)$ from a finite action set $\mathcal{A}(t) \subset \mathbb{R}^d$ with size $K$, depending on their estimated preference vector $\hat{\boldsymbol{\theta}}_{m(t)}(t) \in \mathbb{R}^d$. We assume the expected reward to be linear, with an additive, $\sigma$-sub-Gaussian noise conditionally on the past. Formally, denoting by $\mathcal{F}_0$ the trivial sigma-algebra, and for all $t \geq 1$, by $\mathcal{F}_t$ the sigma-algebra generated by history set $\{m(1), \mathbf{x}(1), y(1), \cdots, m(t), \mathbf{x}(t), y(t), m(t+1)\}$, the received reward $y(t)$ is given by $y(t) = \langle \boldsymbol{\theta}_{m(t)}(t), \mathbf{x}(t) \rangle + \eta(t)$, where $\eta(t)$ is $\mathcal{F}_t$−measurable and

$$\mathbb{E}\left[\eta(t) | \mathcal{F}_{t-1}\right] = 0, \qquad \mathbb{E}\left[\exp(s\eta(t)) | \mathcal{F}_{t-1}\right] \leq \exp\left(\frac{1}{2}\sigma^2 s^2\right) \quad \forall t \geq 1, \forall s \in \mathbb{R}. \tag{1}$$

At the end of a round $t$, all preference vectors are updated into a new estimation $\hat{\boldsymbol{\Theta}}(t)$ while leveraging the structure of graph $\mathcal{G}$, formally by solving the following optimization problem:

$$\hat{\boldsymbol{\Theta}}(t) = \underset{\tilde{\boldsymbol{\Theta}} \in \mathbb{R}^{|\mathcal{V}| \times d}}{\arg \min} \frac{1}{2t} \sum_{\tau=1}^{t} \left( \left\langle \tilde{\boldsymbol{\theta}}_{m(\tau)}, \mathbf{x}(\tau) \right\rangle - y(\tau) \right)^2 + \alpha(t) \sum_{(m,n) \in \mathcal{E}} w_{mn} \left\| \tilde{\boldsymbol{\theta}}_m - \tilde{\boldsymbol{\theta}}_n \right\|, \tag{2}$$

where $\|\cdot\|$ denotes the Euclidean norm for vectors. The performance of our policy is assessed by the expected regret over the $T$ interaction rounds for all tasks:

$$\mathcal{R}(T) = \mathbb{E}\left[ \sum_{t=1}^{T} \left\langle \boldsymbol{\theta}_{m(t)}, \mathbf{x}^\star(t) - \mathbf{x}(t) \right\rangle \right], \tag{3}$$

where $\mathbf{x}^\star(t) \in \arg \max_{\tilde{\mathbf{x}} \in \mathcal{A}(t)} \left\langle \boldsymbol{\theta}_{m(t)}, \tilde{\mathbf{x}} \right\rangle$.

The Optimization problem in (2) is an instance of the Network Lasso [Hallac et al., 2015]. Other instances of the same type were studied by Jung et al. [2018], Jung and Vesselinova [2019], Jung [2020]. The objective is characterized by its second term that, while being just the Laplacian regularization without squaring the norms, promotes a piecewise constant behavior rather than smoothness. For real-valued signals ($d = 1$), this regularization has been extensively studied for image and graph signal denoising, for the problem of trend filtering on graphs [Wang et al., 2016]. According to Wang et al. [2016], that regularization better adapts to the heterogeneity of smoothness of the signal and induces a cluster structure in the data: similar users will not only have similar models but the same model, which offers a compression of the overall model over the graph. Note

that our setting is cluster agnostic; our algorithm does not aim to learn the cluster structure explicitly but to exploit it implicitly using the total variation semi-norm as regularization. The latter's strength is controlled via a time-dependent regularization coefficient $\alpha(t)$, which we will express later in the analysis.

We formalize our assumption on the context generation as follows.

**Assumption 1** (i.i.d action sets). *Context sets $\{\mathcal{A}(t)\}_{t=1}^{T}$ are generated i.i.d. from a distribution $p$ over $\mathbb{R}^{K \times d}$, such that $\|\mathbf{x}\| \leq 1 \forall \mathbf{x} \in \mathcal{A}(t) \, \forall t \geq 1$.*

In addition to the i.i.d assumption, we assume more regularity.

**Assumption 2** (Relaxed symmetry and balanced covariance). *There exists a constant $\nu \geq 1$ such that for all $\mathbf{X} \in \mathbb{R}^{K \times d}$, $p(-\mathbf{X}) \leq \nu p(\mathbf{X})$. Furthermore, there exists $\omega > 0$, such that for any permutation $(a_1, \cdots, a_K)$ of $[K]$, for any $i \in \{2, \cdots, K-1\}$, and for any $\mathbf{w} \in \mathbb{R}^d$, we have*

$$\mathbb{E}\left[\mathbf{x}_{a_i}\mathbf{x}_{a_i}^{\top}[\mathbf{w}^{\top}\mathbf{x}_{a_1} < \cdots < \mathbf{w}^{\top}\mathbf{x}_{a_K}]\right] \preccurlyeq \omega\mathbb{E}\left[(\mathbf{x}_{a_1}\mathbf{x}_{a_1}^{\top} + \mathbf{x}_{a_K}\mathbf{x}_{a_K}^{\top})[\mathbf{w}^{\top}\mathbf{x}_{a_1} < \cdots < \mathbf{w}^{\top}\mathbf{x}_{a_K}]\right],$$

*where $\mathbf{M} \preccurlyeq \mathbf{N}$ means that $\mathbf{N} - \mathbf{M}$ is a PSD matrix.*

This assumption was introduced in Oh et al. [2021], and has already been used in a multi-task setting by Cella et al. [2023]. Parameter $\nu$ controls the skewness, as $\nu = 1$ corresponds to a symmetric distribution. $\omega$ decreases with increasing positive correlation between arms. It verifies $\omega = O(1)$ for multi-variate Gaussians and uniform distributions over the unit sphere [Oh et al., 2021]. The piecewise constant behaviour of the graph signal $\Theta$ is formalized in the next assumption.

**Assumption 3** (Piecewise constant signal). *There exists a partition $\mathcal{P}$ of $\mathcal{V}$, such that for any cluster $\mathcal{C} \in \mathcal{P}$, signal $\Theta$ is constant on $\mathcal{C}$, and the graph obtained by taking the vertices in $\mathcal{C}$ and the edges linking them is connected.*

Assumption 3 basically states that the true preference vectors are clustered and that the given graph induces the cluster structure. It is required for our approach to be beneficial, as we will detail in the analysis section. For the sake of clarity, we defer the statement of other technical assumptions to Section 5.

# 4 Algorithm

Our policy in Algorithm 1 follows a greedy arm selection rule in a multi-task setting, in the same vein as those presented in Oh et al. [2021], Cella et al. [2023]. Indeed, as pointed out in Oh et al. [2021], exploration is implicitly incorporated into regularization parameter $\alpha(t)$'s time dependence. It has the following expression

$$\alpha(t) := \frac{\alpha_0\sigma}{t}\sqrt{t + \sqrt{2\sum_{m \in \mathcal{V}}|\mathcal{T}_m(t)|^2\log\frac{1}{\delta(t)}} + 2\max_{m \in \mathcal{V}}|\mathcal{T}_m(t)|\log\frac{1}{\delta(t)}}, \tag{4}$$

where the set of time steps a task $m$ has been selected up to time $t$ is denoted by $\mathcal{T}_m(t)$.

# 5 Analysis

This section provides the main steps of the analysis. One of the paper's contribution lies in finding an oracle inequality of the network lasso problem given a restricted eigenvalue condition holding for the true multi-task Gram matrix. In this regard, the next major challenge and contribution is to show that the empirical multi-task Gram matrix, estimated in the algorithm, satisfies the restricted eigenvalue condition. We start by proving an oracle inequality for the estimation error of $\Theta$, assuming that the condition given by Definition 2 is verified by the empirical data Gram matrix. Then, we prove that the latter assumption actually holds with high probability given that true multi-task Gram matrix satisfies it. Our final contribution in this work is the establishment of a regret bound for our algorithm.

## 5.1 Notation and technical assumptions

We provide additional notations required for the analysis. We denote by $\partial\mathcal{P}$ the set of all edges in $\mathcal{E}$ connecting vertices from different clusters from partition $\mathcal{P}$ (Assumption 3), and we call it the

**Algorithm 1: Network Lasso Policy**

---

**Input** : $T, \alpha_0 > 0, \mathcal{G}$, function $\delta$

**Initialization** : $\hat{\boldsymbol{\Theta}}(0) = \mathbf{0} \in \mathbb{R}^{|\mathcal{V}| \times d}$

**for** $t \in [1, T]$ **do**

    1. Draw a user $m(t) \in \mathcal{V}$ uniformly at random.

    2. Observe context set $\mathcal{A}(t)$.

    3. Select $\mathbf{x}(t) \in \arg\max_{\tilde{\mathbf{x}} \in \mathcal{A}(t)} \left\langle \hat{\boldsymbol{\theta}}_{m(t-1)}, \tilde{\mathbf{x}} \right\rangle$, breaking ties arbitrarily.

    4. Receive payoff $y(t)$

    5. Update $\alpha(t)$ via Equation (4)

    6. Update $\hat{\boldsymbol{\Theta}}(t)$ via solving the network Lasso problem (2)

**end**

---

boundary of $\mathcal{P}$. Thus, $\partial\mathcal{P}^c$, the complementary set of $\partial\mathcal{P}$, is formed by edges connecting vertices of the same cluster. The total weight of the boundary, *i.e.* the sum of its edges' weights, is referred to as $w(\partial\mathcal{P})$. Given a signal $\mathbf{Z} \in \mathbb{R}^{|\mathcal{V}| \times d}$, we denote by $\mathbf{Z}_\mathcal{P}$ the signal obtained by setting row vectors of $\mathbf{Z}$ to their mean-per-cluster value w.r.t. $\mathcal{P}$. For any edge subset $I \in \mathcal{E}$, we denote the following norms: $\|\cdot\|_F$ as the Frobenius norm, $\|\mathbf{z}\|_\mathbf{M} = \sqrt{\mathbf{z}^\top \mathbf{M} \mathbf{z}}$ as the weighted norm of vector $\mathbf{z} \in \mathbb{R}^d$ induced by matrix $\mathbf{M} \in \mathbb{R}^{d \times d}$ and $\|\boldsymbol{\Theta}\|_I := \sum_{(m,n) \in I} w_{mn} \|\boldsymbol{\theta}_m - \boldsymbol{\theta}_n\|$ as the total variation semi-norm of $\boldsymbol{\Theta} \in \mathbb{R}^{|\mathcal{V}| \times d}$ over $I$. Thus, the regularization term of Problem (2) is equal to $\|\boldsymbol{\Theta}\|_\mathcal{E}$. Also, we define the incidence matrix $\mathbf{B}_I \subset \mathbb{R}^{|\mathcal{E}| \times |\mathcal{V}|}$ restricted to $I \subseteq \mathcal{E}$ to be null except at rows with index $i \in I$ corresponding to edge $(m, n)$, where it equals $w_{mn}(\mathbf{e}_m - \mathbf{e}_n)$, where $\mathbf{e}_m$ is the $m^{\text{th}}$ canonical basis vector of $\mathbb{R}^{|\mathcal{V}|}$. We define $\mathbf{A}_\mathcal{V}(t) := \text{diag}\left(\mathbf{X}_1(t)^\top \mathbf{X}_1(t), \ldots, \mathbf{X}_{|\mathcal{V}|}(t)^\top \mathbf{X}_{|\mathcal{V}|}(t)\right) \in \mathbb{R}^{d|\mathcal{V}| \times d|\mathcal{V}|}$, and subsequently the empirical multi-task Gram matrix up to time step $t$ is given by $\frac{1}{t} \mathbf{A}_\mathcal{V}(t)$. The following definition introduces quantities related to the clusters defined by partition $\mathcal{P}$, with crucial roles that we will elucidate throughout the analysis.

**Definition 1** (Cluster content constants). *Let $\mathcal{C} \in \mathcal{P}$ be a cluster.*

- *We denote by $\partial_v \mathcal{C}$ the inner boundary of $\mathcal{C}$,* i.e. *the vertices of $\mathcal{C}$ that are connected to its complementary. We define the inner isoperimetric ratio of $\mathcal{C}$ as $\iota_\mathcal{G}(\mathcal{C}) := \frac{|\partial_v \mathcal{C}|}{|\mathcal{C}|}$.*

- *By abuse of notation, we denote as $\mathbf{B}_\mathcal{C}$ the incidence matrix restricted to edges linking vertices of $\mathcal{C}$, its associated Laplacian matrix by $\mathbf{L}_\mathcal{C} := \mathbf{B}_\mathcal{C}^\top \mathbf{B}_\mathcal{C}$, and its pseudo-inverse by $\mathbf{L}_\mathcal{C}^\dagger$. The topological centrality index of node $m \in \mathcal{C}$ w.r.t $\mathcal{C}$ is equal to $(\mathbf{L}_\mathcal{C}^\dagger)_{mm}^{-1}$. We define the topological centrality index of $\mathcal{C}$ by $c_\mathcal{G}(\mathcal{C}) := \min_{m \in \mathcal{C}} (\mathbf{L}_\mathcal{C}^\dagger)_{mm}^{-1}$.*

The inner isoperimetric ratio of a cluster measures how many "interior" nodes a cluster contains, in the sense that they are not connected to its complementary. It is at most equal to the isoperimetric ratio for weightless graphs as the size of the inner boundary is at most equal to that of the edge boundary, the latter being connected to the algebraic connectivity via the Cheeger inequality [Cheeger, 1970].

The topological centrality index measures the overall connectedness of a vertex in a network and indicates how robust a node is to edge failures [Ranjan and Zhang, 2013]. Also, it can be tied to electricity spreading in a network according to Van Mieghem et al. [2017]. We refer the interested reader to the two previously mentioned works for a detailed account of the properties of the topological centrality index. In the appendix, we show that for binary weights graphs the minimum topological centrality index is at least equal to the algebraic connectivity theoretically and experimentally, where we showcase that the difference between the two can be significant.

To proceed, we will need the following definition that introduces several notations to reduce the clutter.

**Definition 2** (Restricted Eigenvalue (RE) condition and norm). *Let $\{\mathbf{M}_i\}_{i=1}^{|\mathcal{V}|} \subset \mathbb{R}^{d \times d}$ be a set of positive semi-definite matrices. We say that the matrix $\mathbf{M}_\mathcal{V} := \mathrm{diag}(\mathbf{M}_1, \cdots, \mathbf{M}_{|\mathcal{V}|})$ verifies the restricted eigenvalue condition with constants $\kappa \geq 0$ and $\phi > 0$ if*

$$\phi^2 \|\mathbf{Z}\|_{\mathrm{RE}}^2 \leq \sum_{i \in \mathcal{V}} \|\mathbf{z}_i\|_{\mathbf{M}_i}^2 \quad \forall \mathbf{Z} \in \mathcal{S} \text{ with rows } \{\mathbf{z}_i\}_{i \in \mathcal{V}},$$

*where $\mathcal{S}$ is the cone defined by:*

$$\mathcal{S} := \{\mathbf{Z} \in \mathbb{R}^{|\mathcal{V}| \times d}; a_1(\mathcal{G}, \boldsymbol{\Theta})\|\mathbf{Z}\|_{\partial \mathcal{P}^c} \leq a_2(\mathcal{G}, \boldsymbol{\Theta})\|\overline{\mathbf{Z}}_\mathcal{P}\|_F + (1 - \kappa)^+ \|\mathbf{Z}\|_{\partial \mathcal{P}}\},$$

$$a_1(\mathcal{G}, \boldsymbol{\Theta}) := 1 - \frac{\frac{1}{\alpha_0} + 2\kappa w(\partial \mathcal{P})}{\min_{\mathcal{C} \in \mathcal{P}} \sqrt{c_\mathcal{G}(\mathcal{C})}}, \quad a_2(\mathcal{G}, \boldsymbol{\Theta}) := \frac{1}{\alpha_0} + \sqrt{2}\kappa w(\partial \mathcal{P}) \max_{\mathcal{C} \in \mathcal{P}} \sqrt{\iota_\mathcal{G}(\mathcal{C})},$$

*and the RE semi-norm is defined by $\|\mathbf{Z}\|_{\mathrm{RE}} := \|\overline{\mathbf{Z}}_\mathcal{P}\|_F \vee (1 - \kappa)^+ \|\mathbf{B}_{\partial \mathcal{P}}^\dagger \mathbf{B}_{\partial \mathcal{P}} \mathbf{Z}\|.$*

To interpret the previous definition, we point out that the sum on the right-hand side of Definition 2 can be written as $\|\mathrm{vec}(\mathbf{Z}^\top)\|_{\mathbf{M}_\mathcal{V}}$, where $\mathrm{vec}$ denotes the operation of stacking a matrix's columns vertically. As a result, the condition is analogous to requiring that $\mathbf{M}_\mathcal{V}$ is invertible with minimum eigenvalue $\phi^2$, but weaker since it holds only for signals $\mathbf{Z} \in \mathcal{S}$ and for the $\|\cdot\|_{\mathrm{RE}}$ norm. This requirement has the same form as the compatibility assumption for the Lasso [Bühlmann and van de Geer, 2011, Oh et al., 2021] or the restricted strong convexity assumption [Cella et al., 2023].

We further make the following assumption on the true multi-task Gram matrix:

**Assumption 4** (RE condition for the true multi-task Gram matrix). *For $k \in [K]$, let $\boldsymbol{\Sigma}_k := \mathbb{E}\left[\mathbf{x}_k \mathbf{x}_k^\top\right]$ be the Gram matrix of the $k^{th}$ context vector's marginal distribution, let $\boldsymbol{\Sigma}_\mathcal{V}$ be the true multi-task Gram matrix of the context vector generating distribution, given by*

$$\boldsymbol{\Sigma}_\mathcal{V} := \mathbf{I}_{|\mathcal{V}|} \otimes \overline{\boldsymbol{\Sigma}}, \quad where \quad \overline{\boldsymbol{\Sigma}} = \frac{1}{K} \sum_{k=1}^K \boldsymbol{\Sigma}_k. \tag{5}$$

*We assume that $\boldsymbol{\Sigma}_\mathcal{V}$ verifies RE condition (Definition 2) with some problem dependent constants $\kappa \in \left[0, \frac{1}{2w(\partial \mathcal{P})} \min_{\mathcal{C} \in \mathcal{P}} \sqrt{c_\mathcal{G}(\mathcal{C})}\right)$ and $\phi > 0$.*

This assumption is common to make for Lasso-like bandit problems [Oh et al., 2021, Ariu et al., 2022, Cella et al., 2023]. We will later show that it can be transferred to empirical multi-task Gram matrix.

## 5.2 Oracle inequality

This section is dedicated to provide a bound on the estimation error of the Network Lasso problem given in Equation (2) at a particular step $t$ of Algorithm 1. We assume fixed design, meaning that the context vectors are given and fixed, and we are not concerned by their randomness (due to the context generating distribution), nor by the randomness of their number for each user (due to random selection at each time step).

For a time step $t$, we deliver the oracle inequality controlling the deviation between the estimated preference vectors $\hat{\boldsymbol{\Theta}}(t)$ and the true ones $\boldsymbol{\Theta}$. For the sake of simplicity, we provisionally assume that the RE condition holds for the empirical multi-task Gram matrix $\mathbf{A}_\mathcal{V}(t)$.

**Theorem 1** (Oracle inequality). *Assume that the RE assumption holds for the empirical multi-task Gram matrix with constants $\kappa \in \left[0, \frac{1}{2w(\partial \mathcal{P})} \min_{\mathcal{C} \in \mathcal{P}} \sqrt{c_\mathcal{G}(\mathcal{C})}\right)$ and $\phi > 0$. Suppose that $\max_{m \in \mathcal{V}} |\mathcal{T}_m(t)| \leq bt$ for some $b > 0$. Then, with a probability at least $1 - \delta(t)$, we have*

$$\left\|\boldsymbol{\Theta} - \hat{\boldsymbol{\Theta}}(t)\right\|_F \leq 2\frac{\sigma}{\phi^2 \sqrt{t}} f(\mathcal{G}, \boldsymbol{\Theta}) \sqrt{1 + 2b\sqrt{|\mathcal{V}| \log \frac{1}{\delta(t)}} + 2b \log \frac{1}{\delta(t)}},$$

*where*

$$f(\mathcal{G}, \boldsymbol{\Theta}) := \alpha_0 \left(a_2(\mathcal{G}, \boldsymbol{\Theta}) + \sqrt{2}\mathbb{1}_{\leq 1}(\kappa)w(\partial \mathcal{P})\right) \left(\frac{a_2(\mathcal{G}, \boldsymbol{\Theta}) + \sqrt{2}\mathbb{1}_{\leq 1}(\kappa)w(\partial \mathcal{P})}{a_1(\mathcal{G}, \boldsymbol{\Theta}) \min_{\mathcal{C} \in \mathcal{P}} \sqrt{c_\mathcal{G}(\mathcal{C})}} + 1\right).$$

The proof of the previous theorem mainly relies on a decomposition of the estimation error signal into two parts: one is the projection of the error onto its mean per cluster value, that is, every node within the same cluster is mapped to the mean estimation error of its cluster. The second part of the decomposition is simply the residual part i.e. the deviation from the mean per cluster value, which is related to the incidence matrices of each cluster. The probabilistic statement comes from a high probability bound on the Euclidean norm of an empirical vector process associated with our problem, using a generalization of the Hanson-Wright inequality to the subgaussian case [Hsu et al., 2012, Theorem 2.1]. Compared to the bound of Jung [2020, Theorem 1], we bound a norm of the estimation error rather than just the total variation semi-norm. Additionally, the bound exhibits different behavior depending on whether $\kappa > 1$. Indeed, due to the expressions of $a_1(\boldsymbol{\Theta}, \mathcal{G})$ and $a_2(\boldsymbol{\Theta}, \mathcal{G})$, in the case where $\kappa > 1$, the bound significantly decreases with the products $w(\partial \mathcal{P}) \min_{\mathcal{C} \in \mathcal{P}} \sqrt{\iota(\mathcal{C})}$ and $w(\partial \mathcal{P}) \max_{\mathcal{C} \in \mathcal{P}} c_{\mathcal{G}}(\mathcal{C})^{-\frac{1}{2}}$, which are both small enough for dense intra-cluster edge links and sparse inter-cluster ones. However, when $\kappa < 1$, the $w(\partial \mathcal{P})$ term might dominate if it is moderately large, and its effect can only be mitigated via a small subgaussianity constant $\sigma$ or a large enough RE condition constant $\phi$.

## 5.3 RE condition for the empirical multi-task Gram matrix

To establish the oracle inequality, we assumed that the RE condition holds for the empirical multi-task Gram matrix. The goal of this section is to prove this holds with high probability. To this end, we use the same strategy as in Oh et al. [2021], Cella et al. [2023]. We prove that on the one hand, given the empirical multi-task Gram matrix inherits the RE condition from its adapted counterpart since it concentrates around it. On the other hand, we prove that the adapted Gram matrix verifies the RE condition due to Assumption 1, 2 and 4 made on the context generation distribution.

**Theorem 2** (RE condition holding for the empirical multi-task Gram matrix). *Under assumptions 2 and 4, let $t \geq 1$, and let $\kappa, \phi$ be the constants from Assumption 4. Assume that $\max_{m \in \mathcal{V}} |\mathcal{T}_m(t)| \leq bt$. Then, for any $\gamma \in \left(0, \left(1 + \frac{a_2(\mathcal{G}, \boldsymbol{\Theta}) + (1-\kappa)^+ \sqrt{2} w(\partial \mathcal{P})}{a_1(\mathcal{G}, \boldsymbol{\Theta})}\right)^{-2}\right)$, the empirical multi-task Gram matrix verifies the RE condition with constants $\kappa$ and $\hat{\phi}$, with*

$$\hat{\phi} = \tilde{\phi} \sqrt{1 - \gamma \left(1 + \frac{a_2(\mathcal{G}, \boldsymbol{\Theta}) + (1-\kappa)^+ \sqrt{2} w(\partial \mathcal{P})}{a_1(\mathcal{G}, \boldsymbol{\Theta})}\right)^2}, \tag{6}$$

*with a probability at least equal to $1 - 6d|\mathcal{V}| \exp\left(\frac{-3\gamma^2 \tilde{\phi}^4 (\min_{\mathcal{C} \in \mathcal{P}} (\tilde{c}_{\mathcal{G}}(\mathcal{C}) \wedge \tilde{c}_{\mathcal{G}}(\mathcal{C})^2) t)}{6b + 2\sqrt{2}\gamma \tilde{\phi}^2}\right)$, where*

$\tilde{\phi} := \frac{\phi}{\sqrt{2\nu\omega}}$ *and* $\tilde{c}_{\mathcal{G}}(\mathcal{C}) := c_{\mathcal{G}}(\mathcal{C}) \wedge |\mathcal{C}| \quad \forall \mathcal{C} \in \mathcal{P}.$

The proof follows the same approach as in Oh et al. [2021], Cella et al. [2023]; we prove that the RE condition transfers from the true multi-task Gram matrix to its adapted counterpart $\mathbf{V}_{\mathcal{V}}(t)$, defined as follows:

$$\mathbf{V}_{\mathcal{V}}(t) = \mathrm{diag}\left(\mathbf{V}_1(t), \cdots, \mathbf{V}_{|\mathcal{V}|}(t)\right), \tag{7}$$

where

$$\mathbf{V}_m(t) = \frac{1}{t} \sum_{\tau \in \mathcal{T}_m(t)} \mathbb{E}\left[\mathbf{x}(\tau)\mathbf{x}(\tau)^\top | \mathcal{F}_{\tau-1}\right]. \tag{8}$$

This transfer relies on the work of Oh et al. [2021, lemma 10]. The other step of the proof is showing that the empirical multi-task Gram matrix and $\mathbf{V}_{\mathcal{V}}(t)$ become close to each other with high probability after sufficiently many time steps, the respective distance between the two is measured with a matrix norm induced by the RE semi-norm and the restriction to set $\mathcal{S}$ (Definition 2). The bound showcases a dependence on $\min_{\mathcal{C} \in \mathcal{P}} c_{\mathcal{G}}(\mathcal{C}) \wedge |\mathcal{C}|$, which is of the same order as $|\mathcal{C}|$ for a fully connected cluster with vertices $\mathcal{C}$. It is also clear that with a higher minimum centrality of a cluster, the probability of satisfying the RE condition increases.

## 5.4 Regret bound

To bound the regret, we bound the expected instantaneous regret for each round $t \geq 1$. This bound relies on the oracle inequality holding and on the RE condition being satisfied for the empirical Gram matrix, both with high probability. These two conditions are ensured and Theorem 1 and Theorem 2.

**Theorem 3** (Regret bound). *Let the mean horizon per node be $\overline{T} = \frac{T}{|\mathcal{V}|}$. Let $\min_{\mathcal{C} \in \mathcal{P}} \sqrt{c_{\mathcal{G}}(\mathcal{C})}$ going asymptotically to infinity and $\max_{\mathcal{C} \in \mathcal{P}} \sqrt{\iota_{\mathcal{G}}(\mathcal{C})}$ going asymptotically to zero as well as $\max_{\mathcal{C} \in \mathcal{P}} \sqrt{\iota_{\mathcal{G}}(\mathcal{C})} w(\partial \mathcal{P})$ and $\frac{w(\partial \mathcal{P})}{\min_{\mathcal{C} \in \mathcal{P}} \sqrt{c_{\mathcal{G}}(\mathcal{C})}}$ going asymptotically to zero. Under assumptions 1 to 4 and $\kappa < 1$, the expected regret of the Network Lasso Bandit algorithm is upper bounded as follows:*

$$\mathcal{R}(|\mathcal{V}|\overline{T}) = \mathcal{O}\left(\sqrt{\frac{\overline{T}}{\min_{\mathcal{C} \in \mathcal{P}} c_{\mathcal{G}}(\mathcal{C})}}\left(\sqrt{|\mathcal{V}|} + \sqrt{\log(\overline{T}|\mathcal{V}|)} + \sqrt[4]{|\mathcal{V}\log(\overline{T}|\mathcal{V}|)|}\right) + \frac{1}{A}\log(d|\mathcal{V}|)\right),$$

*with $A = \dfrac{3\gamma^2 \min_{\mathcal{C} \in \mathcal{P}}(\tilde{c}_{\mathcal{G}}(\mathcal{C}) \wedge \tilde{c}_{\mathcal{G}}^2(\mathcal{C}))}{6\frac{\log(|\mathcal{V}|)}{\sqrt{|\mathcal{V}|}} + 2\sqrt{2}\gamma}.$*

Our regret is mainly formed of two parts. The first one is the sublinear time-dependent term and represents the bulk of horizon dependence. Interestingly, it does not depend on the dimension, which is a consequence of using the concentration inequality from Hsu et al. [2012]. Interestingly, it decreases as the topological centrality index grows with the graph size, which proves the importance of intra-cluster high connectivity.

The second significant term comes from ensuring the RE condition for the empirical multi-task Gram matrix, and can be interpreted as the number of time steps necessary for it to hold, as pointed out by Oh et al. [2021]. It has a logarithmic dependence in the graph size and in the dimension, which is a characteristic of regret bound of the "lasso type". Also noteworthy is that the regret grows with $\log(d)$ only in the time-independent term, making our policy useful in high-dimensional settings.

## 6 Experiments

We provide experiments to showcase the effect on the problem's parameters on our algorithm's performance as well as highlighting its advantageous performance compared to other algorithms. At each time step, the algorithm solves the network lasso problem (2) via a primal-dual algorithm used in Jung [2020].

We compare our algorithm to several baselines of the literature. On the one hand, baselines relying on a given graph, GOBLin [Cesa-Bianchi et al., 2013] and GraphUCB [Yang et al., 2020] that use the Laplacian to smooth the preference vectors. On the other hand, we consider online clustering of bandits baselines, namely CLUB [Gentile et al., 2014] and SCLUB [Li et al., 2019]. Since these latter approaches start with a fully connected graph, we provide them the known graph for a fair comparison. As a sanity check, we also compare the independent task learning case with LinUCB (LinUcbITL) where each task is solved independently, and to the case of a LinUCB agent for each cluster (LinUcbOracle). The graph used is generated using stochastic block models in order to ensure that the generated graph induces a cluster structure, where an edge is constructed with probability $p$ within clusters and $q$ between clusters.

Experimentally, we found that normalizing the adjacency matrix, that is we utilize the following normalized edges: $w_{mn} = \dfrac{1}{\sqrt{\deg(m)\deg(n)}}$, where $\deg(m)$ denotes the degree of node $m$, yields significantly better results. Indeed, such a normalization makes the algorithm focus more on edges between low-degree nodes, which improves the propagation of the collected information within the graph. In all experiments we have set $\alpha_0 = 0.1$.

Our results clearly showcase an improvement compared to the other baselines. Apart from the oracle that has complete knowledge of all clusters from the beginning, our policy performs significantly better than the rest beyond the error margins, covering one standard deviation at ten repetitions. We

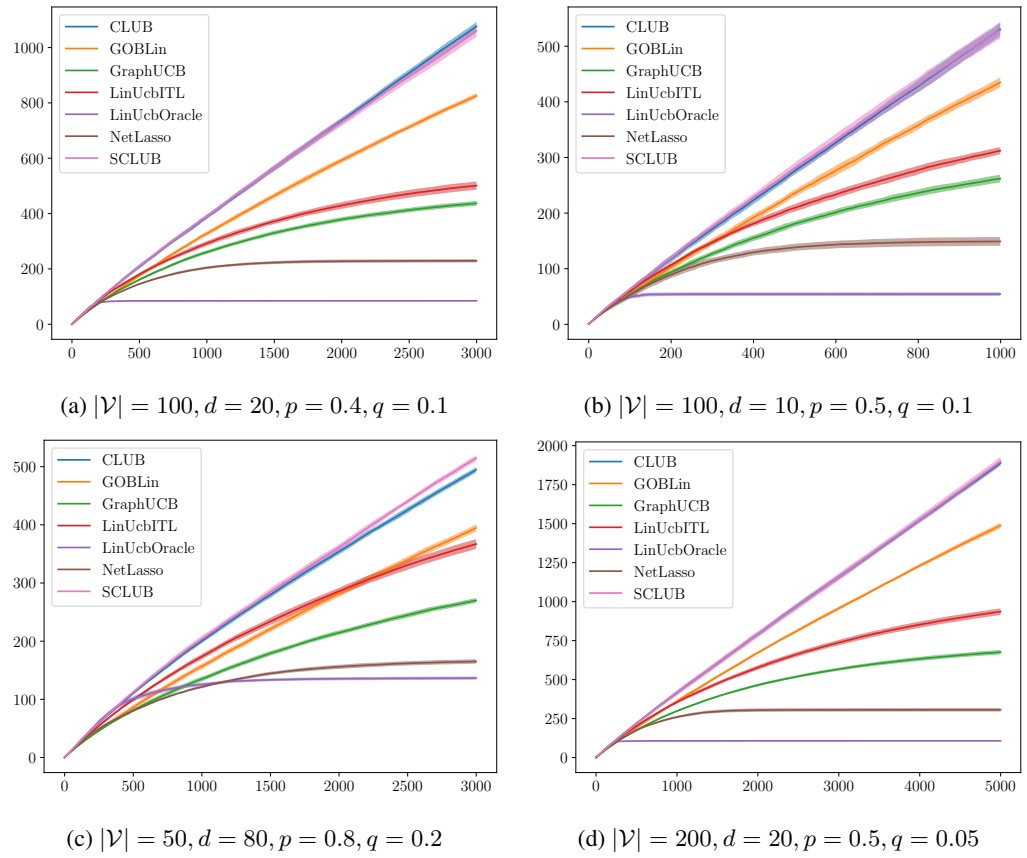

(a) $|\mathcal{V}| = 100, d = 20, p = 0.4, q = 0.1$

(b) $|\mathcal{V}| = 100, d = 10, p = 0.5, q = 0.1$

(c) $|\mathcal{V}| = 50, d = 80, p = 0.8, q = 0.2$

(d) $|\mathcal{V}| = 200, d = 20, p = 0.5, q = 0.05$

Figure 1: Synthetic data experiment showing the cumulative regret of Network Lasso Policy as a function of time-steps compared to other baselines, for different choices of $|\mathcal{V}|, d, p$ and $q$.

provide results for up to $|\mathcal{V}| = 500$ nodes showing the effective transfer of knowledge within the graph.

# 7 Conclusion and future perspectives

In this work, we proposed a multi-task bandit framework that solves the case where the task preference vectors are piecewise constant over a graph. To this end, we used the Network Lasso policy to estimate the task parameters, which bypasses explicit clustering procedures. We showed a sublinear regret bound and as a byproduct, we proved a novel oracle inequality that relies on the small size of the boundary as well as on the high value of the topological centrality index of each node within its cluster. Our experimental evaluations highlight the advantage of our method, especially when either the number of dimensions or nodes increases.

Due to the technical similarity of our problem with the Lasso, a natural extension would be to extend it to a thresholded approach, in the same vein as [Ariu et al., 2022]. Another possible extension would be to use regularization with higher order total variation terms that impose a piecewise polynomial signal on a graph, as explained for scalar signals in Wang et al. [2016], Ortelli and van de Geer [2019].

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

## A  Some helper results

434

435 **Proposition 1** (Bounds on norms of matrix products). *Let* $\mathbf{M} \in \mathbb{R}^{m \times n}$ *and* $\mathbf{N} \in \mathbb{R}^{n \times p}$. *Then*

$$\|\mathbf{M}\mathbf{N}\|_{q,1} \leq \|\mathbf{M}\|_{\infty,1}\|\mathbf{N}\|_{q,1} \quad \forall q \in [1,\infty]$$
$$\|\mathbf{M}\mathbf{N}\|_F \leq \|\mathbf{M}\|\|\mathbf{N}\|_F$$
$$\|\mathbf{M}\mathbf{N}\|_F \leq \sqrt{\|\mathbf{M}^\top \mathbf{M}\|_{\infty,\infty}}\|\mathbf{N}\|_{2,1}$$
$$\|\mathbf{M}\mathbf{N}\|_{2,1} \leq \|\mathbf{M}\|_{2,1}\|\mathbf{N}\|$$

436 *Proof.*

437 **First inequality**  For any $q \in [1,\infty]$, we have:

$$\left\|\mathbf{e}_i^\top \mathbf{M}\mathbf{N}\right\|_q = \left\|\mathbf{e}_i^\top \mathbf{M}\sum_{j=1}^n \mathbf{e}_j \mathbf{e}_j^\top \mathbf{N}\right\|_q \leq \max_{1 \leq j \leq n}\left|\mathbf{e}_i^\top \mathbf{M}\mathbf{e}_j\right|\sum_{j=1}^n \left\|\mathbf{e}_j^\top \mathbf{N}\right\|_q = \max_{1 \leq j \leq n}\left|(\mathbf{M})_{ij}\right|\|\mathbf{N}\|_{q,1}$$

438

439 **Second inequality**  We have

$$\|\mathbf{M}\mathbf{N}\|_F^2 = \sum_{j=1}^p \|\mathbf{M}\mathbf{N}\mathbf{e}_j\|^2 \leq \sum_{j=1}^p \|\mathbf{M}\|\|\mathbf{N}\mathbf{e}_j\|^2 = \|\mathbf{M}\|\|\mathbf{N}\|_F^2$$

440

441 **Third inequality**  We have

$$\|\mathbf{M}\mathbf{N}\|_F^2 = \mathrm{Tr}(\mathbf{M}\mathbf{N}\mathbf{N}^\top \mathbf{M}^\top) \leq \left\|\mathbf{M}^\top \mathbf{M}\right\|_{\infty,\infty}\left\|\mathbf{N}\mathbf{N}^\top\right\|_{1,1}$$

442 Elements of $(i,j)$ entry of matrix $\mathbf{N}\mathbf{N}^\top$ is the inner product $\langle \mathbf{e}_i^\top \mathbf{N}, \mathbf{e}_j^\top \mathbf{N}\rangle$. Hence, we have

$$\left\|\mathbf{N}\mathbf{N}^\top\right\|_{1,1} = \sum_{i,j}\left|\langle \mathbf{e}_i^\top \mathbf{N}, \mathbf{e}_j^\top \mathbf{N}\rangle\right| \leq \sum_{i,j}\left\|\mathbf{e}_i^\top \mathbf{N}\right\|\left\|\mathbf{e}_j^\top \mathbf{N}\right\| = \|\mathbf{N}\|_{2,1}^2.$$

443

444 **Fourth inequality**  We have

$$\|\mathbf{M}\mathbf{N}\|_{2,1} = \sum_{i=1}^m \|\mathbf{e}_i \mathbf{M}\mathbf{N}\| \leq \sum_{i=1}^m \|\mathbf{e}_i \mathbf{M}\|\|\mathbf{N}\| = \|\mathbf{M}\|_{2,1}\|\mathbf{N}\|$$

445 □

446 **Proposition 2** (Decomposition of a signal over a graph). *For any* $\mathcal{C} \in \mathcal{P}$

- *Let $\mathbf{Z} \in \mathbb{R}^{|\mathcal{V}|\times d}$ be a graph signal. Let us denote by $\mathbf{Z}_\mathcal{C}$ the signal obtained from $\mathbf{Z}$ by setting rows of vertices outside of $\mathcal{C}$ to zeros, and let $\mathbf{Z}_{|\mathcal{C}} \in \mathbb{R}^{|\mathcal{C}|\times d}$ be the signal obtained from $\mathbf{Z}_\mathcal{C}$ by removing the rows of vertices outside of $\mathcal{C}$. Also, let $\mathbf{B}_{|\mathcal{C}} \in \mathbb{R}^{|\mathcal{E}_\mathcal{C}|\times|\mathcal{C}|}$ be the matrix obtained by taking $\mathbf{B}_\mathcal{C}$, and removing rows of edges that link $\mathcal{C}$ to its outside, and the resulting null columns. It is clear that*

$$\mathbf{B}_\mathcal{C}\mathbf{Z} = \mathbf{B}_\mathcal{C}\mathbf{Z}_\mathcal{C} = \mathbf{B}_{|\mathcal{C}}\mathbf{Z}_{|\mathcal{C}} \tag{9}$$

- *Let $\mathbf{Q}_\mathcal{C} := \mathbf{B}_\mathcal{C}^\dagger \mathbf{B}_\mathcal{C}$. Then*

$$\mathbf{I}_{|\mathcal{V}|} = \sum_{\mathcal{C}\in\mathcal{P}} \mathbf{J}_\mathcal{C} + \mathbf{Q}_\mathcal{C} \tag{10}$$

$$\mathbf{Q}_{\partial\mathcal{P}^c} :== \mathbf{B}_{\partial\mathcal{P}^c}^\dagger \mathbf{B}_{\partial\mathcal{P}^c} = \sum_{\mathcal{C}\in\mathcal{P}} \mathbf{Q}_\mathcal{C} \tag{11}$$

  *where $\mathbf{J}_\mathcal{C} = \frac{\mathbf{1}_\mathcal{C}\mathbf{1}_\mathcal{C}^\top}{|\mathcal{C}|}$, $\mathbf{Q}_\mathcal{C} = \mathbf{B}_\mathcal{C}^\dagger \mathbf{B}_\mathcal{C}$ $\quad \forall \mathcal{C} \in \mathcal{P}$ and $\mathbf{Q}_{\partial\mathcal{P}^c} := \mathbf{B}_{\partial\mathcal{P}^c}^\dagger \mathbf{B}_{\partial\mathcal{P}^c}$.*

  *While $\sum_{\mathcal{C}\in\mathcal{P}} \mathbf{J}_\mathcal{C}$ projects each entry of a graph signal onto the mean vector value of its respective cluster, its residual $\mathbf{Q}_{\partial\mathcal{P}^c}$ can be interpreted as the projection onto the respective entries deviation from its cluster mean value.*

*Proof.* Since the proof of the first point is trivial, we directly treat the second point. Denoting $\mathbf{B}_{|\mathcal{C}}^\dagger$ the pseudo-inverse of $\mathbf{B}_{|\mathcal{C}}$ it is a well-known linear algebra result that the matrix $Q_{|\mathcal{C}} := \mathbf{B}_{|\mathcal{C}}^\dagger \mathbf{B}_{|\mathcal{C}}$ is the projector onto the null space of $\mathbf{B}_{|\mathcal{C}}$. Since $\mathcal{C}$ is connected, the null space of $\mathbf{B}_{|\mathcal{C}}$ is unidimensional, and is generated by vector $\mathbf{1}_{|\mathcal{C}|} \in \mathbb{R}^{|\mathcal{C}|}$ having only ones as coordinates. Since the projector into that nullspace is $\mathbf{J}_{|\mathcal{C}|} := \frac{\mathbf{1}_{|\mathcal{C}|}\mathbf{1}_{|\mathcal{C}|}}{|\mathcal{C}|}$, we deduce that

$$\mathbf{Z}_{|\mathcal{C}} = \mathbf{J}_{|\mathcal{C}|}\mathbf{Z}_{|\mathcal{C}} + \mathbf{Q}_{|\mathcal{C}}\mathbf{Z}_{|\mathcal{C}}$$
$$\implies \mathbf{Z}_\mathcal{C} = \mathbf{J}_\mathcal{C}\mathbf{Z}_\mathcal{C} + \mathbf{Q}_\mathcal{C}\mathbf{Z}_\mathcal{C}$$
$$= \mathbf{J}_\mathcal{C}\mathbf{Z} + \mathbf{Q}_\mathcal{C}\mathbf{Z}$$

where in the last line, $Q_\mathcal{C} := \mathbf{B}_\mathcal{C}^\dagger \mathbf{B}_\mathcal{C}$. Consequently, we have

$$\mathbf{Z} = \sum_{\mathcal{C}\in\mathcal{P}} \mathbf{Z}_\mathcal{C}$$
$$= \sum_{\mathcal{C}\in\mathcal{P}} \mathbf{J}_\mathcal{C}\mathbf{Z} + \mathbf{Q}_\mathcal{C}\mathbf{Z}$$

To prove the second point, we recall that $\mathbf{B}_{\partial\mathcal{P}^c}$ is the incidence matrix obtained by setting rows corresponding to edges in $\partial\mathcal{P}$ to zero. In other words, $\mathbf{B}_{\partial\mathcal{P}^c}$ is the incidence matrix of the graph after removing the boundary edges, and having exactly $|\mathcal{P}|$ connected components. Hence, $\mathbf{B}_{\partial\mathcal{P}^c}$ has a null space spanned by the set $\{\mathbf{1}_\mathcal{C}\}_{\mathcal{C}\in\mathcal{P}}$, and the orthogonal projector onto this null space is $\sum_{\mathcal{C}\in\mathcal{P}} \mathbf{J}_\mathcal{C}$. Combining this fact with the fact that $\mathbf{Q}_{\partial\mathcal{P}^c}$ is the projector onto the orthogonal of the null space of $\mathbf{B}_{\partial\mathcal{P}^c}$, we arrive at the second point. $\square$

**Proposition 3** (On the minimum topological centrality index of a graph vertex)**.** *Let $\mathcal{G}$ be a connected graph with incidence matrix $\mathbf{B}$ and vertex set size $N$, and let $\mathbf{L} := \mathbf{B}^\top\mathbf{B}$. Let $c(\mathcal{G})$ denote the minimum value of inverses of diagonal element of $\mathbf{L}^\dagger$, called its minimum topological centrality index. Also let $a(\mathcal{G})$ be its algebraic connectivity, defined as the minimum non null eigenvalue of $\mathbf{L}$. Then*

- $c(\mathcal{G}) = \|\mathbf{L}\|_{\infty,\infty}^{-1}.$

- $c(\mathcal{G}) \geq a(\mathcal{G}).$

- *If $\mathcal{G}$ is weightless, then $c(\mathcal{G}) \leq \frac{N^2}{N-1}.$*

*Proof.* Since $\mathbf{L}$ is PSD, $\mathbf{L}^\dagger$ is PSD and hence $\left\|\mathbf{L}^\dagger\right\|_{\infty,\infty}$ is equal to the maximum diagonal entry of $\mathbf{L}^\dagger$. Taking the inverse proves the first point. Also, this implies that

$$c(\mathcal{G}) = \left\|\mathbf{L}^\dagger\right\|_{\infty,\infty}^{-1} \geq \left\|\mathbf{L}^\dagger\right\|^{-1} = a(\mathcal{G}), \tag{12}$$

where we used the fact that $\|\cdot\|_{\infty,\infty} \leq \|\cdot\|$ for matrices. This proves the second point of the proposition.

For the last point, assume $\mathcal{G}$ is weightless, let $\mathbf{L}_{\mathrm{comp}}$ be the Laplaciane of complete graph built on the vertices of $\mathcal{G}$. Then we have $\mathbf{L}_{\mathrm{comp}} = N(\mathbf{I}_N - \mathbf{J}_N)$, where $J$ is the square matrix of dimension $N$ having $1/N$ as entries. From Fontan and Altafini [2021, Lemma 4], we have

$$\mathbf{L}_{\mathrm{comp}}^\dagger = (\mathbf{L}_{\mathrm{comp}} + N\mathbf{J}_N)^{-1} - \frac{1}{N}\mathbf{J}_N = \frac{\mathbf{I}_N}{N} - \frac{1}{N}\mathbf{J}_N \tag{13}$$

which has diagonal elements $\frac{1}{N} - \frac{1}{N^2}$.

On the other hand, $\mathbf{L} \preccurlyeq \mathbf{L}_{\mathrm{comp}}$ Hence, by Fontan and Altafini [2021, lemma 4] we have for any $u \neq 0$

$$\mathbf{L}^\dagger = (\mathbf{L} + a\mathbf{J}_N)^{-1} - \mathbf{J}_N/a \succcurlyeq (\mathbf{L}_{\mathrm{comp}} + a\mathbf{J}_N)^{-1} - \mathbf{J}_N/a = \mathbf{L}_{\mathrm{comp}}^\dagger$$

This implies that the maximum diagonal entry of $\mathbf{L}^\dagger$ is at least equal to that of $\mathbf{L}_{\mathrm{comp}}^\dagger$, *i.e.* to $\frac{1}{N} - \frac{1}{N^2}$. Taking the inverse of that entry finishes the proof.

$\square$

# B  Proofs of the different claims

## B.1  Additional notation

The regularization term can be written more compactly using the incidence matrix of the graph $\mathbf{B} \in \mathbb{R}^{|\mathcal{E}| \times |\mathcal{V}|}$ corresponding to an arbitrary orientation under the following form

$$\sum_{1 \leq m < n \leq |\mathcal{V}|} w_{mn} \|\boldsymbol{\theta}_m - \boldsymbol{\theta}_n\| = \|\mathbf{B}\boldsymbol{\Theta}\|_{2,1} = \|\boldsymbol{\Theta}\|_{\mathcal{E}} \tag{14}$$

where the $\|\cdot\|_{2,1}$ norm denotes the sum of the $L_2$ norms o the rows of a matrix.[1] We provide notations that we use in the proofs of the different statements, in order to reduce the clutter. We define $\mathbf{E} := \hat{\boldsymbol{\Theta}} - \boldsymbol{\Theta}$ as the error signal, and its rows by $\{\boldsymbol{\epsilon}_m\}_{m=1}^{|\mathcal{V}|}$.

While $\sum_{k=1}^C \mathbf{J}_{\mathcal{C}}$ projects each entry of a graph signal onto the mean vector value of its respective cluster, its residual $\mathbf{Q}_{\partial \mathcal{P}^c}$ can be interpreted as the projection onto the respective entries deviation from its cluster mean value.

Let $\boldsymbol{\eta}_m$ be a vector, vertically concatenated by noise terms of rewards received by node $m$, then we define $\mathbf{K} \in \mathbb{R}^{|\mathcal{V}| \times d}$ as the matrix of vertically concatenated row vectors $\boldsymbol{\eta}_m^\top \mathbf{X}_m$.

## B.2  Oracle inequality

In this section, we present all intermediary theoretical results leading to Theorem 1 stating the oracle inequality. To reduce the clutter, we omit the dependence on $t$ of several quantities. For instance, we write $\alpha$ and $\hat{\boldsymbol{\Theta}}$ instead of $\alpha(t)$ and $\hat{\boldsymbol{\Theta}}(t)$.

**Lemma 1** (A first deterministic inequality). *Let $t$ be a time step. We have*

$$\frac{1}{2t\alpha} \sum_{m \in \mathcal{V}} \|\mathbf{X}_m \boldsymbol{\epsilon}_m\|^2 + \|\mathbf{E}\|_{\partial \mathcal{P}^c} \leq \frac{1}{t\alpha} \langle \mathbf{K}, \mathbf{E} \rangle + \|\mathbf{E}\|_{\partial \mathcal{P}} \tag{15}$$

---

[1] It is possible that the notation $\|\cdot\|_{2,1}$ denotes the sum of $2-$norms of columns in the literature.

| Notation | Meaning |
|---|---|
| | Indpendent of time $t$ |
| $\mathcal{V}$ | set of graph vertices |
| $\mathcal{E}$ | set of graph edges |
| $\mathbf{B}_I \in \mathbb{R}^{|\mathcal{E}|\times|\mathcal{V}|}, I \subseteq \mathcal{E}$ | Graph incidence Matrix obtained by setting rows of edges outside $I$ to zeros |
| $\mathbf{B}_{\mathcal{C}} \in \mathbb{R}^{|\mathcal{E}|\times|\mathcal{V}|}$ | cf. Definition 1 |
| $\mathbf{L} \in \mathbb{R}^{|\mathcal{V}|\times|\mathcal{V}|}$ | $\mathbf{B}^\top \mathbf{B}$ |
| $\boldsymbol{\theta}_m \in \mathbb{R}^d$ | true preference vector of user/bandit $m$ |
| $\boldsymbol{\Theta} \in \mathbb{R}^{|\mathcal{V}|\times d}$ | matrix of true vertically concatenated row preferences vectors |
| $\partial\mathcal{P} \subseteq \mathcal{E}$ | Boundary of $\mathcal{P}$: set of edges connecting nodes from different clusters |
| $c_{\mathcal{G}}(\mathcal{C})$ | Minimum topological centrality index of a node of $\mathcal{C}$ restricted to the graph having nodes $\mathcal{C}$ |
| $w(\partial\mathcal{P})$ | Total weight of $\partial\mathcal{P}$, *i.e.* sum of weights of edges in $\mathcal{P}$ |
| $\|\cdot\|$ | Euclidean norm for vectors, largest singular value for matrices |
| $\|\cdot\|_{\mathbf{A}}$ | Semi-norm associated defined by PSD matrix $\mathbf{A}$: $\|\mathbf{x}\|_{\mathbf{A}}^2 := \mathbf{x}^\top \mathbf{A}\mathbf{x}$ |
| $\|\cdot\|_F$ | matrix Frobenius norm |
| $\|\cdot\|_{p,q}$ | $q$-norm of the vector with coordinates equal to the $p-$norm of rows |
| $\|\cdot\|_I, I \subseteq \mathcal{E}$ | Total variation norm of signal over edges of $I$ |
| $\mathbf{A}^\dagger$ | Moore-Penrose pseudo-inverse of matrix $\mathbf{A}$ |
| vec | vectorization operator consisting in concatenating the columns vertically |
| $\otimes$ | Kronecker product |
| $\mathbf{1}_{\mathcal{C}} \in \mathbb{R}^{|\mathcal{V}|}$ | Vector having elements equal to 1 at coordinates corresponding to vertices in $\mathcal{C}$ and 0 elsewhere |
| $\mathbf{J}_{\mathcal{C}} \in \mathbb{R}^{|\mathcal{V}|\times|\mathcal{V}|}$ | equal to $\frac{\mathbf{1}_{\mathcal{C}}\mathbf{1}_{\mathcal{C}}^\top}{|\mathcal{C}|}$ |
| $\mathbf{Q}_{\mathcal{C}} \in \mathbb{R}^{|\mathcal{V}|\times|\mathcal{V}|}$ | equal to $\mathbf{B}_{\mathcal{C}}^\dagger \mathbf{B}_{\mathcal{C}}$ |
| $\mathbf{Q}_I \in \mathbb{R}^{|\mathcal{V}|\times|\mathcal{V}|}, I \subseteq \mathcal{E}$ | equal to $\mathbf{B}_I^\dagger \mathbf{B}_I$ |
| $\mathbf{e}_k$ | elementary vectors of dimension depending on the context |
| $\sigma$ | Subgaussianity constant / variance proxy |
| | Dependent on time $t$ |
| $\mathcal{T}_m(t)$ | set of time steps user $m$ has been encountered before time $t$ |
| $\hat{\boldsymbol{\theta}}_m \in \mathbb{R}^d$ | estimated preference vector of user/bandit $m$ |
| $\boldsymbol{\epsilon}_m \in \mathbb{R}^d$ | estimation error for user/bandit $m : \hat{\boldsymbol{\theta}}_m - \boldsymbol{\theta}_m$ |
| $\mathbf{E} \in \mathbb{R}^{|\mathcal{V}|\times d}$ | vertical concatenation of row vectors $\boldsymbol{\epsilon}_m$ |
| $\boldsymbol{\eta}_m \in \mathbb{R}^{|\mathcal{T}_m(t)|}$ | vector of subgaussian noise of user $m$ |
| $\mathbf{x}(t) \in \mathbb{R}^d$ | context vector received at time $t$ |
| $m(t) \in \mathbb{N}$ | user at time $t$ |
| $\mathbf{X}_m \in \mathbb{R}^{|\mathcal{T}_m(t)|\times d}$ | data matrix of user $m$ |
| $\mathbf{X} \in \mathbb{R}^{t\times d}$ | data matrix of context vectors of all users |
| $\mathbf{A}_m \in \mathbb{R}^{d\times d}$ | $\mathbf{X}_m^\top \mathbf{X}_m$ (potentially associated to time $t$) |
| $\mathbf{A}_{\mathcal{V}} \in \mathbb{R}^{d|\mathcal{V}|\times d|\mathcal{V}|}$ | $\mathrm{diag}(\mathbf{A}_1, \cdots, \mathbf{A}_m)$ |
| $\mathbf{K} \in \mathbb{R}^{|\mathcal{V}|\times d}$ | matrix of vertically concatenated row vectors $\boldsymbol{\eta}_m^\top \mathbf{X}_m$ |

Table 1: Notation table.

506   *Proof.* By optimality of $\hat{\boldsymbol{\Theta}}$, we have

$$\frac{1}{2t}\sum_{m\in\mathcal{V}}\left\|\mathbf{X}_m\hat{\boldsymbol{\theta}}_m - \mathbf{y}_m\right\|^2 + \alpha\|\boldsymbol{\Theta}\|_{\mathcal{E}} \leq \frac{1}{2t}\sum_{m\in\mathcal{V}}\|\mathbf{X}_m\boldsymbol{\theta}_m - \mathbf{y}_m\|^2 + \alpha\|\boldsymbol{\Theta}\|_{\mathcal{E}} \qquad (16)$$

507   where the second line holds by definition of the observed rewards.

On the one hand, given a user index $m \in \mathcal{V}$, and since by definition of the observed rewards we have we have for the least squared terms

$$
\begin{aligned}
\left\| \mathbf{X}_m \hat{\boldsymbol{\theta}}_m - \mathbf{y}_m \right\|^2 &= \left\| \mathbf{X}_m \hat{\boldsymbol{\theta}}_m - \mathbf{X}_m \boldsymbol{\theta}_m - \boldsymbol{\eta}_m \right\|^2 \\
&= \left\| \mathbf{X}_m \boldsymbol{\epsilon}_m - \boldsymbol{\eta}_m \right\|^2 \\
&= \left\| \mathbf{X}_m \boldsymbol{\epsilon}_m \right\|^2 + \left\| \mathbf{X}_m \boldsymbol{\theta}_m - \mathbf{y}_m \right\|^2 - \boldsymbol{\eta}_m^\top \mathbf{X}_m \boldsymbol{\epsilon}_m
\end{aligned}
$$

where we used the fact that $\mathbf{y}_m = \mathbf{X}_m \boldsymbol{\theta}_m + \boldsymbol{\eta}_m$, which holds by definition of the observed rewards. Summing over the users, and using the definition of $\mathbf{K}$, we have

$$
\frac{1}{2t} \sum_{m \in \mathcal{V}} \left\| \mathbf{X}_m \hat{\boldsymbol{\theta}}_m - \mathbf{y}_m \right\|^2 - \frac{1}{2t} \sum_{m \in \mathcal{V}} \left\| \mathbf{X}_m \boldsymbol{\theta}_m - \mathbf{y}_m \right\|^2 = \frac{1}{2t} \sum_{m \in \mathcal{V}} \left\| \mathbf{X}_m \boldsymbol{\epsilon}_m \right\|^2 - \frac{1}{t} \langle \mathbf{K}, \mathbf{E} \rangle \quad (17)
$$

On the other hand, we have for the estimated preference vectors

$$
\begin{aligned}
\|\boldsymbol{\Theta}\|_{\mathcal{E}} &= \sum_{(m,n) \in \mathcal{E}} w_{mn} \left\| \hat{\boldsymbol{\theta}}_m - \hat{\boldsymbol{\theta}}_n \right\| \\
&= \sum_{(m,n) \in \partial \mathcal{P}} w_{mn} \left\| \hat{\boldsymbol{\theta}}_m - \hat{\boldsymbol{\theta}}_n \right\| + \sum_{(m,n) \in \partial \mathcal{P}^c} w_{mn} \left\| \hat{\boldsymbol{\theta}}_m - \hat{\boldsymbol{\theta}}_n \right\| \\
&= \left\| \hat{\boldsymbol{\Theta}} \right\|_{\partial \mathcal{P}} + \left\| \hat{\boldsymbol{\Theta}} \right\|_{\partial \mathcal{P}^c},
\end{aligned}
$$

For the true ones, and for any $\mathcal{C} \in \mathcal{P}$, let $\mathcal{E}_{\mathcal{C}}$ denote the edges linking the nodes of set of nodes $\mathcal{C}$. It is clear that $\partial \mathcal{P}^c = \bigcup_{\mathcal{C} \in \mathcal{P}} \mathcal{E}_{\mathcal{C}}$ as a disjoint union, hence

$$
\begin{aligned}
\|\boldsymbol{\Theta}\|_{\mathcal{E}} &= \sum_{(m,n) \in \mathcal{E}} w_{mn} \|\boldsymbol{\theta}_m - \boldsymbol{\theta}_n\| \\
&= \sum_{(m,n) \in \partial \mathcal{P}} w_{mn} \|\boldsymbol{\theta}_m - \boldsymbol{\theta}_n\| + \sum_{(m,n) \in \partial \mathcal{P}^c} w_{mn} \|\boldsymbol{\theta}_m - \boldsymbol{\theta}_n\| \\
&= \|\boldsymbol{\Theta}\|_{\partial \mathcal{P}} + \sum_{\mathcal{C} \in \mathcal{P}} \sum_{(m,n) \in \mathcal{E}_{\mathcal{C}}} w_{mn} \|\boldsymbol{\theta}_m - \boldsymbol{\theta}_n\| \\
&= \|\boldsymbol{\Theta}\|_{\partial \mathcal{P}}
\end{aligned}
$$

where the last equality holds due to the cluster assumption.

Hence, we have

$$
\begin{aligned}
\|\boldsymbol{\Theta}\|_{\mathcal{E}} - \|\boldsymbol{\Theta}\|_{\mathcal{E}} &= \|\boldsymbol{\Theta}\|_{\partial \mathcal{P}} - \left\| \hat{\boldsymbol{\Theta}} \right\|_{\partial \mathcal{P}} - \left\| \hat{\boldsymbol{\Theta}} \right\|_{\partial \mathcal{P}^c} \\
&\leq \|\mathbf{E}\|_{\partial \mathcal{P}} - \left\| \hat{\boldsymbol{\Theta}} \right\|_{\partial \mathcal{P}^c},
\end{aligned} \quad (18)
$$

where the first inequality holds due to the triangle inequality, and the last one since $\|\boldsymbol{\Theta}\|_{\partial \mathcal{P}^c} = 0$. Combining Equations (16) to (18), we obtain the result of the statement. $\qquad \square$

In the proof for the oracle inequality, we utilize projection operators on the graph signal, that we define as followed:

While $\sum_{k=1}^C \mathbf{J}_{\mathcal{C}}$ projects each entry of a graph signal onto the mean vector value of its respective cluster, its residual $\mathbf{Q}_{\partial \mathcal{P}^c}$ can be interpreted as the projection onto the respective entries deviation from its cluster mean value.

**Lemma 2** (Bounding the error restricted to the boundary). *The total variation of* $\mathbf{E}$ *restricted to the boundary verifies*

$$
\|\mathbf{E}\|_{\partial \mathcal{P}} \leq w(\partial \mathcal{P}) \left( \sqrt{2} \max_{\mathcal{C} \in \mathcal{P}} \sqrt{\iota_{\mathcal{G}}(\mathcal{C})} \left\| \overline{\mathbf{E}}_{\mathcal{P}} \right\|_F + 2 \frac{\|\mathbf{E}\|_{\partial \mathcal{P}^c}}{\min_{\mathcal{C} \in \mathcal{P}} \sqrt{c_{\mathcal{G}}(\mathcal{C})}} \right) \quad (19)
$$

*Proof.* The proof relies on a decomposition of the $\|\mathbf{E}\|_{\partial\mathcal{P}}$ term from Proposition 2. We have

$$
\begin{aligned}
\|\mathbf{E}\|_{\partial\mathcal{P}} &= \left\| \sum_{\mathcal{C}\in\mathcal{P}} \mathbf{J}_{\mathcal{C}}\mathbf{E} + \mathbf{Q}_{\mathcal{C}}\mathbf{E} \right\|_{\partial\mathcal{P}} \\
&= \left\| \overline{\mathbf{E}}_{\mathcal{P}} + \mathbf{B}_{\partial\mathcal{P}^c}^{\dagger}\mathbf{B}_{\partial\mathcal{P}^c}\mathbf{E} \right\|_{\partial\mathcal{P}} \\
&\leq \left\| \overline{\mathbf{E}}_{\mathcal{P}} \right\|_{\partial\mathcal{P}} + \left\| \mathbf{B}_{\partial\mathcal{P}^c}^{\dagger}\mathbf{B}_{\partial\mathcal{P}^c}\mathbf{E} \right\|_{\partial\mathcal{P}}
\end{aligned}
\tag{20}
$$

where $\overline{\mathbf{E}}_{\mathcal{P}}$ is obtained by setting the error signal on every cluster to its mean.

For the first term on the right-hand side, let us denote by $\epsilon_{\mathcal{C}}$ the value of any row of $\overline{\mathbf{E}}_{\mathcal{P}}$ belonging to cluster $\mathcal{C}$, which is equal to the mean of errors $\mathbf{E}$ over that cluster. Also, we denote by $(\overline{\mathbf{E}}_{\mathcal{P}})_{\partial\mathcal{P}}$ the signal obtained from $\overline{\mathbf{E}}_{\mathcal{P}}$ by setting its rows corresponding to nodes that are not adjacent to any edge in the boundary $\partial\mathcal{P}$ to zeros. Also, let $\partial_v\mathcal{C}$ denote the inner boundary of set of nodes $\mathcal{C}$,i.e. nodes of $\mathcal{C}$ that connect it to its complementary. Then it holds that:

$$
\begin{aligned}
\left\| \overline{\mathbf{E}}_{\mathcal{P}} \right\|_{\partial\mathcal{P}} &= \left\| \mathbf{B}_{\partial\mathcal{P}}\overline{\mathbf{E}}_{\mathcal{P}} \right\|_{2,1} \\
&= \left\| \mathbf{B}_{\partial\mathcal{P}}(\overline{\mathbf{E}}_{\mathcal{P}})_{\partial\mathcal{P}} \right\|_{2,1} \\
&\leq \left\| \mathbf{B}_{\partial\mathcal{P}} \right\|_{2,1} \left\| (\overline{\mathbf{E}}_{\mathcal{P}})_{\partial\mathcal{P}} \right\| \quad \text{(by Proposition 1)} \\
&\leq \left\| \mathbf{B}_{\partial\mathcal{P}} \right\|_{2,1} \left\| (\overline{\mathbf{E}}_{\mathcal{P}})_{\partial\mathcal{P}} \right\|_F \\
&= \left\| \mathbf{B}_{\partial\mathcal{P}} \right\|_{2,1} \sqrt{\sum_{\mathcal{C}\in\mathcal{P}} |\partial_v\mathcal{C}| \|\epsilon_{\mathcal{C}}\|^2} \\
&= \left\| \mathbf{B}_{\partial\mathcal{P}} \right\|_{2,1} \sqrt{\sum_{\mathcal{C}\in\mathcal{P}} \frac{|\partial_v\mathcal{C}|}{|\mathcal{C}|}|\mathcal{C}|\|\epsilon_{\mathcal{C}}\|^2} \\
&\leq \left\| \mathbf{B}_{\partial\mathcal{P}} \right\|_{2,1} \max_{\mathcal{C}\in\mathcal{P}} \sqrt{\iota_{\mathcal{G}}(\mathcal{C})} \sqrt{\sum_{\mathcal{C}\in\mathcal{P}} |\mathcal{C}|\|\epsilon_{\mathcal{C}}\|^2} \\
&= \sqrt{2}w(\partial\mathcal{P}) \max_{\mathcal{C}\in\mathcal{P}} \sqrt{\iota_{\mathcal{G}}\mathcal{C}} \left\| \overline{\mathbf{E}}_{\mathcal{P}} \right\|_F
\end{aligned}
\tag{21}
$$

For the second term, we have

$$
\begin{aligned}
\left\| \mathbf{B}_{\partial\mathcal{P}^c}^{\dagger}\mathbf{B}_{\partial\mathcal{P}^c}\mathbf{E} \right\|_{\partial\mathcal{P}} &= \left\| \mathbf{B}_{\partial\mathcal{P}}\mathbf{B}_{\partial\mathcal{P}^c}^{\dagger}\mathbf{B}_{\partial\mathcal{P}^c}\mathbf{E} \right\|_{2,1} \\
&\leq \left\| \mathbf{B}_{\partial\mathcal{P}}\mathbf{B}_{\partial\mathcal{P}^c}^{\dagger} \right\|_{\infty,1} \|\mathbf{E}\|_{\partial\mathcal{P}^c} \\
&\leq \left\| \mathbf{B}_{\partial\mathcal{P}}\mathbf{B}_{\partial\mathcal{P}^c}^{\dagger} \right\|_F \|\mathbf{E}\|_{\partial\mathcal{P}^c} \\
&\leq \left\| (\mathbf{B}_{\partial\mathcal{P}^c}^{\dagger})^{\top}\mathbf{B}_{\partial\mathcal{P}}^{\top} \right\|_F \|\mathbf{E}\|_{\partial\mathcal{P}^c} \\
&\leq \left\| \mathbf{B}_{\partial\mathcal{P}}^{\top} \right\|_{2,1} \sqrt{\left\| \mathbf{B}_{\partial\mathcal{P}^c}^{\dagger}(\mathbf{B}_{\partial\mathcal{P}^c}^{\dagger})^{\top} \right\|_{\infty,\infty}} \|\mathbf{E}\|_{\partial\mathcal{P}^c} \quad \text{(by Proposition 1)} \\
&= \frac{\left\| \mathbf{B}_{\partial\mathcal{P}}^{\top} \right\|_{1,1}}{\min_{\mathcal{C}\in\mathcal{P}} \sqrt{c_{\mathcal{G}}(\mathcal{C})}} \|\mathbf{E}\|_{\partial\mathcal{P}^c}. \\
&= 2\frac{w(\partial\mathcal{P})}{\min_{\mathcal{C}\in\mathcal{P}} \sqrt{c_{\mathcal{G}}(\mathcal{C})}} \|\mathbf{E}\|_{\partial\mathcal{P}^c}.
\end{aligned}
\tag{22}
$$

The result is obtained by combining Equations (20) to (22). $\qquad\square$

**Theorem 4** (Theorem 2.1 of Hsu et al. [2012]). *At time step $t$, let $\mathbf{A}\in\mathbb{R}^{b\times t}$ where $b\in\mathbb{N}^*$, and let $\mathbf{v}\in\mathbb{R}^t$ be a random vector such that for some $\sigma\geq 0$, we have*

$$\mathbb{E}\left[\exp(\langle \mathbf{u}, \mathbf{v}\rangle)\right] \leq \exp\left(\|\mathbf{u}\|^2 \frac{\sigma^2}{2}\right) \quad \forall \mathbf{u} \in \mathbb{R}^t.$$

*Then for any $\delta \in (0,1)$, we have with a probability at least $1 - \delta$:*

$$\|\mathbf{A}\mathbf{v}\|^2 \leq \sigma^2 \left(\|\mathbf{A}\|_F^2 + 2\|\mathbf{A}^\top \mathbf{A}\|_F \sqrt{\log \frac{1}{\delta}} + 2\|\mathbf{A}\|^2 \log \frac{1}{\delta}\right).$$

**Lemma 3** (Empirical process bound). *Let $\mathbf{X}_m \in \mathbb{R}^{|\mathcal{T}_m| \times d}$ denotes the matrix of collected context vectors for task $m \in \mathcal{V}$, then, given collected context matrices $\{\mathbf{X}_m\}_{m \in \mathcal{V}}$, for any $\delta \in (0,1)$ we have with probability of at least $1 - \delta$:*

$$\|\mathbf{K}\|_F \leq \frac{\alpha_\delta(t)}{\alpha_0} t,$$

*where*

$$\alpha_\delta(t) := \frac{\alpha_0 \sigma}{t} \sqrt{t + 2\sqrt{\sum_{m \in \mathcal{V}} |\mathcal{T}_m(t)|^2 \log \frac{1}{\delta}} + 2 \max_{m \in \mathcal{V}} |\mathcal{T}_m(t)| \log \frac{1}{\delta}}, \tag{23}$$

*Proof.* We recall that $\mathbf{K} \in \mathbb{R}^{t \times d}$ is the matrix obtained by stacking the row vectors $\boldsymbol{\eta}_m^\top \mathbf{X}_m$ vertically. On the one hand, we have

$$\|\mathbf{K}\|_F^2 = \sum_{m \in \mathcal{V}} \left\|\mathbf{X}_m^\top \boldsymbol{\eta}_m\right\|^2 = \left\|\mathbf{X}_{\mathcal{V}}^\top \boldsymbol{\eta}\right\|^2, \tag{24}$$

where $\mathbf{X}_{\mathcal{V}} := \text{diag}(\mathbf{X}_1, \cdots, \mathbf{X}_{|\mathcal{V}|}) \in \mathbb{R}^{t \times d|\mathcal{V}|}$.

On the other one, for any $\mathbf{u} = (u_1, \cdots, u_t) \in \mathbb{R}^t$, denoting $P(t) := \exp\left(\sum_{\tau=1}^t u_\tau \eta_\tau\right)$, we have

$$\begin{aligned}
\mathbb{E}\left[P(t)\right] &= \mathbb{E}\left[\mathbb{E}\left[\exp\{u_t \eta_t\} P(t-1) | \mathcal{F}_{t-1}\right]\right] \quad \text{(by the law of total expectation)} \\
&= \mathbb{E}\left[P(t-1)\mathbb{E}\left[\exp(u_t \eta_t)|\mathcal{F}_{t-1}\right]\right] \quad \text{(because $\{\eta_s\}_{s=1}^{t-1}$ are $\mathcal{F}_{t-1}$ measurable.)} \\
&\leq \exp\left(\frac{1}{2}\sigma^2 u_t^2\right)\mathbb{E}\left[P(t-1)\right] \quad \text{(by the conditional subgaussianity assumption)} \\
&\leq \prod_{s=1}^t \exp\left(\frac{1}{2}\sigma^2 u_s^2\right) \quad \text{(by induction)} \\
&= \exp\left(\frac{1}{2}\sigma^2 \|\mathbf{u}\|^2\right). \tag{25}
\end{aligned}$$

From Equations (24) and (25), we can apply Theorem 4 to matrix $\mathbf{X}_{\mathcal{V}}$ and random vector $\boldsymbol{\eta}$, which implies that with a probability at least $1 - \delta$, we have

$$\|\mathbf{X}_{\mathcal{V}}\boldsymbol{\eta}\| \leq \sigma \sqrt{\text{Tr}\left(\sum_{m \in \mathcal{V}} \mathbf{A}_m\right) + 2\sqrt{\sum_{m \in \mathcal{V}} \|\mathbf{A}_m\|_F^2 \log \frac{1}{\delta}} + 2 \max_{m \in \mathcal{V}} \|\mathbf{A}_m\| \log \frac{1}{\delta}},$$

where we used the equalities $\|\mathbf{X}_{\mathcal{V}}\|_F = \sum_{m \in \mathcal{V}} \text{Tr}(\mathbf{A}_m)$, $\|\mathbf{X}_{\mathcal{V}}\|^2 = \max_{m \in \mathcal{V}} \|\mathbf{A}_m\|$ and $\left\|\mathbf{X}_{\mathcal{V}}\mathbf{X}_{\mathcal{V}}^\top\right\|_F^2 = \left\|\mathbf{X}_{\mathcal{V}}^\top \mathbf{X}_{\mathcal{V}}\right\|_F^2 = \sum_{m \in \mathcal{V}} \|\mathbf{A}_m\|_F^2$. To arrive the the statement of the theorem, we use the fact that the context vectors have Euclidean norms of at most 1.

$\square$

**Proposition 4** (Probabilistic inequality). *With a probabability at least $1 - \delta$, we have*

$$\frac{1}{2t\alpha} \sum_{m \in \mathcal{V}} \|\mathbf{X}_m \boldsymbol{\epsilon}_m\|^2 + a_1(\mathcal{G}, \boldsymbol{\Theta})\|\mathbf{E}\|_{\partial \mathcal{P}^c} \leq a_2(\mathcal{G}, \boldsymbol{\Theta})\left\|\overline{\mathbf{E}}_{\mathcal{P}}\right\|_F + (1-\kappa)\|\mathbf{E}\|_{\partial \mathcal{P}}, \tag{26}$$

553 *where* $0 \le \kappa < \frac{\min\limits_{\mathcal{C} \in \mathcal{P}} \sqrt{c_{\mathcal{G}}(\mathcal{C})}}{2w(\partial\mathcal{P})}$, $\frac{1}{\alpha_0} < \min\limits_{\mathcal{C} \in \mathcal{P}} \sqrt{c_{\mathcal{G}}(\mathcal{C})} - 2\kappa w(\partial\mathcal{P})$ *and*

$$a_1(\mathcal{G}, \mathbf{\Theta}) = 1 - \frac{\frac{1}{\alpha_0} + 2\kappa w(\partial\mathcal{P})}{\min\limits_{\mathcal{C} \in \mathcal{P}} \sqrt{c_{\mathcal{G}}(\mathcal{C})}} \tag{27}$$

$$a_2(\mathcal{G}, \mathbf{\Theta}) = \frac{1}{\alpha_0} + \sqrt{2}\kappa w(\partial\mathcal{P}) \max\limits_{\mathcal{C} \in \mathcal{P}} \sqrt{\iota_{\mathcal{G}}(\mathcal{C})}. \tag{28}$$

554 *Proof.* The proof is a combination of the results of Lemmas 1 to 3. We have

$$\frac{1}{2t\alpha_\delta} \sum_{m \in \mathcal{V}} \|\mathbf{X}_m \boldsymbol{\epsilon}_m\|^2 + \|\mathbf{E}\|_{\partial\mathcal{P}^c}$$

$$\le \frac{1}{t\alpha_\delta} \langle \mathbf{K}, \mathbf{E} \rangle + \|\mathbf{E}\|_{\partial\mathcal{P}} \quad \text{(by Lemma 1)}$$

$$\le \frac{1}{\alpha_0} \|\mathbf{E}\|_F + \kappa \|\mathbf{E}\|_{\partial\mathcal{P}} + (1-\kappa)\|\mathbf{E}\|_{\partial\mathcal{P}} \quad \text{(by Lemma 3)}$$

$$\le \frac{\|\overline{\mathbf{E}}_{\mathcal{P}}\|_F}{\alpha_0} + \frac{\|\mathbf{E}\|_{\partial\mathcal{P}^c}}{\alpha_0 \min\limits_{\mathcal{C} \in \mathcal{P}} \sqrt{c_{\mathcal{G}}(\mathcal{C})}} + \kappa w(\partial\mathcal{P}) \left( \sqrt{2} \max\limits_{\mathcal{C} \in \mathcal{P}} \sqrt{\iota_{\mathcal{G}}(\mathcal{C})} \|\overline{\mathbf{E}}_{\mathcal{P}}\|_F + 2 \frac{\|\mathbf{E}\|_{\partial\mathcal{P}^c}}{\min\limits_{\mathcal{C} \in \mathcal{P}} \sqrt{c_{\mathcal{G}}(\mathcal{C})}} \right) + (1-\kappa)\|\mathbf{E}\|_{\partial\mathcal{P}},$$

555 where the last line is an application of Lemma 2. Grouping the terms by the type of norm applied to
556 $\mathbf{E}$ finishes the proof. $\square$

557 **Theorem 1** (Oracle inequality)**.** *Assume that the RE assumption holds for the empirical multi-*
558 *task Gram matrix with constants* $\kappa \in \left[0, \frac{1}{2w(\partial\mathcal{P})} \min\limits_{\mathcal{C} \in \mathcal{P}} \sqrt{c_{\mathcal{G}}(\mathcal{C})}\right)$ *and* $\phi > 0$. *Suppose that*
559 $\max_{m \in \mathcal{V}} |\mathcal{T}_m(t)| \le bt$ *for some* $b > 0$. *Then, with a probability at least* $1 - \delta(t)$, *we have*

$$\left\|\mathbf{\Theta} - \hat{\mathbf{\Theta}}(t)\right\|_F \le 2\frac{\sigma}{\phi^2\sqrt{t}} f(\mathcal{G}, \mathbf{\Theta}) \sqrt{1 + 2b\sqrt{|\mathcal{V}| \log \frac{1}{\delta(t)}} + 2b \log \frac{1}{\delta(t)}},$$

560 *where*

$$f(\mathcal{G}, \mathbf{\Theta}) := \alpha_0 \left( a_2(\mathcal{G}, \mathbf{\Theta}) + \sqrt{2}\mathbb{1}_{\le 1}(\kappa)w(\partial\mathcal{P}) \right) \left( \frac{a_2(\mathcal{G}, \mathbf{\Theta}) + \sqrt{2}\mathbb{1}_{\le 1}(\kappa)w(\partial\mathcal{P})}{a_1(\mathcal{G}, \mathbf{\Theta}) \min\limits_{\mathcal{C} \in \mathcal{P}} \sqrt{c_{\mathcal{G}}(\mathcal{C})}} + 1 \right).$$

561 *Proof.* Using the previously established results, we obtain

$$\frac{1}{2t} \sum_{m \in \mathcal{V}} \|\mathbf{X}_m \boldsymbol{\epsilon}_m\|^2 + \alpha \|\mathbf{E}\|_{\partial\mathcal{P}^c}$$

$$\le \alpha_\delta a_2(\mathbf{\Theta}, \mathcal{G})\|\mathbf{E}_{\mathcal{P}}\|_F + \alpha_\delta(1-\kappa)^+ \|\mathbf{E}\|_{\partial\mathcal{P}} \quad \text{(by Proposition 4)}$$

$$= \alpha_\delta a_2(\mathbf{\Theta}, \mathcal{G})\|\mathbf{E}_{\mathcal{P}}\|_F + \alpha_\delta(1-\kappa)^+ \left\|\mathbf{B}_{\partial\mathcal{P}}\mathbf{B}_{\partial\mathcal{P}}^\dagger \mathbf{B}_{\partial\mathcal{P}}\mathbf{E}\right\|_{2,1} \quad \text{(by properties of the pseudo-inverse)}$$

$$\le \alpha_\delta a_2(\mathbf{\Theta}, \mathcal{G})\|\mathbf{E}_{\mathcal{P}}\|_F + \alpha_\delta \|\mathbf{B}_{\partial\mathcal{P}}\|_{2,1} \mathbb{1}_{\le 1}(\kappa)(1-\kappa)^+ \left\|\mathbf{B}_{\partial\mathcal{P}}^\dagger \mathbf{B}_{\partial\mathcal{P}}\mathbf{E}\right\| \quad \text{(by Proposition 1)}$$

$$\le \alpha_\delta(a_2(\mathbf{\Theta}, \mathcal{G}) + \mathbb{1}_{\le 1}(\kappa)\sqrt{2}w(\partial\mathcal{P}))\|\mathbf{E}\|_{\mathrm{RE}} \quad \text{(by definition of the } \|\cdot\|_{\mathrm{RE}} \text{ norm)}$$

$$\le \alpha \frac{a_2(\mathbf{\Theta}, \mathcal{G}) + \mathbb{1}_{\le 1}(\kappa)\sqrt{2}w(\partial\mathcal{P})}{\phi\sqrt{t}} \sqrt{\sum_{m \in \mathcal{V}} \|\boldsymbol{\epsilon}_m\|_{\mathbf{A}_m}^2} \quad \text{(using the RE assumption)}$$

$$\le \frac{\beta\alpha_\delta^2(a_2(\mathbf{\Theta}, \mathcal{G}) + \mathbb{1}_{\le 1}(\kappa)\|\mathbf{B}_{\partial\mathcal{P}}\|_{2,1})^2}{2\phi^2} + \frac{1}{2\beta t} \sum_{m \in \mathcal{V}} \|\mathbf{X}_m \boldsymbol{\epsilon}_m\|^2, \tag{29}$$

 where the last inequality holds for any $\beta > 0$, and is a consequence of the property that $uv \leq \dfrac{u^2 + v^2}{2}$
for any $u, v \in \mathbb{R}$.

As a result, we can bound the norm of $\mathbf{Q}_{\partial \mathcal{P}^c} \mathbf{E}$ as follows:

$$
\begin{aligned}
\left\| \mathbf{Q}_{\partial \mathcal{P}^c} \mathbf{E} \right\|_F &= \left\| \mathbf{B}_{\partial \mathcal{P}^c}^{\dagger} \mathbf{B}_{\partial \mathcal{P}^c} \mathbf{E} \right\|_F \\
&\leq \sqrt{\left\| \mathbf{L}_{\partial \mathcal{P}^c}^{\dagger} \right\|_{\infty, \infty}} \| \mathbf{E} \|_{\partial \mathcal{P}^c} \\
&\leq \frac{2\alpha_{\delta}(a_2(\boldsymbol{\Theta}, \mathcal{G}) + \mathbb{1}_{\leq 1}(\kappa) \| \mathbf{B}_{\partial \mathcal{P}} \|_{2,1})^2}{\phi^2 a_1(\boldsymbol{\Theta}, \mathcal{G}) \min_{\mathcal{C} \in \mathcal{P}} \sqrt{c_{\mathcal{G}}(\mathcal{C})}} \quad \text{(Equation (29) with } \beta = 1\text{).} \quad (30)
\end{aligned}
$$

We can also bound the norm of $\overline{\mathbf{E}}_{\mathcal{P}}$ as follows:

$$
\begin{aligned}
\left\| \overline{\mathbf{E}}_{\mathcal{P}} \right\|_F^2 &\leq \frac{1}{t\phi^2} \sum_{m \in \mathcal{V}} \| \mathbf{X}_m \boldsymbol{\epsilon}_m \|^2 \quad \text{(by RE assumption on empirical multi-task Gram matrix)} \\
&\leq \frac{4\alpha_{\delta}^2 (a_2(\boldsymbol{\Theta}, \mathcal{G}) + \mathbb{1}_{\leq 1}(\kappa) \| \mathbf{B}_{\partial \mathcal{P}} \|_{2,1})^2}{\phi^4} \quad \text{(by Equation (29) with } \beta = 2\text{).} \quad (31)
\end{aligned}
$$

The result is then obtained by combining Equations (30) and (31) along with using the fact that $\mathbf{E} = \overline{\mathbf{E}}_{\mathcal{P}} + \mathbf{Q}_{\partial \mathcal{P}^c} \mathbf{E}$ and the expressions of $a_1(\boldsymbol{\Theta}, \mathcal{G})$ and $a_2(\boldsymbol{\Theta}, \mathcal{G})$, and bounding $\alpha_{\delta}(t)$ as follows:

$$
\begin{aligned}
\frac{\alpha_{\delta}(t)^2}{\alpha_0^2} &= \frac{\sigma^2}{t^2} \left( \sum_{m \in \mathcal{V}} \| \mathbf{X}_m \|_F^2 + 2 \sqrt{\sum_{m \in \mathcal{V}} \| \mathbf{X}_m \mathbf{X}_m^{\top} \|_F^2 \log \frac{1}{\delta}} + 2 \max_{m \in \mathcal{V}} \| \mathbf{X}_m \|^2 \log \frac{1}{\delta} \right) \\
&\leq \frac{\sigma^2}{t^2} \left( t + 2 \sqrt{\sum_{m \in \mathcal{V}} |\mathcal{T}_m(t)|^2 \log \frac{1}{\delta}} + 2 \max_{m \in \mathcal{V}} |\mathcal{T}_m(t)| \log \frac{1}{\delta} \right) \\
&\leq \frac{\sigma^2}{t^2} \left( t + 2t \sqrt{\log \frac{1}{\delta}} + 2t \log \frac{1}{\delta} \right) \\
&\leq 2 \frac{\sigma^2}{t} \left( 1 + \sqrt{\log \frac{1}{\delta}} \right)^2
\end{aligned}
$$

$\square$

### B.3 Inheriting the RE condition from the true to the empirical data Gram matrix

#### B.3.1 From the adapted to the empirical multi-task Gram matrix

**Lemma 4** (Bounding a quadratic form using projections). *Let* $\mathbf{M}_1, \cdots, \mathbf{M}_p \in \mathbb{R}^{d \times d}$ *be symmetric matrices, and let* $\mathbf{J} := \frac{1}{p} \mathbf{1} \mathbf{1}^{\top}$, *and* $\mathbf{Q} = \mathbf{I} - \mathbf{J}$. *Then, for any* $\mathbf{Z} \in \mathbb{R}^{p \times d}$ *with rows* $\{ \mathbf{z}_i \}_{i=1}^p$, *we have:*

$$
\left| \sum_{i=1}^p \mathbf{z}_i^{\top} \mathbf{M}_i \mathbf{z}_i \right| \leq \frac{1}{p} \left\| \sum_{i=1}^p \mathbf{M}_i \right\| \| \mathbf{Z} \|_{\mathbf{J}}^2 + 2 \sqrt{\left\| \frac{1}{p} \sum_{i=1}^p \mathbf{M}_i^2 \right\|} \| \mathbf{Z} \|_{\mathbf{Q}} \| \mathbf{Z} \|_{\mathbf{J}} + \max_{1 \leq i \leq p} \| \mathbf{M}_i \| \| \mathbf{Z} \|_{\mathbf{Q}}^2
$$

*Proof.* We have

$$
\begin{aligned}
\left| \sum_{i=1}^p \mathbf{z}_i^{\top} \mathbf{M}_i \mathbf{z}_i \right| &= \left| \sum_{i=1}^p \bar{\mathbf{z}}^{\top} \mathbf{M}_i \bar{\mathbf{z}} + 2 \sum_{i=1}^p (\mathbf{z}_i - \bar{\mathbf{z}})^{\top} \mathbf{M}_i \bar{\mathbf{z}} + \sum_{i=1}^p (\mathbf{z}_i - \bar{\mathbf{z}})^{\top} \mathbf{M}_i (\mathbf{z}_i - \bar{\mathbf{z}}) \right| \\
&\leq \left| \bar{\mathbf{z}}^{\top} \sum_{i=1}^p \mathbf{M}_i \bar{\mathbf{z}} \right| + 2 \left| \sum_{i=1}^p \mathbf{e}_i^{\top} \mathbf{Q} \mathbf{Z} \mathbf{M}_i \bar{\mathbf{z}} \right| + \left| \sum_{i=1}^p \mathbf{e}_i^{\top} \mathbf{Q} \mathbf{Z} \mathbf{M}_i \mathbf{Z}^{\top} \mathbf{Q} \mathbf{e}_i \right| \quad (32)
\end{aligned}
$$

574    where we used the fact that $\mathbf{z}_i - \bar{\mathbf{z}} = \mathbf{Z}^\top \mathbf{e}_i - \mathbf{Z}^\top \mathbf{J} \mathbf{e}_i = \mathbf{Z}^\top \mathbf{Q} \mathbf{e}_i$.

575    Let us now examine every term on the right-hand side of Equation (32). For the first term, we have

$$\left| \bar{\mathbf{z}}^\top \sum_{i=1}^p \mathbf{M}_i \bar{\mathbf{z}} \right| \le \left\| \sum_{i=1}^p \mathbf{M}_i \right\| \|\bar{\mathbf{z}}\|^2 = \left\| \frac{1}{p} \sum_{i=1}^p \mathbf{M}_i \right\| \|\mathbf{Z}\|_{\mathbf{J}}^2. \tag{33}$$

576    For the second term, we have

$$
\begin{aligned}
\left| \sum_{i=1}^p \mathbf{e}_i^\top \mathbf{Q} \mathbf{Z} \mathbf{M}_i \bar{\mathbf{z}} \right| &\le \left\| \sum_{i=1}^p \mathbf{M}_i \mathbf{Z}^\top \mathbf{Q} \mathbf{e}_i \right\| \|\bar{\mathbf{z}}\| \\
&= \left\| \sum_{i=1}^p (\mathbf{e}_i^\top \otimes \mathbf{M}_i) \operatorname{vec}(\mathbf{Z}^\top \mathbf{Q}) \right\| \|\bar{\mathbf{z}}\| \\
&\le \left\| \sum_{i=1}^p (\mathbf{e}_i^\top \otimes \mathbf{M}_i) \right\| \left\| \operatorname{vec}(\mathbf{Z}^\top \mathbf{Q}) \right\| \|\bar{\mathbf{z}}\| \\
&= \left\| \sum_{i=1}^p (\mathbf{e}_i^\top \otimes \mathbf{M}_i) \right\| \|\mathbf{Q}\mathbf{Z}\|_F \|\bar{\mathbf{z}}\| \\
&= \sqrt{\left\| \left( \sum_{i=1}^p (\mathbf{e}_i^\top \otimes \mathbf{M}_i) \right)^\top \sum_{i=1}^p (\mathbf{e}_i^\top \otimes \mathbf{M}_i) \right\|} \|\mathbf{Q}\mathbf{Z}\|_F \|\bar{\mathbf{z}}\| \\
&= \sqrt{\left\| \sum_{i=1}^p \sum_{j=1}^p (\mathbf{e}_i^\top \otimes \mathbf{M}_i))(\mathbf{e}_j \otimes \mathbf{M}_j) \right\|} \|\mathbf{Q}\mathbf{Z}\|_F \|\bar{\mathbf{z}}\| \\
&= \sqrt{\left\| \sum_{i=1}^p \sum_{j=1}^p (\mathbf{e}_i^\top \mathbf{e}_j \otimes \mathbf{M}_i \mathbf{M}_j) \right\|} \|\mathbf{Q}\mathbf{Z}\|_F \|\bar{\mathbf{z}}\| \\
&= \sqrt{\left\| \sum_{i=1}^p \mathbf{M}_i^2 \right\|} \|\mathbf{Q}\mathbf{Z}\|_F \|\bar{\mathbf{z}}\|. \tag{34}
\end{aligned}
$$

577    Finally, for the last term, we have

$$
\begin{aligned}
\left| \sum_{i=1}^p \mathbf{e}_i^\top \mathbf{Q} \mathbf{Z} \mathbf{M}_i \mathbf{Z}^\top \mathbf{Q} \mathbf{e}_i \right| &\le \sum_{i=1}^p \|\mathbf{M}_i\| \left\| \mathbf{Z}^\top \mathbf{Q} \mathbf{e}_i \right\|^2 \\
&\le \max_{1 \le i \le p} \|\mathbf{M}_i\| \sum_{i=1}^p \left\| \mathbf{Z}^\top \mathbf{Q} \mathbf{e}_i \right\|^2 \\
&= \max_{1 \le i \le p} \|\mathbf{M}_i\| \|\mathbf{Q}\mathbf{Z}\|_F^2. \tag{35}
\end{aligned}
$$

578    Combining Equations (33) to (35) yields the result. $\qquad\square$

579    We also define an operator norm that is induced by the $\|\,\|_{\mathrm{RE}}$ introduced in Definition 2.

580    **Definition 3** ((RE,$\mathcal{S}$)-induced operator norm)**.** *Let $\{\mathbf{M}_m\}_{m \in \mathcal{V}} \subseteq \mathbb{R}^{d \times d}$ be symmetric matrices*
581    *associated to the graph nodes $\mathcal{V}$, and let $\mathbf{M}_\mathcal{V} := \operatorname{diag}\left( \mathbf{M}_1, \cdots, \mathbf{M}_{|\mathcal{V}|} \right) \in \mathbb{R}^{d|\mathcal{V}| \times d|\mathcal{V}|}$. For any*
582    *cluster $\mathcal{C} \in \mathcal{P}$, let the cluster mean and mean of squares associated to those matrices be given by*

$$\overline{\mathbf{M}}_\mathcal{C} := \frac{1}{|\mathcal{C}|} \sum_{m \in \mathcal{C}} \mathbf{M}_m, \qquad \overline{\mathbf{M}^2}_\mathcal{C} := \frac{1}{|\mathcal{C}|} \sum_{m \in \mathcal{C}} \mathbf{M}_m^2.$$

583    *The RE-induced operator norm of $\mathbf{M}_\mathcal{V}$ is defined as*

$$\|\mathbf{M}\|_{\mathrm{RE},\mathcal{S}} := \max_{\mathcal{C} \in \mathcal{P}} \left\| \overline{\mathbf{M}}_\mathcal{C} \right\| \vee \sqrt{\min_{\mathcal{C} \in \mathcal{P}} c_\mathcal{G}(\mathcal{C})^{-1} \max_{\mathcal{C} \in \mathcal{P}} \left\| \overline{\mathbf{M}^2}_\mathcal{C} \right\|} \vee \min_{\mathcal{C} \in \mathcal{P}} c_\mathcal{G}(\mathcal{C})^{-1} \max_{m \in \mathcal{V}} \|\mathbf{M}_m\|. \tag{36}$$

### B.3.2 Linking the adapted to the empirical Gram

We first start by establishing that given the closeness of two PSD matrices in a certain sense, the RE condition can be transferred between them.

**Proposition 5** (Restricted spectral norm). *Let $\mathbf{Z} \in \mathbb{R}^{|\mathcal{V}| \times d}$ verifying*

$$a_1(\mathcal{G}, \boldsymbol{\Theta}) \|\mathbf{Z}\|_{\partial \mathcal{P}^c} \leq a_2(\mathcal{G}, \boldsymbol{\Theta}) \|\overline{\mathbf{Z}}_{\mathcal{P}}\|_F + (1-\kappa)^+ \|\mathbf{Z}\|_{\partial \mathcal{P}}$$

*Let $\{\mathbf{M}_m\}_{m \in \mathcal{V}} \subseteq \mathbb{R}^{d \times d}$ be symmetric matrices associated to the graph nodes $\mathcal{V}$, and let $\mathbf{M}_{\mathcal{V}} :=$ $\mathrm{diag}(\mathbf{M}_1, \cdots, \mathbf{M}_{|\mathcal{V}|}) \in \mathbb{R}^{d|\mathcal{V}| \times d|\mathcal{V}|}$. Then we have:*

$$\left| \sum_{m \in \mathcal{V}} \mathbf{z}_m^{\top} \mathbf{M}_m \mathbf{z}_m \right| \leq \|\mathbf{M}\|_{\mathrm{RE},\mathcal{S}}^2 \left( 1 + \frac{a_2(\mathcal{G}, \boldsymbol{\Theta}) + (1-\kappa)^+ \|\mathbf{B}_{\partial \mathcal{P}}\|_{2,1}}{a_1(\mathcal{G}, \boldsymbol{\Theta})} \right)^2 \|\mathbf{Z}\|_{\mathrm{RE}}^2. \tag{37}$$

*Proof.* For any cluster $\mathcal{C}$, we denote by $\mathbf{B}_{\mathcal{C}}$ the incidence matrix obtained by setting the rows of $\mathbf{B}$ outside the edges linking nodes in $\mathcal{C}$ to null vectors. The latter's nullspace is the span of the vector $\mathbf{1}_{\mathcal{C}}$ having coordinates 1 at nodes in $\mathcal{C}$ and zeros elsewhere. Hence, the projector onto the orthogonal of $\mathbf{1}_{\mathcal{C}}$ is $\mathbf{Q}_{\mathcal{C}} := \mathbf{B}_{\mathcal{C}}^{\dagger} \mathbf{B}_{\mathcal{C}}$.

On the one hand, for any signal $\mathbf{Z} \in \mathbb{R}^{|\mathcal{V}| \times d}$ we have

$$\|\mathbf{Z}\|_{\partial \mathcal{P}^c} = \sum_{\mathcal{C} \in \mathcal{P}} \|\mathbf{B}_{\mathcal{C}} \mathbf{Z}\|_{2,1}$$

$$\geq \sum_{\mathcal{C} \in \mathcal{P}} \frac{\left\| \mathbf{B}_{\mathcal{C}}^{\dagger} \mathbf{B}_{\mathcal{C}} \mathbf{Z} \right\|_F}{\sqrt{\left\| \mathbf{L}_{\mathcal{C}}^{\dagger} \right\|_{\infty,\infty}}}$$

$$\geq \min_{\mathcal{C} \in \mathcal{P}} \sqrt{c_{\mathcal{G}}(\mathcal{C})} \sum_{\mathcal{C} \in \mathcal{P}} \|\mathbf{Z}\|_{\mathbf{Q}_{\mathcal{C}}}$$

Hence, by the proposition's assumptions, $\mathbf{Z}$ verifies

$$\min_{\mathcal{C} \in \mathcal{P}} \sqrt{c_{\mathcal{G}}(\mathcal{C})} a_1(\mathcal{G}, \boldsymbol{\Theta}) \sum_{\mathcal{C} \in \mathcal{P}} \|\mathbf{Z}\|_{\mathbf{Q}_{\mathcal{C}}} \leq (a_2(\mathcal{G}, \boldsymbol{\Theta}) \|\overline{\mathbf{Z}}_{\mathcal{P}}\|_F + (1-\kappa) \|\mathbf{Z}\|_{\partial \mathcal{P}})$$

$$\leq a_2(\mathcal{G}, \boldsymbol{\Theta}) \|\overline{\mathbf{Z}}_{\mathcal{P}}\|_F + (1-\kappa)^+ \|\mathbf{B}_{\partial \mathcal{P}}\|_{2,1} \left\| \mathbf{B}_{\partial \mathcal{P}}^{\dagger} \mathbf{B}_{\partial \mathcal{P}} \mathbf{Z} \right\|$$

$$\leq (a_2(\mathcal{G}, \boldsymbol{\Theta}) + (1-\kappa)^+ \|\mathbf{B}\|_{2,1}) \|\mathbf{Z}\|_{\mathrm{RE}}$$

From Lemma 4, we have

$$\left| \sum_{m \in \mathcal{V}} \mathbf{z}_m^{\top} \mathbf{M}_m \mathbf{z}_m \right|$$

$$\leq \sum_{\mathcal{C} \in \mathcal{P}} \left| \sum_{m \in \mathcal{C}} \mathbf{z}_m^{\top} \mathbf{M}_m \mathbf{z}_m \right|$$

$$\leq \sum_{\mathcal{C} \in \mathcal{P}} \|\overline{\mathbf{M}}_{\mathcal{C}}\| \|\mathbf{Z}\|_{\mathbf{J}_{\mathcal{C}}}^2 + 2 \sum_{\mathcal{C} \in \mathcal{P}} \sqrt{\left\| \overline{\mathbf{M}^2}_{\mathcal{C}} \right\|} \|\mathbf{Z}\|_{\mathbf{Q}_{\mathcal{C}}} \|\mathbf{Z}\|_{\mathbf{J}_{\mathcal{C}}} + \sum_{\mathcal{C} \in \mathcal{P}} \max_{m \in \mathcal{C}} \|\mathbf{M}_m\| \|\mathbf{Z}\|_{\mathbf{Q}_{\mathcal{C}}}^2, \tag{38}$$

where we used Equation (9).

This allows us to bound every term in Equation (38). For the second term on the right-hand side, we have

$$\sum_{\mathcal{C} \in \mathcal{P}} \sqrt{\left\| \overline{\mathbf{M}^2}_{\mathcal{C}} \right\|} \|\mathbf{Z}\|_{\mathbf{Q}_\mathcal{C}} \|\mathbf{Z}\|_{\mathbf{J}_\mathcal{C}}$$

$$\leq \max_{\mathcal{C} \in \mathcal{P}} \sqrt{\left\| \overline{\mathbf{M}^2}_{\mathcal{C}} \right\|} \|\overline{\mathbf{Z}}_{\mathcal{P}}\|_F \sqrt{\sum_{\mathcal{C} \in \mathcal{P}} \|\mathbf{Z}\|_{\mathbf{Q}_\mathcal{C}}^2}$$

$$\leq \frac{\min_{\mathcal{C} \in \mathcal{P}} c_{\mathcal{G}}(\mathcal{C})^{-\frac{1}{2}}}{a_1(\mathcal{G}, \boldsymbol{\Theta})} \max_{\mathcal{C} \in \mathcal{P}} \sqrt{\left\| \overline{\mathbf{M}^2}_{\mathcal{C}} \right\|} (a_2(\mathcal{G}, \boldsymbol{\Theta}) + (1 - \kappa)^+ \|\mathbf{B}\|_{2,1}) \|\mathbf{Z}\|_{\mathrm{RE}}^2 \qquad (39)$$

As for the third term, we have

$$\sum_{\mathcal{C} \in \mathcal{P}} \max_{m \in \mathcal{C}} \|\mathbf{M}_m\| \|\mathbf{Z}\|_{\mathbf{Q}_\mathcal{C}}^2 \leq \max_{m \in \mathcal{V}} \|\mathbf{M}_m\| \left( \sum_{\mathcal{C} \in \mathcal{P}} \|\mathbf{Z}\|_{\mathbf{Q}_\mathcal{C}} \right)^2$$

$$\leq \max_{m \in \mathcal{V}} \|\mathbf{M}_m\| \frac{\min_{\mathcal{C} \in \mathcal{P}} c_{\mathcal{G}}(\mathcal{C})^{-1}}{a_1(\mathcal{G}, \boldsymbol{\Theta})^2} (a_2(\mathcal{G}, \boldsymbol{\Theta}) + (1 - \kappa)^+ \|\mathbf{B}\|_{2,1})^2 \|\mathbf{Z}\|_{\mathrm{RE}}^2 \quad (40)$$

Consequently, denoting $v = \dfrac{a_2(\mathcal{G}, \boldsymbol{\Theta}) + (1 - \kappa)^+ \|\mathbf{B}\|_{2,1}}{a_1(\mathcal{G}, \boldsymbol{\Theta})}$, and combining Equations (38) to (40), we obtain

$$\left| \sum_{m \in \mathcal{V}} \mathbf{z}_m^\top \mathbf{M}_m \mathbf{z}_m \right|$$

$$\left( \max_{\mathcal{C} \in \mathcal{P}} \|\overline{\mathbf{M}}_\mathcal{C}\| + 2v \max_{\mathcal{C} \in \mathcal{P}} \sqrt{\left\| \overline{\mathbf{M}^2}_{\mathcal{C}} \right\|} + v^2 \max_{i \in \mathcal{V}} \|\mathbf{M}_i\| \right) \|\mathbf{Z}\|_{\mathrm{RE}}^2$$

$$\leq \left( \max_{\mathcal{C} \in \mathcal{P}} \|\overline{\mathbf{M}}_\mathcal{C}\| \right) \vee \sqrt{\min_{\mathcal{C} \in \mathcal{P}} c_{\mathcal{G}}(\mathcal{C})^{-1} \max_{\mathcal{C} \in \mathcal{P}} \left\| \overline{\mathbf{M}^2}_{\mathcal{C}} \right\|} \vee \min_{\mathcal{C} \in \mathcal{P}} c_{\mathcal{G}}(\mathcal{C})^{-1} \max_{i \in \mathcal{V}} \|\mathbf{M}_i\| \right) (1 + v)^2 \|\mathbf{Z}\|_{\mathrm{RE}}^2,$$

which finishes the proof. $\qquad \square$

**Proposition 6** (Inheritance of a RE condition from a close matrix). *Assume that the matrix $\mathbf{V}_\mathcal{V}$ verifies the RE condition with constant $\phi > 0$, and that $\left\| \dfrac{\mathbf{A}_\mathcal{V}}{t} - \mathbf{V}_\mathcal{V} \right\|_{\mathrm{op,RE}} \leq \gamma \phi^2$ for some $\gamma \in \left( 0, \left( 1 + \frac{a_2(\mathcal{G}, \boldsymbol{\Theta}) + (1 - \kappa)^+ \sqrt{2} w(\partial \mathcal{P})}{a_1(\mathcal{G}, \boldsymbol{\Theta})} \right)^{-2} \right)$. Then $\dfrac{\mathbf{A}_\mathcal{V}}{t}$ verifies the RE condition with constant*

$$\hat{\phi} = \phi \sqrt{1 - \gamma \left( 1 + \frac{a_2(\mathcal{G}, \boldsymbol{\Theta}) + (1 - \kappa)^+ \sqrt{2} w(\partial \mathcal{P})}{a_1(\mathcal{G}, \boldsymbol{\Theta})} \right)^2} \qquad (41)$$

*Proof.* From Proposition 4, we know that

$$\frac{1}{t} \epsilon_\mathcal{V}^\top \mathbf{A}_\mathcal{V} \epsilon_\mathcal{V} = \frac{1}{|\mathcal{V}|} \epsilon_\mathcal{V}^\top \mathbf{V}_\mathcal{V} \epsilon_\mathcal{V} + \epsilon_\mathcal{V}^\top \boldsymbol{\Delta}_\mathcal{V} \epsilon_\mathcal{V}$$

$$\geq \frac{1}{|\mathcal{V}|} \epsilon_\mathcal{V}^\top \mathbf{V}_\mathcal{V} \epsilon_\mathcal{V} - \left| \epsilon_\mathcal{V}^\top \boldsymbol{\Delta}_\mathcal{V} \epsilon_\mathcal{V} \right|$$

$$\geq \left( \phi^2 - \max_{m \in \mathcal{V}} \|\boldsymbol{\Delta}_\mathcal{V}\|_{\mathrm{op,RE}} \left( 1 + \frac{a_2(\mathcal{G}, \boldsymbol{\Theta}) + (1 - \kappa)^+ \|\mathbf{B}_{\partial \mathcal{P}}\|_{2,1}}{a_1(\mathcal{G}, \boldsymbol{\Theta})} \right)^2 \right) \|\mathbf{E}\|_{\mathrm{RE}}^2$$

$$\geq \left( \phi^2 - \gamma \phi^2 \left( 1 + \frac{a_2(\mathcal{G}, \boldsymbol{\Theta}) + (1 - \kappa)^+ \|\mathbf{B}_{\partial \mathcal{P}}\|_{2,1}}{a_1(\mathcal{G}, \boldsymbol{\Theta})} \right)^2 \right) \|\mathbf{E}\|_{\mathrm{RE}}^2$$

608 where the third inequality is an applicaiton of Proposition 5. $\qquad\square$

**Theorem 5** (Matrix Freedman Inequality, Tropp [2011]). *Consider a matrix martingale $\{\mathbf{M}(t)\}_{t\geq 1}$*
*with dimension $d_1 \times d_2$. Let $\{\mathbf{N}(t)\}_{t\geq 1}$ be the associated difference sequence. Assume that for some*
*$A > 0$, we have $\|\mathbf{N}(t)\| \leq A \quad \forall t \geq 1$ almost surely. Define for any $t \geq 1$:*

$$\mathbf{W}_{col}(t) := \sum_{\tau=1}^{t} \mathbb{E}\left[\mathbf{N}(\tau)\mathbf{N}(\tau)^{\top}|\mathcal{F}_{\tau-1}\right]$$

$$\mathbf{W}_{row}(t) := \sum_{\tau=1}^{t} \mathbb{E}\left[\mathbf{N}(\tau)^{\top}\mathbf{N}(\tau)|\mathcal{F}_{\tau-1}\right].$$

612 *Then, for any $u, v > 0$,*

$$\mathbb{P}\left[\exists t \geq 1; \|\mathbf{M}(t)\| \geq u \text{ and } \|\mathbf{W}_{col}\|(t) \vee \|\mathbf{W}_{row}(t)\| \leq v\right] \leq (d_1 + d_2)\exp\left(-\frac{3u^2}{6v + 2Au}\right)$$

**Corollary 1.** *Let $\{\mathbf{N}(\tau)\}_{\tau=1}^{t}$ by a sequence of matrices of dimension $d_1 \times d_2$, adapted to filtration*
*$\{\mathcal{F}_\tau\}_{\tau=1}^{t}$. Let $\{t_i\}_{i=1}^{N}$ an increasing sequence with elements in $[t]$ for some $N \leq t$. Consider the*
*sequence $\{\mathbf{M}(n)\}_{\tau=1}^{N}$ of random matrices defined by*

$$\mathbf{M}(n) = \sum_{i=1}^{n} \mathbf{N}(t_i) - \mathbb{E}\left[\mathbf{N}(t_i)|\mathcal{F}_{t_i-1}\right] \tag{42}$$

616 *Then $\{\mathbf{M}(n)\}_{n=1}^{N}$ is a martingale adapted to the filtration $\{\mathcal{F}_{t_n}\}_{n=1}^{N}$.*

617 *Moreover, if $\|\mathbf{N}(\tau)\| \leq b \quad \forall \tau \in [t]$ for some $b > 0$, then we have*

$$\mathbb{P}\left[\|\mathbf{M}(N)\| \geq u\right] \leq (d_1 + d_2)\exp\left(-\frac{3u^2}{6Nb^2 + 2\sqrt{2}bu}\right). \tag{43}$$

*Proof.* We denote $\mathbb{E}\left[\cdot|\mathcal{F}_s\right]$ as $\mathbb{E}_s\left[\cdot\right]$ for any $s \in \mathbb{N}$. Also, let $\mathbf{C}(s) := \mathbb{E}_{s-1}\left[\mathbf{N}(s)\right]$, which is
$\mathcal{F}_{s-1}$-measurable by construction. We have for any $n \in [N]$,

$$\mathbb{E}_{t_{n-1}}\left[\mathbf{C}(t_n)\right] = \mathbb{E}_{t_{n-1}}\left[\mathbb{E}_{t_n-1}\left[\mathbf{N}(t_n)\right]\right] = \mathbb{E}_{t_{n-1}}\left[\mathbf{N}(t_n)\right] \tag{44}$$
$$\implies \mathbb{E}_{t_{n-1}}\left[\mathbf{N}(t_n) - \mathbf{C}(t_n)\right] = 0 \tag{45}$$

620 where the first equality is due to the tower rule since $\mathcal{F}_{t_{n-1}} \subset \mathcal{F}_{t_n-1}$. Also, we have for any $\tau \geq 1$

$$\|\mathbf{N}(\tau) - \mathbf{C}(\tau)\|^2 = \left\|(\mathbf{N}(\tau) - \mathbf{C}(\tau))^2\right\| \tag{46}$$
$$\leq \text{Tr}\left((\mathbf{N}(\tau) - \mathbf{C}(\tau))^2\right) \tag{47}$$
$$= \text{Tr}\left((\mathbf{N}(\tau) - \mathbf{C}(\tau))^2\right) \tag{48}$$
$$= \|\mathbf{N}(\tau)\|_F^2 - 2\text{Tr}(\mathbf{C}(\tau)\mathbf{N}(\tau)) + \text{Tr}\left(\mathbf{C}(\tau)^2\right) \tag{49}$$
$$\leq \|\mathbf{N}(\tau)\|_F^2 + \text{Tr}\left(\mathbf{C}(\tau)^2\right) \leq 2b^2 \tag{50}$$

621 Hence $\mathbf{N}(\tau) - \mathbf{C}(\tau)$ is integrable for any $\tau \geq 1$. This shows that $\mathbf{M}(n)$ is a sequence of partial sums
622 of matrix martingale differences, hence it is a matrix martingale.

623 The second part of the corollary statement is a consequence of Theorem 5. The boundedness of
624 the sequence of martingale differences has already been established above. To verify the second
625 requirement of the theorem, let us compute bounds on the norms of $\mathbf{W}_{col}$ and $\mathbf{W}_{row}$ from Theorem 5.
626 Notice that the two matrices are equal since the difference sequence matrices $\mathbf{N}(t_s)$ are symmetric.

627 Hence, for any $n \in [N]$, we have

$$\|\mathbf{W}_{\text{col}}(N)\| \vee \|\mathbf{W}_{\text{row}}(N)\| \leq \text{Tr}(\mathbf{W}_{\text{col}}(N)) \vee \text{Tr}(\mathbf{W}_{\text{row}}(N)) \tag{51}$$

$$= \text{Tr}\left(\sum_{n=1}^{N} \mathbb{E}_{t_n-1}\left[(\mathbf{N}(t_n) - \mathbf{C}(t_n))^2\right]\right) \tag{52}$$

$$= \sum_{n=1}^{N} \mathbb{E}_{t_n-1}\left[\|\mathbf{N}(t_n)\|_F^2\right] - \mathbb{E}_{t_n-1}\left[2\,\text{Tr}(\mathbf{C}(t_n)\mathbf{N}(t_n))\right] + \text{Tr}\left(\mathbf{C}(t_n)^2\right) \tag{53}$$

$$= \sum_{n=1}^{N} \mathbb{E}_{t_n-1}\left[\|\mathbf{N}(t_n)\|_F^2\right] - \text{Tr}\left(\mathbf{C}(t_n)^2\right) \tag{54}$$

$$\leq \sum_{n=1}^{N} \mathbb{E}_{t_n-1}\left[\|\mathbf{N}(t_n)\|_F^2\right] \leq Nb^2. \tag{55}$$

628 By Theorem 5, we have for any $u > 0$

$$2d \exp\left(-\frac{3u^2}{6Nb^2 + 2\sqrt{2}bu}\right) \geq \mathbb{P}\left[\exists n \geq 1; \|\mathbf{M}(n)\| \geq u \text{ and } \|\mathbf{W}_{\text{col}}(n)\| \leq Nb^2\right] \tag{56}$$

$$\geq \mathbb{P}\left[\|\mathbf{M}(N)\| \geq u \text{ and } \|\mathbf{W}_{\text{col}}(N)\| \leq Nb^2\right] \tag{57}$$

$$= \mathbb{P}\left[\|\mathbf{M}(N)\| \geq u\right] \tag{58}$$

629 where the last line holds because we showed that the inequality $\|\mathbf{W}_{\text{col}}(N)\| \leq Nb^2$ holds almost
630 surely. □

631 **Proposition 7** (Concentration of the empirical multi-task Gram matrix around the adapted one)**.** *Let*
632 $t \geq 1$, $b > 0$. *Then we have:*

$$\mathbb{P}\left[\left\|\frac{\mathbf{A}_{\mathcal{V}}(t)}{t} - \mathbf{V}_{\mathcal{V}}\right\|_{\text{op,RE}} > \gamma \,\Big|\, \max_{m \in \mathcal{V}} |\mathcal{T}_m(t)| \leq bt\right] \leq d(2|\mathcal{P}|e^{-A_1 t} + (|\mathcal{V}| + |\mathcal{P}|)e^{-A_2 t} + 2|\mathcal{V}|e^{-A_3 t}),$$

633 *where*

$$A_1 := \frac{3\gamma^2 \min_{\mathcal{C} \in \mathcal{P}} |\mathcal{C}| t}{6b + 2\sqrt{2}\gamma}$$

$$A_2 := \frac{3\gamma^2 \min_{\mathcal{C} \in \mathcal{P}} c_{\mathcal{G}}(\mathcal{C})t}{6b + 2\sqrt{2}\gamma\sqrt{\dfrac{\min_{\mathcal{C} \in \mathcal{P}} c_{\mathcal{G}}(\mathcal{C})}{\min_{\mathcal{C} \in \mathcal{P}} |\mathcal{C}|}}}$$

$$A_3 := \frac{3\gamma^2 \min_{\mathcal{C} \in \mathcal{P}} c_{\mathcal{G}}(\mathcal{C})^2 t}{6b + 2\sqrt{2}\gamma \min_{\mathcal{C} \in \mathcal{P}} c_{\mathcal{G}}(\mathcal{C})}$$

634 *Proof.* For $\gamma > 0$, let us define

$$\mathbf{\Delta}_m := \frac{\mathbf{A}_{\mathcal{V}}}{t} - \mathbf{V}_{\mathcal{V}} \quad \text{and } G_{\text{Gram},\gamma} := \left\{\frac{1}{t}\|\mathbf{\Delta}_{\mathcal{V}}\|_{\text{RE},\mathcal{S}} \leq \gamma\right\},$$

635 where $\mathbf{\Delta}_{\mathcal{V}}$ is block diagonal matrix formed by $\{\mathbf{\Delta}_m\}_{m \in \mathcal{V}}$. We also define $\overline{\mathbf{\Delta}}_{\mathcal{C}}$ and $\overline{\mathbf{\Delta}^2}_{\mathcal{C}}$ in the same
636 pattern of Definition 3. We can express the complementary of this event as the disjunction of a finite

number of events as follows:

$$G^c_{\text{Gram},\gamma} \tag{59}$$

$$= \left\{ \max_{\mathcal{C}\in\mathcal{P}} \left\|\overline{\boldsymbol{\Delta}}_{\mathcal{C}}\right\| \vee \sqrt{\min_{\mathcal{C}\in\mathcal{P}} c_{\mathcal{G}}(\mathcal{C})^{-1} \max_{\mathcal{C}\in\mathcal{P}} \left\|\overline{\boldsymbol{\Delta}^2}_{\mathcal{C}}\right\|} \vee \min_{\mathcal{C}\in\mathcal{P}} c_{\mathcal{G}}(\mathcal{C})^{-1} \max_{m\in\mathcal{V}} \left\|\boldsymbol{\Delta}_m\right\| > t\gamma \right\} \tag{60}$$

$$= \bigcup_{\mathcal{C}\in\mathcal{P}} \left\{\left\|\overline{\boldsymbol{\Delta}}_{\mathcal{C}}\right\| > t\gamma\right\} \cup \bigcup_{\mathcal{C}\in\mathcal{P}} \left\{\left\|\overline{\boldsymbol{\Delta}^2}_{\mathcal{C}}\right\| > t^2\gamma^2 \min_{\mathcal{C}\in\mathcal{P}} c_{\mathcal{G}}(\mathcal{C})\right\} \cup \bigcup_{m\in\mathcal{V}} \left\{\left\|\boldsymbol{\Delta}_m\right\| > t\gamma \min_{\mathcal{C}\in\mathcal{P}} c_{\mathcal{G}}(\mathcal{C})\right\} \tag{61}$$

The first and third event can be bounded by considering the sequence $\mathbf{x}\mathbf{x}^\top(\tau)$ adapted to the filtration $\{\mathcal{F}_\tau\}$, verifying $\left\|\mathbf{x}\mathbf{x}^\top(\tau)\right\| \leq$.

**Bounding the probability of the first event**    Let $\mathcal{C} \in \mathcal{P}$ be a cluster. By definition, we have

$$|\mathcal{C}|\overline{\boldsymbol{\Delta}}_{\mathcal{C}}(t) = \sum_{m\in\mathcal{C}} \sum_{\tau\in\mathcal{T}_m(t)} \mathbf{x}\mathbf{x}(\tau) - \mathbb{E}\left[\mathbf{x}\mathbf{x}(\tau)|\mathcal{F}_{\tau-1}\right]$$

$$= \sum_{\tau\in\bigcup_{m\in\mathcal{C}}\mathcal{T}_m(t)} \mathbf{x}\mathbf{x}(\tau) - \mathbb{E}\left[\mathbf{x}\mathbf{x}(\tau)|\mathcal{F}_{\tau-1}\right]$$

We will apply Corollary 1 for the sequence of time indices in $\mathcal{C}$, *i.e.* $\bigcup_{m\in\mathcal{V}} \mathcal{T}_m(t)$. Hence $|\mathcal{C}|\overline{\boldsymbol{\Delta}}_{\mathcal{C}}$ is a martingale sequence, and we have

$$\mathbb{P}\left[\left\|\overline{\boldsymbol{\Delta}}_{\mathcal{C}}(t)\right\| > \gamma t \,\Big|\, \max_{m\in\mathcal{V}} |\mathcal{T}_m(t)| \leq bt\right] \leq 2d\exp\left(\frac{-3\gamma^2|\mathcal{C}|^2t^2}{6\sum_{m\in\mathcal{C}}|\mathcal{T}_m(t)| + 2\sqrt{2}\gamma|\mathcal{C}|t}\right)$$

$$\leq 2d\exp\left(\frac{-3\gamma^2|\mathcal{C}|^2t^2}{6|\mathcal{C}|bt + 2\sqrt{2}\gamma|\mathcal{C}|t}\right)$$

$$= 2d\exp\left(\frac{-3\gamma^2|\mathcal{C}|t}{6b + 2\sqrt{2}\gamma}\right)$$

$$\leq 2d\exp\left(\frac{-3\gamma^2 \min_{\mathcal{C}\in\mathcal{P}} |\mathcal{C}|t}{6b + 2\sqrt{2}\gamma}\right) \tag{62}$$

**Bounding the probability of the third event**    Let $m \in \mathcal{V}$ be a task index. We apply Corollary 1 for the sequence of time steps in $\mathcal{T}_m(t)$. We have

$$\boldsymbol{\Delta}_m(t) = \sum_{\tau\in\mathcal{T}_m(t)} \mathbf{x}\mathbf{x}(\tau) - \mathbb{E}\left[\mathbf{x}\mathbf{x}(\tau)|\mathcal{F}_{\tau-1}\right]$$

is a martingale sequence, hence

$$\mathbb{P}\left[\left\|\boldsymbol{\Delta}_m(t)\right\| > \gamma \min_{\mathcal{C}\in\mathcal{P}} c_{\mathcal{G}}(\mathcal{C})t \,\Big|\, \max_{m\in\mathcal{V}} |\mathcal{T}_m(t)| \leq bt\right] \leq 2d\exp\left(\frac{-3\gamma^2 \min_{\mathcal{C}\in\mathcal{P}} c_{\mathcal{G}}(\mathcal{C})^2t^2}{6|\mathcal{T}_m(t)| + 2\sqrt{2}\gamma \min_{\mathcal{C}\in\mathcal{P}} c_{\mathcal{G}}(\mathcal{C})t}\right)$$

$$\leq 2d\exp\left(\frac{-3\gamma^2 \min_{\mathcal{C}\in\mathcal{P}} c_{\mathcal{G}}(\mathcal{C})^2t^2}{6bt + 2\sqrt{2}\gamma \min_{\mathcal{C}\in\mathcal{P}} c_{\mathcal{G}}(\mathcal{C})t}\right)$$

$$= 2d\exp\left(\frac{-3\gamma^2 \min_{\mathcal{C}\in\mathcal{P}} c_{\mathcal{G}}(\mathcal{C})^2t}{6b + 2\sqrt{2}\gamma \min_{\mathcal{C}\in\mathcal{P}} c_{\mathcal{G}}(\mathcal{C})}\right). \tag{63}$$

**Bounding the probability of the second event** Let $\mathcal{C} \in \mathcal{P}$ be a cluster, and let us denote $\mathbf{e}_m$ the $m^{\text{th}}$ canonical vector of $\mathbb{R}^{|\mathcal{C}|}$. We have

$$
\left\| \overline{\boldsymbol{\Delta}^2}_{\mathcal{C}}(t) \right\| = \frac{1}{|\mathcal{C}|} \left\| \sum_{m \in \mathcal{C}} \left( \sum_{\tau \in \mathcal{T}_m(t)} \mathbf{xx}(\tau) - \mathbb{E}\left[ \mathbf{xx}(\tau) | \mathcal{F}_{\tau-1} \right] \right)^2 \right\|
$$

$$
= \frac{1}{|\mathcal{C}|} \left\| \sum_{m \in \mathcal{C}} \mathbf{e}_m^\top \otimes \left( \sum_{\tau \in \mathcal{T}_m(t)} \mathbf{xx}(\tau) - \mathbb{E}\left[ \mathbf{xx}(\tau) | \mathcal{F}_{\tau-1} \right] \right) \right\|^2
$$

$$
= \frac{1}{|\mathcal{C}|} \left\| \sum_{\tau \in \bigcup_{m \in \mathcal{C}} \mathcal{T}_m(t)} \mathbf{e}_{m(\tau)}^\top \otimes \left( \mathbf{xx}(\tau) - \mathbb{E}\left[ \mathbf{xx}(\tau) | \mathcal{F}_{\tau-1} \right] \right) \right\|^2
$$

$$
= \frac{1}{|\mathcal{C}|} \left\| \sum_{\tau \in \bigcup_{m \in \mathcal{C}} \mathcal{T}_m(t)} \mathbf{e}_{m(\tau)}^\top \otimes \mathbf{xx}(\tau) - \mathbb{E}\left[ \mathbf{e}_{m(\tau)} \otimes \mathbf{xx}(\tau) | \mathcal{F}_{\tau-1} \right] \right\|^2 ,
$$

where the last equality holds since $m(\tau)$ is measurable w.r.t. $\mathcal{F}_{\tau-1}$. We will apply the Corollary 1 to the set of time steps $\bigcup_{m \in \mathcal{C}} \mathcal{T}_m(t)$ and the adapted sequence $\mathbf{e}_{m(\tau)}^\top \otimes \mathbf{xx}(\tau)$ of matrices in $\mathbb{R}^{d \times d|\mathcal{C}|}$. Hence we have

$$
\mathbb{P}\left[ \sqrt{\left\| \overline{\boldsymbol{\Delta}^2}_{\mathcal{C}}(t) \right\|} > \gamma t \min_{\mathcal{C} \in \mathcal{P}} \sqrt{c_{\mathcal{G}}(\mathcal{C})} \Big| \max_{m \in \mathcal{V}} |\mathcal{T}_m(t)| \le bt \right]
$$

$$
\le d(1 + |\mathcal{C}|) \exp\left( \frac{-3\gamma^2 |\mathcal{C}| \min_{\mathcal{C} \in \mathcal{P}} c_{\mathcal{G}}(\mathcal{C}) t^2}{6 \sum_{m \in \mathcal{C}} |\mathcal{T}_m(t)| + 2\sqrt{2}\gamma \sqrt{|\mathcal{C}| \min_{\mathcal{C} \in \mathcal{P}} c_{\mathcal{G}}(\mathcal{C})} t} \right)
$$

$$
\le d(1 + |\mathcal{C}|) \exp\left( \frac{-3\gamma^2 |\mathcal{C}| \min_{\mathcal{C} \in \mathcal{P}} c_{\mathcal{G}}(\mathcal{C}) t}{6|\mathcal{C}|b + 2\sqrt{2}\gamma \sqrt{|\mathcal{C}| \min_{\mathcal{C} \in \mathcal{P}} c_{\mathcal{G}}(\mathcal{C})}} \right)
$$

$$
= d(1 + |\mathcal{C}|) \exp\left( \frac{-3\gamma^2 \min_{\mathcal{C} \in \mathcal{P}} c_{\mathcal{G}}(\mathcal{C}) t}{6b + 2\sqrt{2}\gamma \sqrt{\frac{\min_{\mathcal{C} \in \mathcal{P}} c_{\mathcal{G}}(\mathcal{C})}{|\mathcal{C}|}}} \right)
$$

$$
\le d(1 + |\mathcal{C}|) \exp\left( \frac{-3\gamma^2 \min_{\mathcal{C} \in \mathcal{P}} c_{\mathcal{G}}(\mathcal{C}) t}{6b + 2\sqrt{2}\gamma \sqrt{\frac{\min_{\mathcal{C} \in \mathcal{P}} c_{\mathcal{G}}(\mathcal{C})}{\min_{\mathcal{C} \in \mathcal{P}} |\mathcal{C}|}}} \right) \tag{64}
$$

**Union bound** We conclude the result of the statement via a union bound using Equation (61). $\quad\square$

**Proposition 8** (Concentration of the empirical multi-task Gram matrix around the adapted one, simplified). *propEmpCovConcentrationSimplified Let $t \ge 1$, $b > 0$. Assume that $\max_{m \in \mathcal{V}} |\mathcal{T}_m(t)| \le bt$. Then we have:*

$$
\mathbb{P}\left[ \left\| \frac{\mathbf{A}_{\mathcal{V}}}{t} - \mathbf{V}_{\mathcal{V}} \right\|_{\text{op,RE}} > \gamma \right] \le 6d|\mathcal{V}| \exp\left( \frac{-3\gamma^2 (\min_{\mathcal{C} \in \mathcal{P}} (\tilde{c}_{\mathcal{G}}(\mathcal{C}) \wedge \tilde{c}_{\mathcal{G}}(\mathcal{C})^2) t}{6b + 2\sqrt{2}\gamma} \right),
$$

*where $\tilde{c}_{\mathcal{G}}(\mathcal{C}) := c_{\mathcal{G}}(\mathcal{C}) \wedge |\mathcal{C}| \quad \forall \mathcal{C} \in \mathcal{P}$.*

*Proof.* The proof will rely on simple calculus inequalities. Hence, let $u = \min_{\mathcal{C} \in \mathcal{P}} c_{\mathcal{G}}(\mathcal{C}), v = \min_{\mathcal{C} \in \mathcal{P}} |\mathcal{C}|, f = 3\gamma^2, g = 6b, h = 2\sqrt{2}\gamma$, which are all positive. Then, we have

$$A_1 = \frac{fu}{f+g} \geq \frac{(u \wedge v)f}{f+g} \geq (u \wedge v)\frac{(1 \wedge u \wedge v)f}{f+g(1 \wedge u \wedge v)}$$

$$A_2 = \frac{fv}{f+g\frac{v}{u}} \geq \frac{(v \wedge u)f}{f+g\frac{v\wedge u}{u}} \geq \frac{(v \wedge u)f}{f+g} \geq (u \wedge v)\frac{(1 \wedge u \wedge v)f}{f+(1 \wedge u \wedge v)g}$$

$$A_3 = \frac{fv^2}{f+gv} \geq \frac{(v \wedge u)^2}{f+(v \wedge u)g} \geq (u \wedge v)\frac{(1 \wedge u \wedge v)f}{f+(1 \wedge u \wedge v)g}$$

where we used the fact that functions of the form $x \mapsto \frac{x}{\beta_1 x + \beta_2}$ for positive $\beta_1, \beta_2$ are increasing on $\mathbb{R}_+$.

As a final step, we use the inequality $\dfrac{(1 \wedge x)f}{f + (1 \wedge x)g} \geq \dfrac{x \wedge 1}{f+g}$ taken for $x = u \wedge v$, we apply the $\exp(- \cdot t)$ function and we use the result of Proposition 7, we deduce the result. $\square$

### B.3.3 From the true to the adapted Gram matrix

For all of the proofs in this subsection, we follow an approach similar to that of Oh et al. [2021]. In particular, we use their Lemma 10.

**Theorem 6** (Lemma 10 of Oh et al. [2021]). *Under Assumption 2 on the context generating distribution, let $t \geq 1$. We have for any $\boldsymbol{\theta} \in \mathbb{R}^d$:*

$$\sum_{\mathbf{x} \in \mathcal{A}(t)} \mathbb{E}\left[\mathbf{x}\mathbf{x}^\top \mathbb{1}\left\{\mathbf{x} \in \arg\max_{\tilde{\mathbf{x}} \in \mathcal{A}(t)} \langle \boldsymbol{\theta}, \tilde{\mathbf{x}}\rangle\right\}\right] \succcurlyeq \frac{1}{2\nu\omega}\overline{\boldsymbol{\Sigma}} \tag{65}$$

**Proposition 9** (RE condition from the true to the adapted Gram matrix). *Under Assumption 2, for any $t \geq 1$, the adapted Gram matrix $\mathbf{V}_{\mathcal{V}}(t)$ verifies the compatibility condition with constants $\kappa$ and $\dfrac{\phi}{\sqrt{2\nu\omega}}$.*

*Proof.* For $t \geq 1$, we have

$$\mathbb{E}\left[\mathbf{x}(t)\mathbf{x}(t)^\top | \mathcal{F}_{t-1}\right] = \mathbb{E}\left[\sum_{\mathbf{x} \in \mathcal{A}(t)} \mathbf{x}(t)\mathbf{x}(t)^\top | \mathcal{F}_{t-1}\right] \tag{66}$$

Let $m \in \mathcal{V}$. We have

$$\begin{aligned}
\mathbf{V}_m(t) &= \frac{1}{t}\sum_{\tau \in \mathcal{T}_m(t)} \mathbb{E}\left[\mathbf{x}(\tau)\mathbf{x}(\tau)^\top | \mathcal{F}_{\tau-1}\right] \\
&= \frac{1}{t}\sum_{\tau \in \mathcal{T}_m(t)} \mathbb{E}\left[\mathbb{E}\left[\mathbf{x}(\tau)\mathbf{x}(\tau)^\top | \boldsymbol{\theta}_m(\tau-1), \mathcal{F}_{\tau-1}\right] | \mathcal{F}_{\tau-1}\right] \quad \text{(law of total expectation)} \\
&= \frac{1}{t}\sum_{\tau \in \mathcal{T}_m(t)} \mathbb{E}\left[\mathbf{x}(\tau)\mathbf{x}(\tau)^\top | \boldsymbol{\theta}_m(\tau-1)\right] \quad (\mathbf{x}(\tau) \text{ is fully determined by } \boldsymbol{\theta}_m(\tau-1)) \\
&= \frac{1}{t}\sum_{\tau \in \mathcal{T}_m(t)} \mathbb{E}\left[\sum_{\mathbf{x} \in \mathcal{A}(\tau)} \mathbf{x}\mathbf{x}^\top \mathbb{1}\left\{\mathbf{x} \in \arg\max_{\tilde{\mathbf{x}} \in \mathcal{A}(t)} \langle \boldsymbol{\theta}, \tilde{\mathbf{x}}\rangle\right\} | \boldsymbol{\theta}_m(\tau-1)\right] \\
&\succcurlyeq \frac{1}{2\nu\omega}\overline{\boldsymbol{\Sigma}} \quad \text{(by Theorem 6).}
\end{aligned} \tag{67}$$

Now, let $\mathbf{Z} \in \mathcal{S}$, where $\mathcal{S}$ is defined with constant $\kappa$ of Assumption 4. Then

$$\begin{aligned}
\sum_{m \in \mathcal{V}} \|\mathbf{z}\|_{\mathbf{V}_m(t)} &\geq \frac{1}{2\nu\omega}\sum_{m \in \mathcal{V}} \|\mathbf{z}_m\|_{\overline{\boldsymbol{\Sigma}}} \quad \text{by Equation (67)} \\
&\geq \frac{\phi^2}{2\nu\omega}\|\mathbf{Z}\|_{\mathrm{RE}}^2 \quad \text{(by Assumption 4),}
\end{aligned}$$

which finishes the proof. $\square$

674 **Theorem 2** (RE condition holding for the empirical multi-task Gram matrix). *Under assumptions 2*
675 *and 4, let $t \geq 1$, and let $\kappa, \phi$ be the constants from Assumption 4. Assume that $\max_{m \in \mathcal{V}} |\mathcal{T}_m(t)| \leq bt$.*
676 *Then, for any $\gamma \in \left( 0, \left( 1 + \frac{a_2(\mathcal{G}, \boldsymbol{\Theta}) + (1-\kappa)^+ \sqrt{2} w(\partial \mathcal{P})}{a_1(\mathcal{G}, \boldsymbol{\Theta})} \right)^{-2} \right)$, the empirical multi-task Gram matrix*
677 *verifies the RE condition with constants $\kappa$ and $\hat{\phi}$, with*

$$\hat{\phi} = \tilde{\phi} \sqrt{1 - \gamma \left( 1 + \frac{a_2(\mathcal{G}, \boldsymbol{\Theta}) + (1-\kappa)^+ \sqrt{2} w(\partial \mathcal{P})}{a_1(\mathcal{G}, \boldsymbol{\Theta})} \right)^2}, \tag{6}$$

678 *with a probability at least equal to $1 - 6d|\mathcal{V}| \exp\left( \dfrac{-3\gamma^2 \tilde{\phi}^4 (\min_{\mathcal{C} \in \mathcal{P}} (\tilde{c}_{\mathcal{G}}(\mathcal{C}) \wedge \tilde{c}_{\mathcal{G}}(\mathcal{C})^2) t}{6b + 2\sqrt{2} \gamma \tilde{\phi}^2} \right)$, where*

679 $\tilde{\phi} := \dfrac{\phi}{\sqrt{2\nu\omega}}$ *and* $\tilde{c}_{\mathcal{G}}(\mathcal{C}) := c_{\mathcal{G}}(\mathcal{C}) \wedge |\mathcal{C}| \quad \forall \mathcal{C} \in \mathcal{P}.$

680 *Proof.* For the sake of readability, let $\tilde{\phi} = \frac{\phi}{\sqrt{2\nu\omega}}$ the compatibility constant of the adapted Gram
681 matrix, according to Proposition 9. Then:

$$1 - 6d|\mathcal{V}| \exp\left( \frac{-3\gamma^2 \tilde{\phi}^4 (\min_{\mathcal{C} \in \mathcal{P}} (\tilde{c}_{\mathcal{G}}(\mathcal{C}) \wedge \tilde{c}_{\mathcal{G}}(\mathcal{C})^2) t}{6b + 2\sqrt{2} \gamma \tilde{\phi}^2} \right) \tag{68}$$

$$\leq \mathbb{P}\left[ \left\| \frac{\mathbf{A}_{\mathcal{V}}}{t} - \mathbf{V}_{\mathcal{V}} \right\|_{\text{op,RE}} \leq \gamma \tilde{\phi}^2 \right] \quad \text{(by Proposition 8)} \tag{69}$$

$$\leq \mathbb{P}\left[ \frac{\mathbf{A}_{\mathcal{V}}}{t} \text{ satisfies the RE condition with constant } \kappa \text{ and } \hat{\phi} \right] \quad \text{(by Proposition 6),} \tag{70}$$

682 where $\hat{\phi} = \tilde{\phi} \sqrt{1 - \gamma \left( 1 + \frac{a_2(\mathcal{G}, \boldsymbol{\Theta}) + (1-\kappa)^+ \sqrt{2} w(\partial \mathcal{P})}{a_1(\mathcal{G}, \boldsymbol{\Theta})} \right)^2}.$ $\qquad \square$

### 683 B.4 Regret bound

684 **Lemma 5** (Concentration of the fraction of observations per task). *lemma Assume that $|\mathcal{V}| \geq 2$. Then*
685 *for $\delta \in (0, 1)$, we have with a probability at least $1 - \delta$:*

$$\max_{m \in \mathcal{V}} \frac{|\mathcal{T}_m(t)|}{t} \leq \frac{1}{|\mathcal{V}|} + 2\sqrt{\frac{1}{t|\mathcal{V}|} \log \frac{|\mathcal{V}|}{\delta}} + \frac{4}{3t} \log \frac{|\mathcal{V}|}{\delta}. \tag{71}$$

*Proof.* We have $|\mathcal{T}_m(t)| := \sum_{\tau=1}^{t} [m(\tau) = m]$, where $\forall t, \forall m \in \mathcal{V}, \mathbb{P}[m(t) = m] = \frac{1}{|\mathcal{V}|}$, meaning that the binary variable $[m(t) = m]$ follows a Bernoulli distribution $\mathcal{B}(\frac{1}{\mathcal{V}})$. Then, the random variable $X_t := [m(t) = m] - \frac{1}{|\mathcal{V}|}$ has mean 0, variance $\frac{1}{|\mathcal{V}|}(1 - \frac{1}{|\mathcal{V}|})$, and verifies $|X_t| \leq 1 - \frac{1}{|\mathcal{V}|}$ since $|\mathcal{V}| \geq 2$. As a result, via the Bernstein inequality, we have for any $m \in \mathcal{V}$, and for any $w \geq 0$,

$$\mathbb{P}\left[ \frac{|\mathcal{T}_m(t)|}{t} \geq \frac{1}{|\mathcal{V}|} + w \right] \leq \exp\left( -\frac{tw^2}{2(1 - \frac{1}{|\mathcal{V}|})(\frac{1}{|\mathcal{V}|} + \frac{w}{3})} \right) \leq \exp\left( -\frac{tw^2}{2(\frac{1}{|\mathcal{V}|} + \frac{w}{3})} \right)$$

686 For the right-hand side to hold with a probability at most $\delta \in (0, 1)$, it is sufficient to have

$$t \frac{w^2}{2(\frac{1}{|\mathcal{V}|} + \frac{w}{3})} \geq \log \frac{1}{\delta}$$

$$\Longleftarrow \frac{w^2}{2} \geq \frac{2 \frac{1}{|\mathcal{V}|} \log \frac{1}{\delta}}{t} \text{ and } \frac{w^2}{2} \geq \frac{2w \log \frac{1}{\delta}}{3t}$$

$$\Longleftarrow w = 2\sqrt{\frac{\frac{1}{|\mathcal{V}|} \log \frac{1}{\delta}}{t}} + \frac{4 \log \frac{1}{\delta}}{3t}$$

Hence, and via a union bound, we get

$$\mathbb{P}\left[\frac{|\mathcal{T}_m(t)|}{t} \geq \frac{1}{|\mathcal{V}|} + 2\sqrt{\frac{1}{|\mathcal{V}|}\log\frac{1}{\delta}} + \frac{4}{3t}\log\frac{1}{\delta}\right] \leq \delta$$

$$\Longrightarrow \mathbb{P}\left[\max_{m\in\mathcal{V}}\frac{|\mathcal{T}_m(t)|}{t} \geq \frac{1}{|\mathcal{V}|} + 2\sqrt{\frac{\frac{1}{|\mathcal{V}|}\log\frac{1}{\delta}}{t}} + \frac{4\log\frac{1}{\delta}}{3t}\right] \leq |\mathcal{V}|\delta$$

The result is obtained by adjusting the value of $\delta$. $\qquad\square$

**Theorem 3** (Regret bound). *Let the mean horizon per node be $\overline{T} = \frac{T}{|\mathcal{V}|}$. Let $\min\limits_{\mathcal{C}\in\mathcal{P}}\sqrt{c_{\mathcal{G}}(\mathcal{C})}$ going asymptotically to infinity and $\max_{\mathcal{C}\in\mathcal{P}}\sqrt{\iota_{\mathcal{G}}(\mathcal{C})}$ going asymptotically to zero as well as $\max_{\mathcal{C}\in\mathcal{P}}\sqrt{\iota_{\mathcal{G}}(\mathcal{C})}w(\partial\mathcal{P})$ and $\frac{w(\partial\mathcal{P})}{\min\limits_{\mathcal{C}\in\mathcal{P}}\sqrt{c_{\mathcal{G}}(\mathcal{C})}}$ going asymptotically to zero. Under assumptions1 to 4 and $\kappa < 1$, the expected regret of the Network Lasso Bandit algorithm is upper bounded as follows:*

$$\mathcal{R}(|\mathcal{V}|\overline{T}) = \mathcal{O}\left(\sqrt{\frac{\overline{T}}{\min\limits_{\mathcal{C}\in\mathcal{P}}c_{\mathcal{G}}(\mathcal{C})}}\left(\sqrt{|\mathcal{V}|} + \sqrt{\log(\overline{T}|\mathcal{V}|)} + \sqrt[4]{|\mathcal{V}\log(\overline{T}|\mathcal{V}|)|}\right) + \frac{1}{A}\log(d|\mathcal{V}|)\right),$$

*with $A = \dfrac{3\gamma^2 \min_{\mathcal{C}\in\mathcal{P}}(\tilde{c}_{\mathcal{G}}(\mathcal{C}) \wedge \tilde{c}_{\mathcal{G}}^2(\mathcal{C}))}{6\frac{\log(|\mathcal{V}|)}{\sqrt{|\mathcal{V}|}} + 2\sqrt{2}\gamma}$.*

*Proof.* For any time step $t$, we will define a list of good events under which the Oracle inequality and the RE condition for the empirical multi-task Gram matrix both hold with high probability. Then, we will use those bounds to sum up over time steps until horizon $T$.

**Good events** We formalize these requirements as three families of time-depending "good" events.

- $G_{\text{pro}}(t)$ is the event that the mean of the empirical process bounded by $\alpha(t)$ up to a constant $c$, which is equivalent to saying that it converges:

$$G_{\text{pro}}(t) := \left\{\frac{1}{t}\|\mathbf{K}\|_F \leq \frac{\alpha(t)}{\alpha_0}\right\} \tag{72}$$

- $G_{\text{sel}}(t)$ is the event that the number of selections of all tasks is bounded by its expected value up to a small constant $\rho(t)$

$$G_{\text{sel}}(t) := \left\{\max_{m\in\mathcal{V}}\frac{|\mathcal{T}_m(t)|}{t} \leq \frac{1}{|\mathcal{V}|} + \frac{\rho(t)}{t}\right\} \tag{73}$$

- $G_{\text{RE}}(t)$ is the event that the empirical multi-task Gram matrix $\frac{1}{t}\mathbf{A}_{\mathcal{V}}(t)$ satisfies the RE condition.

$$G_{\text{RE}}(t) := \left\{\frac{1}{t}\mathbf{A}_{\mathcal{V}}(t) \text{ verifies the RE condition with constants } \kappa, \hat{\phi}\right\} \tag{74}$$

Event $G_{\text{pro}}(t)$ is the most straightforward to cover since our bound on the empirical process given in Lemma 3 holds with a probability of at least $1 - \delta(t)$, thus:

$$\mathbb{P}\left[G_{\text{pro}}(t)^c | G_{\text{sel}}(t)\right] \leq \delta(t), \tag{75}$$

where we included the time dependency on $\delta(t)$ in contrast to the previous section. This way we emphasize to adjust $\delta(t)$ after each round, to guarantee a sub linear regret bound. The probability of event $G_{\text{sel}}(t)$ can be determined using Bernstein's inequality:

From Lemma 5 we can select $\rho(t) = 2\sqrt{\frac{t}{|\mathcal{V}|}\log\frac{|\mathcal{V}|}{\delta_{\text{sel}}(t)}} + \frac{4}{3}\log\frac{|\mathcal{V}|}{\delta_{\text{sel}}(t)}$ as well as $\mathbb{P}\left[G_{\text{sel}}(t)^c\right] \leq \delta_{\text{sel}}(t)$.

### B.4.1 Instantaneous regret decomposition

Now, given the event probabilities, we condition the instantaneous regret $r(t)$ on the good events at a time $t > t_0$. We have for its expectation:

$$
\begin{aligned}
\mathbb{E}\left[r(t)\right] &\leq \mathbb{E}\left[r(t)|G_{\mathrm{sel}}(t)\right] + 2\mathbb{P}\left[G_{\mathrm{sel}}(t)^c\right] \\
&\leq \mathbb{E}\left[r(t)|G_{\mathrm{pro}}(t) \cap G_{\mathrm{RE}}(t) \cap G_{\mathrm{sel}}(t)\right] \\
&\quad + 2\left(\mathbb{P}\left[G_{\mathrm{pro}}(t)^c|G_{\mathrm{sel}}(t)\right] + \mathbb{P}\left[G_{\mathrm{RE}}(t)^c|G_{\mathrm{sel}}(t)\right] + \mathbb{P}\left[G_{\mathrm{sel}}(t)^c\right]\right),
\end{aligned} \tag{76}
$$

where we used the worst case bound $r(t) \leq 2$ if any one of the good events does not hold.

**Bounding the regret**   Inserting our results of the event probabilities, the oracle inequality and the decomposition of the expected instantaneous regret in Equation (76) and bounding the sum over rounds, yields the final result. Thus, we start by bounding the sum over the first term i.e. the expected regret in case all good events hold:

$$
\sum_{t=1}^{T} \mathbb{E}\left[r(t)|G_{\mathrm{pro}}(t) \cap G_{\mathrm{RE}}(t) \cap G_{\mathrm{sel}}(t)\right] \leq \sum_{t=1}^{T} \left\|\boldsymbol{\Theta} - \hat{\boldsymbol{\Theta}}(t)\right\|_F
$$

Taking the result of our oracle inequality in Theorem 1, we point out that only $\alpha(t)$ is time dependent such that the rest of the terms can be pulled outside the sum:

$$
\begin{aligned}
\sum_{t=1}^{T} \left\|\boldsymbol{\Theta} - \hat{\boldsymbol{\Theta}}(t)\right\|_F &\leq \sum_{t=1}^{T} 2\frac{\sigma}{\hat{\phi}^2\sqrt{t}} f(\mathcal{G}, \boldsymbol{\Theta}) \sqrt{1 + 2b\sqrt{|\mathcal{V}|\log\frac{1}{\delta(t)}} + 2b\log\frac{1}{\delta(t)}} \\
&= \frac{2\sigma}{\hat{\phi}^2} f(\mathcal{G}, \boldsymbol{\Theta}) \sum_{t=1}^{T} \sqrt{\frac{1}{t} + \frac{2b}{t}\sqrt{2|\mathcal{V}|\log(t)} + \frac{4b}{t}\log(t)} \\
&\leq \frac{2\sigma}{\hat{\phi}^2} f(\mathcal{G}, \boldsymbol{\Theta}) \int_0^T \frac{1}{\sqrt{t}} + \sqrt{\frac{2b}{t}\left(\sqrt{2|\mathcal{V}|\log(T)} + 2\log(T)\right)}\, dt \\
&\leq \frac{2\sigma}{\hat{\phi}^2} f(\mathcal{G}, \boldsymbol{\Theta})\left(2\sqrt{T} + \left(\frac{\sqrt{8T}}{|\mathcal{V}|} + 4\sqrt[4]{\frac{32\log(|\mathcal{V}|T)T}{|\mathcal{V}|}} + \sqrt{\frac{16}{3}\log(|\mathcal{V}|T)\log(T)}\right)\right. \\
&\quad \left.\left(\sqrt[4]{2|\mathcal{V}|\log(T)} + \sqrt{2\log(T)}\right)\right) \\
&= \mathcal{O}\left(\frac{f(\mathcal{G}, \boldsymbol{\Theta})\sqrt{T}}{\hat{\phi}^2}\left(\sqrt{|\mathcal{V}|} + \sqrt{\log(\overline{T}|\mathcal{V}|)} + \sqrt[4]{|\mathcal{V}\log(\overline{T}|\mathcal{V}|)|}\right)\right),
\end{aligned}
$$

where

$$
f(\mathcal{G}, \boldsymbol{\Theta}) := \left(a_2(\mathcal{G}, \boldsymbol{\Theta}) + \sqrt{2}\mathbb{1}_{\leq 1}(\kappa)w(\partial\mathcal{P})\right)\left(\frac{a_2(\mathcal{G}, \boldsymbol{\Theta}) + \sqrt{2}\mathbb{1}_{\leq 1}(\kappa)w(\partial\mathcal{P})}{a_1(\mathcal{G}, \boldsymbol{\Theta})\min_{\mathcal{C}\in\mathcal{P}}\sqrt{c_{\mathcal{G}}(\mathcal{C})}} + 1\right).
$$

We upper bounded the sum with an integral i.e. $\sum_{t=1}^T f(t) \leq \int_0^T f(t)dt$ for monotonically decreasing functions $f(t)$ in the last inequality. Also $b$ is the bound on the concentration of the fraction of observation per task provided by Lemma 5. For $t_0 = \sqrt{|\mathcal{V}|}$ we find by inserting the result to Lemma 5 for all $t > t_0$:

$$\frac{1}{|\mathcal{V}|} + 2\sqrt{\frac{1}{t|\mathcal{V}|}\log\frac{|\mathcal{V}|}{\delta}} + \frac{4}{3t}\log\frac{|\mathcal{V}|}{\delta} \leq \frac{1}{|\mathcal{V}|} + 2\sqrt{\frac{2\log\left(|\mathcal{V}|\sqrt{|\mathcal{V}|}\right)}{\sqrt{|\mathcal{V}|}|\mathcal{V}|}} + \frac{8\log\left(|\mathcal{V}|\sqrt{|\mathcal{V}|}\right)}{3\sqrt{|\mathcal{V}|}}$$

$$= \frac{1}{|\mathcal{V}|} + \frac{2}{\sqrt{|\mathcal{V}|}}\left[\sqrt{\frac{3}{\sqrt{|\mathcal{V}|}}\log(|\mathcal{V}|)} + 2\log(|\mathcal{V}|)\right]$$

$$= \mathcal{O}\left(\frac{\log(|\mathcal{V}|)}{\sqrt{|\mathcal{V}|}}\right) = b.$$

724    Finally we bound the sum over the instantaneous regret term for the bad events:

$$\sum_{t=1}^{T} 2\left(\mathbb{P}\left[G_{\mathrm{pro}}(t)^c | G_{\mathrm{sel}}(t)\right] + \mathbb{P}\left[G_{\mathrm{RE}}(t)^c | G_{\mathrm{sel}}(t)\right] + \mathbb{P}\left[G_{\mathrm{sel}}(t)^c\right]\right)$$

725    By construction, we have $\max(\mathbb{P}\left[G_{\mathrm{pro}}(t)^c | G_{\mathrm{sel}}(t)\right], \mathbb{P}\left[G_{\mathrm{sel}}(t)^c\right]) \leq \delta(t) = \frac{1}{t^2}$. Hence,

$$\sum_{t=1}^{T} \mathbb{P}\left[G_{\mathrm{pro}}(t)^c | G_{\mathrm{sel}}(t)\right] + \mathbb{P}\left[G_{\mathrm{sel}}(t)^c\right] \leq 2\sum_{t=1}^{T}\frac{1}{t^2} \leq 2\left(1 + \int_1^T \frac{dt}{t^2}\right) \leq 4 \qquad (77)$$

726    As for the RE condition event, letting $A := \dfrac{3\gamma^2\min_{\mathcal{C}\in\mathcal{P}}(\tilde{c}_{\mathcal{G}}(\mathcal{C}) \wedge \tilde{c}_{\mathcal{G}}^2(\mathcal{C}))}{6b + 2\sqrt{2}\gamma}$, we have for any $t_0 \geq 1$

$$\sum_{t=t_0}^{T} \mathbb{P}\left[G_{\mathrm{RE}}(t)^c | G_{\mathrm{sel}}(t)\right] \leq 6d|\mathcal{V}|\sum_{t=t_0}^{T}\exp(-At) \quad \text{(by Theorem 2)}$$

$$\leq 6d|\mathcal{V}|\frac{e^{-At_0}}{1 - e^{-A}} \leq 6d|\mathcal{V}|e^{-At_0}\left(1 + \frac{1}{A}\right)$$

$$\leq 6d|\mathcal{V}|e^{-At_0}\left(1 + \frac{1}{A}\right)$$

727    where in the last line, we used the inequality $\exp(A) \geq A + 1$. Hence, for any $u > 0$, choosing

$$t_0 = \left\lceil\sqrt{|\mathcal{V}|}\right\rceil \vee \left\lceil\frac{1}{A}\log\left(\frac{6d|\mathcal{V}|(1 + \frac{1}{A})}{u}\right)\right\rceil$$

728    implies that $\sum_{t=t_0}^{T}\mathbb{P}\left[G_{\mathrm{RE}}(t)^c | G_{\mathrm{sel}}(t)\right] \leq u$. Before we continue with the regret bound, we need to
729    find an appropriate bound on $\frac{f(\mathcal{G},\boldsymbol{\Theta})}{\hat{\phi}^2}$. Given our result in Theorem 1 and assuming that $\kappa > 1$, we
730    get:

$$\frac{f(\mathcal{G},\boldsymbol{\Theta})}{\hat{\phi}^2} = \frac{\alpha_0 a_2(\mathcal{G},\boldsymbol{\Theta})}{\hat{\phi}^2}\left(\frac{a_2(\mathcal{G},\boldsymbol{\Theta})}{a_1(\mathcal{G},\boldsymbol{\Theta})\min_{\mathcal{C}\in\mathcal{P}}\sqrt{c_{\mathcal{G}}(\mathcal{C})}}+1\right)$$

$$= \frac{\left(\sqrt{2}\kappa w(\partial\mathcal{P})\max_{\mathcal{C}\in\mathcal{P}}\sqrt{\iota_{\mathcal{G}}(\mathcal{C})}\alpha_0+1\right)\left(\frac{\sqrt{2}\kappa w(\partial\mathcal{P})\max_{\mathcal{C}\in\mathcal{P}}\sqrt{\iota_{\mathcal{G}}(\mathcal{C})}\alpha_0+1}{\alpha_0\left(\min_{\mathcal{C}\in\mathcal{P}}\sqrt{c_{\mathcal{G}}(\mathcal{C})}-2\kappa w(\partial\mathcal{P})\right)-1}+1\right)}{1-\gamma\left(1+\frac{\sqrt{2}\kappa w(\partial\mathcal{P})\max_{\mathcal{C}\in\mathcal{P}}\sqrt{\iota_{\mathcal{G}}(\mathcal{C})}\alpha_0+1}{\alpha\left(1-\frac{2\kappa w(\partial\mathcal{P})}{\min_{\mathcal{C}\in\mathcal{P}}\sqrt{c_{\mathcal{G}}(\mathcal{C})}}\right)-\frac{1}{\min_{\mathcal{C}\in\mathcal{P}}\sqrt{c_{\mathcal{G}}(\mathcal{C})}}}\right)^2}$$

$$= \mathcal{O}\left(\frac{\max_{\mathcal{C}\in\mathcal{P}}\iota_{\mathcal{G}}(\mathcal{C})+\max_{\mathcal{C}\in\mathcal{P}}\sqrt{\iota_{\mathcal{G}}(\mathcal{C})}+1}{\min_{\mathcal{C}\in\mathcal{P}}\sqrt{c_{\mathcal{G}}(\mathcal{C})}}+\max_{\mathcal{C}\in\mathcal{P}}\iota_{\mathcal{G}}(\mathcal{C})+1\right)$$

$$= \mathcal{O}\left(\frac{1}{\min_{\mathcal{C}\in\mathcal{P}}\sqrt{c_{\mathcal{G}}(\mathcal{C})}}\right).$$

The first big $\mathcal{O}$ notation is obtained due to the fact that for for large $\min_{\mathcal{C}\in\mathcal{P}}\sqrt{c_{\mathcal{G}}(\mathcal{C})}$ and small $\max_{\mathcal{C}\in\mathcal{P}}\sqrt{\iota_{\mathcal{G}}(\mathcal{C})}$ the denominator term i.e. $\hat{\phi}^2$ behaves like $1-\gamma$, which leaves the numerator dominating the rest of the term. Now, we simply have to insert all our results into the sum of instantaneous regrets:

$$\mathcal{R}(\overline{T}) \leq t_0 + 2u + 8 + \mathcal{O}\left(\frac{f(\mathcal{G},\boldsymbol{\Theta})\sqrt{\overline{T}}}{\hat{\phi}^2}\left(\sqrt{|\mathcal{V}|}+\sqrt{\log(\overline{T}|\mathcal{V}|)}+\sqrt[4]{|\mathcal{V}\log(\overline{T}|\mathcal{V}|)|}\right)\right)$$

$$\leq \left\lceil\sqrt{|\mathcal{V}|}\right\rceil + \left\lceil\frac{1}{A}\log\left(\frac{6d|\mathcal{V}|(1+\frac{1}{A})}{u}\right)\right\rceil + 2u + 8$$

$$+ \mathcal{O}\left(\frac{f(\mathcal{G},\boldsymbol{\Theta})\sqrt{\overline{T}}}{\hat{\phi}^2}\left(\sqrt{|\mathcal{V}|}+\sqrt{\log(\overline{T}|\mathcal{V}|)}+\sqrt[4]{|\mathcal{V}\log(\overline{T}|\mathcal{V}|)|}\right)\right)$$

$$\leq \left\lceil\sqrt{|\mathcal{V}|}\right\rceil + \left\lceil\frac{1}{A}\log(12d|\mathcal{V}|(1+A))\right\rceil + \frac{1}{A} + 8$$

$$+ \mathcal{O}\left(\frac{f(\mathcal{G},\boldsymbol{\Theta})\sqrt{\overline{T}}}{\hat{\phi}^2}\left(\sqrt{|\mathcal{V}|}+\sqrt{\log(\overline{T}|\mathcal{V}|)}+\sqrt[4]{|\mathcal{V}\log(\overline{T}|\mathcal{V}|)|}\right)\right)$$

$$\leq \left\lceil\sqrt{|\mathcal{V}|}\right\rceil + \left\lceil\frac{1}{A}\log(12d|\mathcal{V}|(1+A))\right\rceil + \frac{1}{A} + 8$$

$$+ \mathcal{O}\left(\frac{f(\mathcal{G},\boldsymbol{\Theta})\sqrt{\overline{T}}}{\hat{\phi}^2}\left(\sqrt{|\mathcal{V}|}+\sqrt{\log(\overline{T}|\mathcal{V}|)}+\sqrt[4]{|\mathcal{V}\log(\overline{T}|\mathcal{V}|)|}\right)\right)$$

$$= \mathcal{O}\left(\frac{1}{A}\log(d|\mathcal{V}|)+\sqrt{\frac{\overline{T}}{\min_{\mathcal{C}\in\mathcal{P}}c_{\mathcal{G}}(\mathcal{C})}}\left(\sqrt{|\mathcal{V}|}+\sqrt{\log(\overline{T}|\mathcal{V}|)}+\sqrt[4]{|\mathcal{V}\log(\overline{T}|\mathcal{V}|)|}\right)\right),$$

where we set $u = \frac{1}{2A}$ in the third inequality.

$\square$

## C  Additional related work

**Homophily and modularity in social networks**    Given the large number of users on social networks, one may be able to learn their preferences more quickly by leveraging the similarities between them. This idea relies on the notion of *homophily* in social networks McPherson et al. [2001], Easley et al. [2010]. In modelling social networks, users' preferences relationships are encoded in a graph, where neighboring nodes are users with similar preferences. This graph can be known *a priori* or it can be inferred from previously collected feedback Dong et al. [2019]. Exploiting this information and integrating them into bandit algorithms can lead to a significant increase in performance Yang et al. [2020]. Indeed, the knowledge of user relations allows the algorithm to tackle the data sparsity issue that is inherent to bandit settings.

Another fundamental point that can be used for integration of information from social networks is that, social networks show large *modularity* measures Newman [2006] Borge-Holthoefer et al. [2011]. This implies that we have high density of edges within clusters and low density of edges between clusters. As a result, users can be clustered based on the graph topology and a preference vector can be learned for each cluster, substantially reducing the dimensionality of the problem. In other words, discovering the clustering structure of users can reduce the computational burden of large social networks. Consequently, there have been attempts in exploiting the clustered structures of social networks in bandit algorithms Gentile et al. [2014], Nguyen and Lauw [2014], Yang and Toni [2018], Li et al. [2019], Nourani-Koliji et al. [2023], Cheng et al. [2023].

**Bandit meta-learning**    In contrast to the multi-task setting, meta learning deals with sequentially arriving tasks that have to be learnt and generalizing the gained information to improve performance for future tasks. Here, as in the multi-task setting, it is assumed that the tasks share some common structure that is ought to be learnt and exploited. In the work of Bilaj et al. [2024] it is assumed that the tasks were sampled from a common distribution such that they are concentrated around an affine subspace, which is learnt through PCA algorithm. The resulting projection matrices could then be exploited to improve learning for new tasks in an adapted UCB and Thompson sampling approach. Other lines of work are Cella et al. [2020], Kveton et al. [2021], Basu et al. [2021], which learns the mean of the distribution under the assumption that the covariance of the prior is known or Peleg et al. [2022] which generalizes this assumption and attempts to learn the covariance as well.

## D  Additional experimental details

### D.1  About experiments of the main paper

The experiments have been conducted with an intel i7 CPU with 12 2.6 GHz cores and 32 GB of RAM. The two experiments with the highest number of tasks (200) and dimension (80) take about 8 hours, parallelized over the 12 cores.

To generate clusters, we generate $|\mathcal{P}|$ variables $v_{i i \in \mathcal{P}}$ from the uniform distribution, then we use them to construct a categorical distribution with probabilities proportional to $e^{v_i}$. These probabilities defines the cluster proportions.

### D.2  Solving the Network Lasso problem

We implement the Primal-Dual algorithm proposed in Jung [2020] to solve the Network Lasso problem but we do not vectorize the matrices (in the sense of stacking their columns into a vector), which speeds up computation.

### D.3  Algebraic connectivity vs topological centrality index

Given two fully connected graphs weightless $\mathcal{G}_1$ and $\mathcal{G}_2$ with size 100 each, we progressively link them by edges, we construct the Laplcian $\mathbf{L}$ of the resulting graph $\mathcal{G}$. We measure the minimum topological centrality index $\min_{1 \leq i \in 200}(L_C^\dagger)_{ii}^{-1}$, and the algebraic connectivity, i.e. the minimum non-null eigenvalue of $L$.

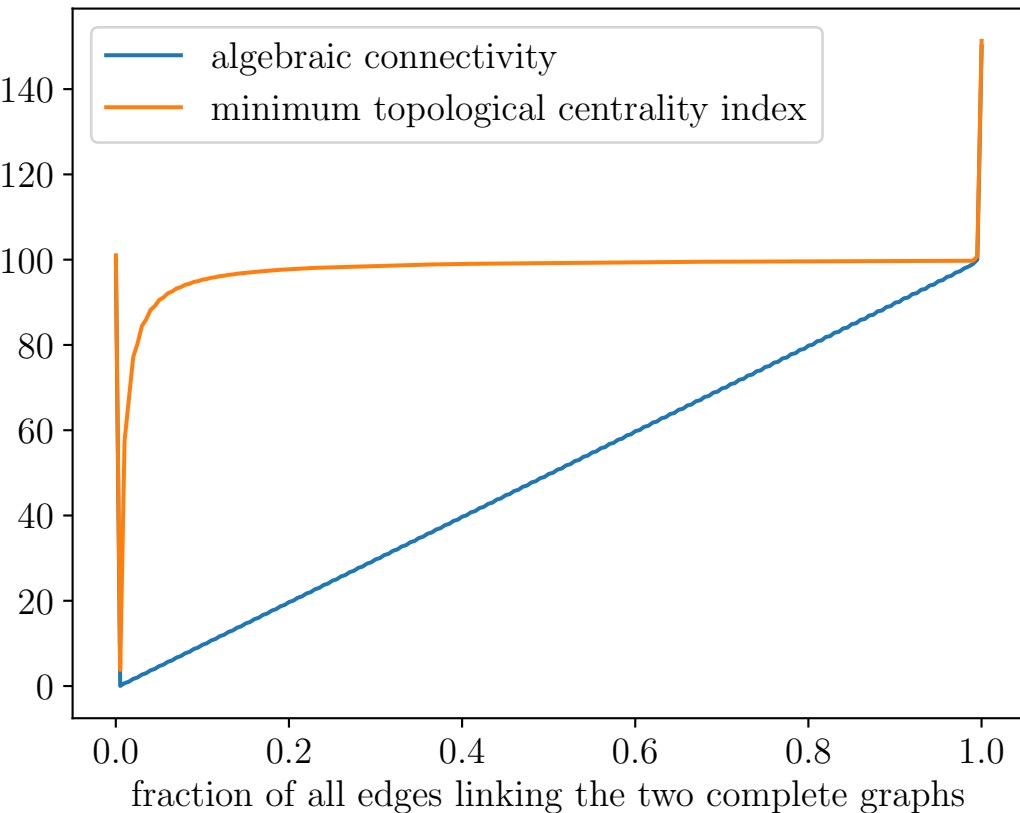

Figure 2: Minimum Topological centrality index vs Algebraic Connectivity, for a graph formed by connecting two fully connected initial graphs $\mathcal{G}_1, \mathcal{G}_2$ with size 100 each.

Clearly, the minimum topological centrality index grows faster than the algebraic connectivity in this case, and seems to saturate at some level that is reached in a linear progress by the algebraic connectivity.

### D.4 Limitations

The first limitation of the paper is the restriction to the setting of i.i.d generated action sets. This restriction is common to all papers relying on Lasso-type optimization objectives [Bastani and Bayati, 2019, Oh et al., 2021, Cella and Pontil, 2021, Ariu et al., 2022, Cella et al., 2023]. Also, we do not provide a lower bound for the regret, a challenge that we let for future work. Besides, our optimization problem is not strongly convex, which can be mitigated by adding a squared $L^2$ norm regularization. However, such an addition would probably drastically change the theoretical analysis.

### D.5 Broader Impacts

As our method can be applied to transfer knowledge between users of a recommender system, it has the potential to improve their overall experience by learning their preferences quickly. However, one must be careful with the strength of the integrated prior knowledge as it can lead to an adverse effect of slowing down the learning process.

