# OpenReview forum: "Network Lasso Bandits"
_NeurIPS.cc/2024/Conference — Submitted to NeurIPS 2024_

### Official Review · Reviewer_3afv · 2024-06-16

**Soundness:** 2
**Presentation:** 2
**Contribution:** 2
**Rating:** 3
**Confidence:** 5

**Summary:**

The authors propose to use network Lasso to learn a multi-task bandit problem with given network structure. More specifically, the network structure has pre-defined unknown clustering structure, where within each cluster all the bandit tasks share the same model. The authors propose a bandit algorithm that can learn and provide a sublinear guarantee. The key difference of this paper with GOBLin in Cesa-Bianchi et al., 2013 is that this paper uses a network Lasso (or something like a group Lasso) penalty while GOBLin use a ridge penalty.

**Strengths:**

Even though I think the authors overclaimed their contributions, which I will state below, I still feel it's meaningful to discuss network Lasso and design a bandit algorithm based on certain network structure, given the limited literature on multitask bandit. Compared to previous network bandit literature such as Cesa-Bianchi et al., 2013, this paper characterizes the network structure in more detail.

**Weaknesses:**

1. I think the authors need to provide more real-world examples to show why their network structure (and the correpondingly induced network Lasso bandit algorithm) is pratical, instead of stating their algorithm is good because it provides piecewise constant property in constrast to smoothness of GOBLin in Cesa-Bianchi et al., 2013. The key assumption in this paper is that the network structure is given, and the network can be splitted into connected clusters, within which the task parameters are the same. Can authors find or describe a couple of pratical examples/datasets where such a network exists, given that bandit is a very pratical problem?

2. The literature review that compare with the previous literature are not very accurate and sufficient IMO. For example, the authors mention Gentile et al., 2014 and Li et al., 2019 can cause overconfidence in constructing clusters. However, these algorithms do not have prior information about clusters such as a given network and thus they have to learn the clusters conditioned on the task similarities. In that sense, these algorithms are more pratical because oftentimes in practice network information is lacking. Here one should also add a related reference Context-Based Dynamic Pricing with Online Clustering by Miao et al., 2022. There are also robust multitask bandit algorithms (e.g., Multitask Learning and Bandits via Robust Statistics by Xu and Bastani, 2024) that can also solve the network bandit problem if the network structure follows certain assumptions; Multi-Task Learning for Contextual Bandits by Deshmukh et al., 2017 discuss a multitask bandit problem but use kernal based method. I suggest the authors add a more detailed literature review to discuss their paper's connection with these current multitask bandit algorithms.

3. Typically, greedy algorithm (the method proposed in this paper is greedy too due to Assumption 2) has better performance in bandit simulations compared to UCB-based algorithm (see Bastani and Bayati, 2020). Therefore, it's not a fair comparison in Figure 1, where all benchmarks are UCB algorithms. I think it's necessary to add the following benchmarks: OLS-Bandit or Lasso-Bandit (Bastani and Bayati, 2020) without task sharing, and Cella et al., 2023 which use low-rank structure to do multitask bandit, and also a few others mentioned above as a benchmark to show that the network structure indeed helps, fixing the difference due to UCB and greedy algorithms.

4. I feel it's a false claim that the regret bound in Theorem 3 "doesn't depend on the dimension" and it's due to the concentration inequality from Hsu et al., 2012. Intuitively, the regret bound should depend on the dimension unless one assume that the number of tasks in a cluster is d-dependent so the regret bound is smaller compared to a typical single bandit regret bound. I think the reason why here the bound seems to be d-independent is because the dimension $d$ is hiden in the problem-dependent parameter $\phi$. Since the authors assume the context $x$ has norm 1, the minimum eigenvalue of $E[xx^\top]$ should scale as $1/d$, and hence $\phi^2$. I don't think a typical tail inequality in Hsu et al., 2012 can improve the bound regarding the context dimension.

5. I think the asymptotic assumptions in Theorem 3 is incompatible with the finite-sample analysis in a typical bandit analysis and looks weird. I suggest the authors keep the isoperimetric ratio and centrality index as part of the regret bound (instead of forcing them out using the asymptotic assumptions), even though it might add an additional T-linear term due to the misspecification error caused by the inter-cluster edge connections. I feel that's the case because in the extreme case when each cluster has size 1, there will be misspecification error penalizing tasks connected the inter-cluster edges towards each other. I think it's totally fine to have such non-sublinear terms to provide a more comprehensive understanding of the limit of such network structures.

I am willing to raise my rating if the authors can solve my questions and concerns.

**Questions:**

See my points above.

---

> ### Author Rebuttal · Authors · 2024-08-07
>
> ## Providing a real-world example
> We would like to point out that the bulk of the contributions are theoretical. They lie especially in the establishment of the oracle inequality, and ensuring the RE condition for the empirical covariance matrix.
>
> With that being said, the cluster structure is well-motivated in the literature, especially with a high number of tasks. Practical examples for this setting can be social networks, where muliple similar uses can be linked within a graph structure or personalized medicine, where the links in the graph are represented by physical proximity. Especially in the social network setting the number of users i.e. the number of nodes can grow very large, making traditional models infeasible. Under that setting, our work answers how to exploit a certain structure when it is available in addition to the clustering goal, which is a standard approach in multi-task learning. It can even be used as a way to compress the feature vectors data given a large dimension and a large number of users.
>
> Concerning the comparison to methods that do not rely on a graph, we understand the reviewer's concern, but it is common in the literature to compare the methods that do not exploit a given structure. For example, the Lasso bandit papers of [1,3], the authors compare to the LinUCB algorithm, which relies on the principle of optimism in the face of uncertainty, despite the action sets being generated i.i.d.
> ## Inaccuracy and Insufficiency of the literature
> We apologize for any inaccuracy or insufficiency in the literature, and we will take the references pointed out by the reviewer into account.
>
> ## Comparison to OLS bandits, and low-rank multi-task bandits
> * OLS: we implemented a simple algorithm that uses the ordinary least squares estimator for each task independently.
> * For the low-rank multi-task bandits [6],  following the authors' recommendation, we implemented the *TraceNormBandit* algorithm and used the accelerated gradient method presented in [9]. Interestingly, its performance is most of the times second to our approach. This can be due to the fact that the cluster structure of $\mathbf \Theta$ can be mathematically written as $\mathbf{\Theta} = \sum_{C \in \mathcal{P}}\mathbf{1}_C\mathbf{\theta}_C^\top$, where $\mathbf{1}_C$ is the indicator vector of cluster $C$ (coordinates equal to 1 on the nodes belonging to $C$ and zeros elsewhere) and $\mathbf{\theta}_C$ is the true vector of every node in $C$. It is clear that the range of $\mathbf \Theta$ is equal to the span of $\{\mathbf{1}_C; C \in \mathcal{P}\}$, implying that its rank is at most equal to $\min(d, |\mathcal{P}|)$. It will then satisfy the low rank assumption for $|\mathcal{P}| < d$.
>
> ## Dependence on the dimension
> We kindly invite the reviewer to look at the general rebuttal for a first answer on this issue.
> In addition to that, we appreciate the reviewer's observation about the dimension that could be hidden in the $\phi$ factor, although $\phi$ does not necessarily correspond to the smallest eigenvalue. Also, we point out that we specify that only one regret term is independent of the dimension, not the whole regret itself. The term that is due to ensuring the RE condition for the empirical $\bf\Sigma$ matrix depends on it.
> For the first part, we will instead write that our regret has better dependence in the dimension and $\phi$ than other LASSO-type contributions in the literature.
> With that being said, we note that in the works of [1,2,3,4,5,6], the dependence of $\phi$ on parameter problems such as the dimension has not been discussed.
>
> As a result, we will replace our claim after Theorem 3 with "Except for a possible dependence on the dimension hidden in the RE condition constant $\phi$", the part of the regret due to the empirical process does not depend on the dimension.
>
> ## Problem with asymptotic assumptions in Theorem 3
> We will remove additional asymptotic assumptions from the regret, and point to them either as a comment or as a corollary. Our main aim was to simplify the regret bound. As for the linear dependency and the misspecification error, we do not understand where it can come from and we kindly ask the reviewer for clarification concerning that point. Indeed, our asymptotic assumptions on the graph do not change the horizon dependency, and our assumption on the relation between the horizon per task and the number of users does not change the fact that our regret is sublinear in the total horizon.

---

> ### Comment · Reviewer_3afv · 2024-08-13
> **Reply to the authors**
>
> I thank the authors for their responses and resolve some of my concerns in literature and empirical results. However, the rest of the questions are still not addressed well.
>
> 1. I still feel the authors' explanation about the regret dependence on the dimension is doubtful. Intuitively, as long as the bound is gap-independent, i.e., O(sqrt(T)), and there're no assumptions on model, e.g., sparsity on the parameters, the bound must depend on the dimension d. It could be possible due to the sharing structure that the d cancels out with the number of tasks that share the same parameter. However, I didn't see any assumption on the relation between d and the number of tasks in a clusters. Actually, from the explanation, now I feel it's actually all relevant terms of M that implicitly contains the depedence on d, e.g., the trace of $M \in \mathbb{R}^{d \times d}$ should be proportional to d. But this is not explained anywhere clearly.
>
> 2. My intuition about potential misspecification error that will be linearly dependent on the time T is as follows. The network structure here has both inter-cluster and intra-cluster edges. If the parameters within the same cluster are the same and the model allows heterogeneity across clusters, the penalty w_{m, n} \|\theta_m - \theta_n\|_2 will introduce extra bias if m and n do not belong to the same cluster. The trick the authors use to avoid this misspecification error is to use asymptotics assuming that inter-cluster edges are sparse. However, without this assumption and only with a static network assumption, i.e., the network is fixed, this will introduce linear term on T. I am curious after removing additional asymptotic assumptions how the authors can provide a bound that doesn't contain misspecification error, without assuming away the intra-cluster edges.
>
> I'm happy for further discussion but would like more clear intuition and explanations on my questions. According to the above technical concerns, which is important for a theory paper, I won't change my score for now.

---

> ### Author Response · Authors · 2024-08-13
> **About the dimension dependence and the possible linear regret (part 1)**
>
> We thank the reviewer for the raised issues, and we address them below.
> ## Dependence on the dimension
> We restate that our regret does depend on the dimension logarithmically, such dependence being the result of ensuring that the restricted eigenvalue condition (Definition 2 and Assumption 4) holds for the empirical multi-task Gram matrix. The other part of the regret, the result of the oracle inequality bound, does not depend on the dimension. We kindly invite the reviewer to check our proofs in the supplementary material. To facilitate navigating through the proofs from Lemma 1 to Theorem 1, where we intentionally separated the results concerning deterministic inequalities (Lemma 1 and Lemma 2), and probabilistic ones (from Lemma 3 and Proposition 4). We provide the following explanation of every technical result's role:
>
> * Lemma 1 follows from the optimality of $\hat{\mathbf{\Theta}}$ and the piecewise constant assumption on $\mathbf{\Theta}$. Indeed, the total variation norm of $\mathbf{\Theta}$ only comes from the differences between the parameter vectors across $\partial\mathcal{P}$.
> * Lemma 2 bounds the total variation of the error signal by leveraging a graph-based decomposition, stated in Proposition 2 (where we decompose the identity matrix $\mathbf{I}\_{|\mathcal{V}|}$).
> * Lemma 3 relies on Theorem 2.1 of [10]. It amounts to bounding the squared Euclidean norm of $\mathbf{X}\_{\mathcal{V}}^\top\mathbf{\eta}$, where $\mathbf{X}\_{\mathcal{V}}$ is the $t \times d|\mathcal{V}|$ matrix that is block-diagonal with blocks equal to $\mathbf{X}\_1, \cdots, \mathbf{X}\_{|\mathcal{V}|}$, the matrices obtained by concatenating the row context vectors encountered for every task. This is equivalent to bounding the quadratic form $\mathbf{\eta}^\top M\mathbf{\eta}$, where $M$ is the matrix we defined in the general rebuttal response, and that has size $t \times t$ rather than $d \times d$. Now, considering any matrix square root $N \in \mathbb{R}^{t \times t}$ of the PSD matrix $M$ (i.e. verifying $N^2 = M$), we obtain
>     $$ \Vert \mathbf{X}\_{\mathcal{V}}\mathbf{\eta} \Vert^2 = \mathbf{\eta}^\top M\mathbf{\eta} = \Vert N\mathbf{\eta} \Vert^2.$$
> We make use of $N$ above only because the result in [10] is stated for square matrices. Their $\mu$ vector is null as shown in the proof of Lemma 3, since $\mathbf{\eta}$ is formed of centered noise coordinates. What remains is a part that solely depends on $M$, more precisely on the trace of $M$, the trace of $M^2$, and the spectral norm of $M$. We have $tr(M^2) \leq tr(M)^2$, simply because the left-hand side is the sum of squares of $M$'s eigenvalues, and the right-hand side is the square of their sum, and these eigenvalues are all nonnegative since $M$ is PSD. The spectral norm is the maximum of the eigenvalues and can be bounded by their sum, the trace. Hence, the only dependence of the bound on $\mathbf{X}\_{\mathcal V}$ is via $tr(M)$. We have
> $$tr(M) = tr(\mathbf{X}\_{\mathcal{V}}\mathbf{X}\_{\mathcal{V}}^\top) = \Vert \mathbf{X}\_{\mathbf{\mathcal{V}}}\Vert^2\_F = \sum\_{m \in \mathcal{V}}\Vert{\mathbf{X}\_m}\Vert^2\_F \leq \sum\_{m \in \mathcal{V}} |\mathcal{T}\_m(t)|^2 \leq t^2,$$
> where $\Vert \mathbf{X}\_m\Vert^2\_F$ is the sum of the squared Euclidean norms of context vectors encountered by task $m$, all of which are bounded by $1$ by assumption. In comparison, bounding the stochastic process for the Lasso for example [1,2,3,5] relies on bounding the infinity norm, which introduces the dimension via a union bound.
> * Proposition 4 combines Lemmas 1,2,3
> * Theorem 1 is the only place where we make use of our RE assumption, that assumption introducing two constants $\phi$ and $\kappa$.
>     * For $\phi$, in addition to our statement that no LASSO-type bandit paper discussed its dependence on $d$ or any other problem parameter, we point out that even in the offline learning literature studying the LASSO problem and its related problems, such a dependence is not considered. Still, we indicated that we will indeed mention a potential dependence of $\phi$ on the dimension.
>     * For $\kappa$, the interval to which it belongs depends only on the structure of the graph clusters and the graph boundary.
>
> Concerning the reviewer's remark about the gap-independent bounds being dependent on the dimension, in our case that would be equivalent to studying the worst case of our setting, and stating an instance-independent bound. Here, one must specify what the bandit environment class is. If the environment contains for example all of the signals that are piecewise constant on the fixed graph, then maybe a dependence on the dimension might appear, but that is not what we study.

---

> ### Author Response · Authors · 2024-08-13
> **About the dimension dependence and the possible linear regret (part 2)**
>
> We would like to point out that in references [3,5,6], all of which consider assumptions that are akin to ours, the dependence of $d$ in the $T$-dependent regret term is due to summing up the time-dependent regularization coefficient over time. This regularization term's choice follows from bounding their respective stochastic processes. All of their regularizations are norms ($L\_1$ and nuclear norm for the multi-task setting of [6]), whereas in our case we use a semi-norm. Hence, intuitively, the behavior of the bounds might differ. In addition to that, our semi-norm acts at the level of the nodes, not at the level of the dimension as it is the case in [6].
>
> Other potential sources of dimension dependence might be the constants appearing in the relaxed symmetry and balanced covariance assumptions. However, none of such dependencies was discussed in previous work. We will point out such potential dependence.
>
> ## Potential linear reget due to misspecification
>
> Our regret is based on a converging oracle inequality which holds with high probability. All the event probabilities required for it to hold are discussed in the proofs. The complementary events, which could potentially result in linear regret bounds, have vanishing probabilities associated with them, thus a linear term in the regret is ruled out.
>
> In addition, our asymptotic assumptions in the regret affect two terms:
> * Total horizon $T$, equal to $|\mathcal{V}|\bar{T}$, on which the regret depends sublinearly.
> * Term $f(\mathbf{\Theta}, \mathcal{G})$ that does not have any dependence on time, hence no dependence on the horizon, and no possibility to cause a linear horizon dependence.
>
> Such assumptions are not made to assume that inter-cluster edges are sparse, they are made to simplify the bound for the reader, and to offer more intuition on the dependence on the graph, but cannot introduce a linear dependence, as explained for the $f(\mathbf{\Theta}, \mathcal{G})$ term. Our RE assumption, and especially the part concerning the RE-norm, accounts for the inter-cluster variations. Indeed, it involves the  $(1-\kappa)^+\Vert \mathbf{B}\_{\partial\mathcal{P}}^\dagger\mathbf{B}\_{\partial\mathcal{P}}\mathbf{Z}\Vert$ term which clearly depends on the boundary. However, for a well-behaving graph, we can assume that $\kappa>1$ (kindly refer to the response to reviewer QF4d, in which we express more intuition on this aspect), and even for $\kappa < 1$, no linear dependence is possible. Indeed, it is sufficient to examine the proof of the oracle inequality to see that it has a $\tilde O(\cfrac{1}{\sqrt{t}})$ dependence  (where $\tilde O$ hides logarithmic factors), which results in the $\sqrt{T}$ part in the regret.

---

### Official Review · Reviewer_Z89V · 2024-07-12

**Soundness:** 3
**Presentation:** 3
**Contribution:** 3
**Rating:** 5
**Confidence:** 3

**Summary:**

In this paper, authors work under the multi-task contextual bandit settings by representing the task correlations through the graph structure. To solve this problem, authors propose an algorithm that utilizes a linear regression formulation with Lasso constraint in terms of the node connectivity. Theoretical analysis as well as experiments against several baselines are presented to demonstrate the effectiveness of proposed method.

-	The paper is generally well-written, with crisply clear descriptions of required assumptions, and the proposed solution is intuitive and well-motivated.

-	Good empirical performances. Authors compare proposed algorithm with several clustering of bandits baselines, showing the effectiveness of the proposed method. The performance gain over existing methods is impressive.

-	Novel theoretical analysis roadmap. Overall, the theoretical analysis pipeline is novel and the looks promising to me. With the additional introduced RE assumption, authors are able to improve the regret bound to $\tilde{O}(\sqrt{\bar{T}})$ instead of the vanilla time horizon.

-	My major question is regarding the numerous assumptions required for the theoretical analysis. For instance, in Assumption 1, authors assume the candidate arms across different arounds are generated i.i.d. from a fixed distribution. This is different from existing clustering of bandits works, where the candidate arm contexts in each round is conditioned on previous observed arms. In this case, assumption 1 is somehow deviates from the actual applications of recommender systems where the candidate arms of each round are refined along with more information collected from the environment.

-	For the experiments, authors have compared against multiple clustering of bandit works. In this case, it would be good if authors can include additional discussions comparing your theoretical outcomes with those of existing clustering of bandits works, which can offer more intuitive comparison with existing approaches.

**Strengths:**

Please see my comments above.

**Weaknesses:**

Please see my comments above.

**Questions:**

Please refer to my questions above.

**Limitations:**

Please see my comments above.

---

> ### Author Rebuttal · Authors · 2024-08-07
>
> ## Assumption on the arm-generating process
> Kindly refer to the general rebuttal where we address this point.
>
> ## Comparing the theoretical outcomes
> We will take the advice and add comparisons of theoretical results to the used baselines. Here we will compare with previous works in clustering and multi-task learning:
> From Theorem 3, our result yields a regret bound of $\mathcal{O}\left(\sqrt{\frac{\bar{T}}{c}}\left(\sqrt{|V|}+\sqrt{\log(\overline{T}|\mathcal{V}|)}+\sqrt[4]{|\mathcal{V}|\log(\bar{T}|\mathcal{V}|)}\right)+\frac{1}{A}\log(d|\mathcal{V}|)\right)$, where $c$ denotes the minimum topological centrality index of any cluster.
>
> Compared to the result of Gentile et. al. 2014 (clustering approach) $\tilde{\mathcal{O}}\left(\left(\sigma\sqrt{d|\mathcal{P}||\mathcal{V}|\bar{T}}+ \sqrt{|\mathcal{P}||\mathcal{V}|\bar{T}}\right)\left(1+\sum_{j=1}^{|\mathcal{P}|}\sqrt{\frac{v_j}{\lambda|\mathcal{V}|}}\right)\right)$ (Note that log dependencies vanish in their notation) we provide a better dependency on the dimension. Also, our result does not depend on the number of clusters $|\mathcal{P}|$, instead, our result depends on the topological centrality index, since we do not aim to learn the cluster structure explicitly.
>
> The result of Cella et. al. 2023 (multi-task setting) yields: $\tilde{\mathcal{O}}(|\mathcal{V}|\sqrt{r\bar{T}}+\sqrt{rd|\mathcal{V}|\bar{T}})$. Apart from the better dependency on the dimension, our result showcases a better dependency on the number of tasks $|\mathcal{V}|$. Here a convenient comparison would be between the first term of our bound $\sqrt{\frac{\bar{T}|\mathcal{V}|}{c}}$, which is not logarithmic and depends on $c$, and the first term in Cellas bound $|\mathcal{V}|\sqrt{r\bar{T}}$, which essentially amounts to the part of their regret, in which an oracle is aware of the low-rank structure. Here we could draw an analogy between the rank $r$ and the minimum topological centrality index of any cluster $c$. Under optimal conditions, $r$ would be low, and analogously $c$ would be large. Between these two non logarithmic terms our result showcases a better dependency on the number of tasks.

---

> ### Comment · Reviewer_Z89V · 2024-08-12
> **Thank you for your response.**
>
> I thank authors for your detailed discussion in terms of theoretical comparisons with clustering of bandit works. Although I still think it can be strong to have the i.i.d. assumption for real-world bandit learning scenarios (maybe for this series of works in Lasso Bandits), I will keep my current positive score given authors' theoretical contributions.

---

### Official Review · Reviewer_QF4D · 2024-07-16

**Soundness:** 2
**Presentation:** 2
**Contribution:** 2
**Rating:** 5
**Confidence:** 2

**Summary:**

This paper addresses the multi-task bandit problem using graph information. The given graph represents the relationships between tasks. Assuming that the preference vector of clustered tasks is constant, the problem is formulated as a network lasso problem to estimate the lasso estimator.
A modified restricted eigenvalue condition, commonly used in high-dimensional statistics, is defined to derive the oracle inequality for the network lasso estimator on non-i.i.d. data. The oracle inequality of the proposed network lasso estimator is derived under the assumption that the true multi-task Gram matrix satisfies the adapted RE condition.
Based on the derived oracle inequality, a greedy-type algorithm is presented, achieving $\sqrt{T}$ regret. Numerical experiments support the theoretical performance of the proposed algorithm.

**Strengths:**

- The proposed algorithm efficiently learns task preference vectors by using graph information that encodes relationships between tasks. Specifically, it employs a network lasso estimator under the assumption that preferences within clustered tasks are constant, demonstrating its effectiveness in high-dimensional contexts.
- The paper adapts the restricted eigenvalue condition from high-dimensional statistics to the graph-based multi-task bandit setting. Based on the adapted RE condition, they established oracle inequality for network lasso estimator and showed that the proposed algorithm achieved $\sqrt{T}$ regert even though I haven't verified every proof in detail.
- The algorithm's performance seems robust even as the number of tasks and dimensions increase.

**Weaknesses:**

- Since I'm not very familiar with the graph-based multi-task bandit setting, it may be that the concepts explaining the restricted eigenvalue condition (Def 2) are too heavy. It would be helpful to include comparisons or examples from existing RE conditions in high-dimensional statistics or high-dimensional contextual bandits to improve understanding.

**Questions:**

1. You mentioned that the proposed algorithm is cluster-agnostic, but if a graph representing the relationships between tasks is given, can't we identify which tasks are clustered? Compared to Gentile et al. (2014), it seems that the problem setting uses additional information. If the same information were given to Gentile et al. (2014), what advantages does the proposed algorithm have compared to not needing clustering?
2. Is the estimation error in the oracle inequality (Theorem 1) dimension($d$)-independent?
3. Can the restricted eigenvalue condition be transformed into a compatibility condition?
4. What is the definition of $\overline{Z}_P$?
5. The regret bound of high-dimensional contextual bandit using the Lasso estimator is proportional to the sparsity level (e.g., usually denoted by $s_0$) instead of the feature dimension $d$. What does sparsity correspond to in the network lasso instance, and how does it appear in the regret bound?

**Limitations:**

The authors have well-addressed the limitations in Appendix D.4 and further research directions in Section 7.

The content discussed in this paper appears to have little to no negative societal impact.

---

> ### Author Rebuttal · Authors · 2024-08-07
>
> ## Explaining the RE condition and relating it to its counterpart in high-dimensional statistics
> For our RE condition, if we do not restrict the signal $Z$ to the cone $\mathcal{S}$, and if we replace the RE norm with the Frobenius norm, we obtain a non-null minimum eigenvalue condition. If we assume that $\kappa>1$, then we are left with the Frobenius norm of $\overline{Z}_{\mathcal{P}}$.  The latter represented the orthogonal projection of the signal $Z$ onto the space of constant signals per cluster. This is analogous to the fact that an RE condition in the LASSO case involves the Euclidean norm of a vector restricted to its coordinates in the sparsity set, which is its projection onto the space of vectors having null coordinates outside of the sparsity set.
>
> ## Identifying the clusters
> Clusters can indeed be estimated using the graph. However, we will have to deal with their uncertainty, as mistaking the cluster to which a node belongs can cause estimation errors to accumulate. Instead, we do not rely on an explicit cluster estimation, and we rather use a regularization that uses it implicitly.  To draw an analogy with the LASSO estimator, estimating the clusters explicitly that would be similar to estimating the set of indices of non-null features of the parameter vector in LASSO regression
>
> ## Advantage provided to the CLUB algorithm given the graph
> The implementation of the CLUB algorithm from [7] takes a graph as an input, which is not necessarily the complete graph. That is what we ensured in our experiments, but still, our algorithm performed better.
>
> ## Dependence of the oracle inequality on the dimension
> Kindly refer to the general rebuttal for this issue.
>
> ## Possibility of transforming the RE condition to a compatibility condition
> Our RE condition can be transformed into a compatibility condition, as the compatibility condition is more general: it only requires the concerned quadratic form to be bounded from below by a constant multiplied by a convenient choice of norm. In that sense, the RE condition can be seen as a particular choice of "compatibility". The choice of such norm depends on the norms chosen for the Hölder inequality used to bound the inner product between our empirical process $\bf K$ and the error signal $\bf E$ (Lemma 3 and Proposition 4), where we used Cauchy-Schwarz for matrices (i.e. bounding by the product of Frobenius norms).
>
> We hesitated between using a compatibility condition or an RE one, but we think that the RE one is more interpretable as it is a more straightforward generalization of the least eigenvalue or a function's curvature in general. We recommend reading [11] for intuition on how the RE condition points out at curvature in some directions in space.
>
> ## Definition of $\overline{\bf Z}\_{\mathcal P}$
> We apologize for the typo of writing ${\bf Z}\_{\mathcal P}$ instead of $\overline{\bf Z}\_{\mathcal P}$.  As mentioned in line 175 it is the signal obtained by replacing each node vector with the average vector of the true cluster containing it.
>
> ## Sparsity parameter counterpart in our case
> In [1,2,3], the regret part that dominates in terms of the horizon dependency is proportional to $s_0$ the size the sparsity set. This part is obtained by summing up all of the bounds of the oracle inequality under well-defined good events. Applying that reasoning to our case, it is sufficient to look at the result of the oracle inequality (Theorem 1), and in particular at the term $f(\mathcal{G},\mathbf{\Theta})$ that we simplify in Theorem 3 using additional assumptions on the asymptotic behavior of the clustering.
>
> For the sake of simplicity, let us first consider the case where $\kappa>1$. This case is intuitive as it is possible for example when the total weight of the signal boundary is negligible compared to the minimum topological centrality indices of clusters (that we denote by $c$ here to reduce the clutter): $w(\partial\mathcal{P}) \ll c$. Such a condition expresses some coherence between the graph and the clustering.
> Under $\kappa>1$, we have $f(\mathcal{G},\mathbf{\Theta}) = \cfrac{a_2^2}{a_1 \sqrt{c}} + a_2$. Let us now take a look at the expressions of $a_1$ and $a_2$ given in Definition 2. On the one hand, $a_2$ decreases with the product $w(\partial \mathcal{P})\iota$, where $\iota$ here denotes the maximum inner isoperimetric ratio of a cluster in the graph (cf. Definition 1). On the other one, denominator $a_1\sqrt{c}$ grows when the ratio $\cfrac{w(\partial \mathcal{\mathcal{P}})}{c}$ decreases.
>
> Intuitively, both $c$ and $\frac{1}{\iota}$ capture a notion of how "full" the clusters are, and their "fullness" or connectedness should dominate the total weight of the boundary for our approach to be beneficial, in the same way that the condition $d \gg s_0$ makes the LASSO approach beneficial.
>
> For the case where $\kappa>1$, we will have an additive contribution of $w(\partial \mathbf{\Theta})$, which results in a benefit when $w(\partial \mathbf{\Theta}) \ll \phi$. We did not study the behavior of constant $\phi$ but it can be an interesting future research direction.

---

> ### Comment · Reviewer_QF4D · 2024-08-12
>
> Thank you for the detailed responses to my questions. I have no further questions.

---

### Official Review · Reviewer_wAUZ · 2024-07-19

**Soundness:** 2
**Presentation:** 3
**Contribution:** 2
**Rating:** 5
**Confidence:** 3

**Summary:**

The paper introduces a multi-task contextual bandit algorithm that leverages a graph structure to model relationships between tasks. The algorithm assumes that the preference vectors of the tasks are piecewise constant over the graph, forming clusters. By solving an online network lasso problem with a time-dependent regularization parameter, the algorithm estimates the preference vectors, achieving a sublinear regret bound lower than independent task learning. Theoretical findings are supported by experimental evaluations against other graph bandit and online clustering algorithms.

**Strengths:**

(1) The paper introduces a approach by incorporating graph structures to model relationships between tasks.

(2) The algorithm is supported by comprehensive theoretical analysis, including a oracle inequality and a regret bound.

(3) Extensive experiments validate the proposed method, showing that it outperforms existing baselines in terms of cumulative regret, highlighting its practical applicability and effectiveness.

**Weaknesses:**

(1) The problem setting and algorithm presented are primarily adaptations of existing works, such as Oh et al. [2021]. The main difference is the inclusion of a graph matrix in the user preference vector, but this is not the first algorithm to incorporate a graph in contextual bandits, limiting the overall novelty.

(2)  The i.i.d. assumption in contextual bandits is quite strong. Even in clustering approaches like CLUB, a conditional i.i.d. assumption is used. The current regret upper bound complexity is \(\sqrt{VT}\). There should be special cases where the algorithm can improve over \(V\) to demonstrate a more significant advantage.

(3)  Since 2019, there have been many more works on clustering in bandits. The authors should conduct a broader survey to include these more recent works and relevant baselines. Using SCLUB, which is considered outdated, as a baseline, limits the comprehensiveness and relevance of the comparative analysis.

**Questions:**

What are the unique challenges in regret analysis with Lasso regularization compared to L2 regularization?

---

> ### Author Rebuttal · Authors · 2024-08-07
>
> ## Limited novelty
> Despite the similarity in the technical tools used in the analysis to those in Oh et al. 2021, we respectfully disagree. Indeed, such techniques have also been used in [3,4,5,6] but we still faced the challenge of formulating a suitable RE condition (Definition 2), ensuring that it holds with high probability for the empirical Gram matrix (Theorem 2) and proving a novel oracle inequality (Theorem 1) that was not established even in the offline learning literature with i.i.d samples. Additionally, we have put additional effort into linking the analysis to the properties of the graph such as the total weight of the boundary, the maximum inner isoperimetric ratio of a cluster, and the minimum topological centrality of a cluster.
>
> ## I.i.d. assumption:
> Kindly refer to the general rebuttal where we address this point.
>
> ## Broader survey for clustering of bandits and comparing to more baselines
> We apologize for the lack of some references, and we will take them into account. As for the baselines, kindly refer to the ones we mentioned in the general rebuttal.
>
> ## Unique challenges in regret analysis with Lasso regularization compared to L2 regularization
> We faced several challenges that would not arise in the case of the ridge(L2 squared) regularization. We apologize for not having been able to point out all of them in the main material as we were limited by the number of pages.
>
> First, to the ridge regularization or regularization of its type (e.g. Laplacian regularization in [13]), there is no analytical solution to the optimization problem, which requires a completely different approach. It is similar in spirit to the difference between analyzing the LinUCB algorithm and the Lasso-type bandit algorithms [1-6].
>
> Second, different from [1,2,3,5] which adapt the i.i.d case oracle inequality to the adapted case, we established a new one that can be of interest in the i.i.d. case in particular. We also point out at the challenges faced to formulate the RE condition and to ensure that it transfers well to the empirical Gram matrix, that we already mentioned when we addressed the limited novelty point. For the RE condition in particular, we had to make sure that our algorithm guarantees that the graph structure accelerates the estimation of the Gram matrix, as formalized in Proposition 7 using Definition 3, both in the appendix.

---

> > ### Comment · Reviewer_wAUZ · 2024-08-12
> >
> > Thank you for your response. Can you clarify which assumption in references [3, 5, 6] corresponds to the i.i.d. arm context assumption?

---

> > > ### Author Response · Authors · 2024-08-12
> > > **On the the iid context set generation assumption in references [3,5,6]**
> > >
> > > Thank you for your question. The i.i.d context  generation assumption in those references are stated as follows:
> > > * Reference [3]: at the beginning of Section 2.2 titled "Generalized Linear Contextual Bandits" (end of the third sentence there), which states: "where the tuple $\mathcal X_t$ is drawn i.i.d. over $t \in [T]$ from an unknown joint distribution with probability density $p_{\mathcal X}$".
> > > * Reference [5]: at the beginning of Section 3.1 titled "Model and Notation" (beginning of the third sentence there),  which states: "The successive sets $(\mathcal{A}_t)\_{t \geq 1}$ form an i.i.d. sequence with distribution $p_A$".
> > > * Reference [6]: at the last sentence of Assumption 1 under Section 3.1 titled "Stochastic Linear Contextual Bandits", which states: "We assume the tuples $\mathcal{D}_1, \cdots, \mathcal{D}_N$" to be drawn i.i.d. from a fixed unknown zero mean sub-Gaussian joint distribution $p$ on $\mathbb{R}^{Kd}$".

---

> > > > ### Comment · Reviewer_wAUZ · 2024-08-12
> > > >
> > > > Thank you for your clarifications. After checking related Lasso bandit works, I see that the i.i.d. assumption is indeed used. However, I still think that this assumption should be removed in the context of the bandit problem, as it limits the exploration, which is a crucial component of these algorithms. The current approach seems more aligned with a greedy algorithm. Nonetheless, with the theoretical contributions made in this paper, I have accordingly increased the score.

---

### Author Rebuttal · Authors · 2024-08-07

We would like to express our deep gratitude to the reviewers for the substantial effort they put into reading and evaluating our work.

Upon recognizing that several reviewers have raised common concerns, we will address these in a general rebuttal. Specific responses to individual concerns will be provided separately for each reviewer. We kindly invite the reviewers to refer to the lists of references we provide at the end of this general rebuttal.

## Context generating process (reviewers wAUZ, Z89V)
We assume that the context sets are i.i.d. generated, and verify relaxed-symmetry and balanced covariance assumptions. These are standard assumptions that have been used for some time in the literature [3,5,6]. In [2], a different assumption stating that the arms within a set of actions are i.i.d. is used.

As for [7], the authors still require the context set elements to be i.i.d. sampled from a full rank process matrix with minimum eigenvalue $\lambda>0$ (Such an eigenvalue assumption is stronger than our RE assumption).  Furthermore, we do not offer a clustering approach in contrast to other clustering algorithms, which aim to learn the cluster structure of tasks explicitly. Instead, we leverage the relevant cluster information implicitly, using an a priori available graph, in the same way that a LASSO estimator leverages the sparsity structure of the true parameters vector.

## Dependence of the oracle inequality (and hence the regret) on the dimension (Reviewers QF4D, 3afv)

In our regret bound (Theorem 3), we have :
* a part having a logarithmic dependence in the dimension, resulting from the need to ensure the compatibility condition for the empirical Gram matrix.
* a part that does not depend on the dimension, representing the bulk of the horizon dependence of the regret. This part is the result of summing up the bounds of the oracle inequality over time in the proof of Theorem 3.

As a result, to understand the absence of dependence on the dimension (except maybe for the constant $\phi$), we need to look at the establishment of the oracle inequality. To prove the latter, we had to bound a noise process of the form $\sum\_{\tau=1}^t x_\tau \eta_\tau$, which we treat rigorously in Lemma 3 of the appendix. There, we use the generalization of the Hanson-Wright inequality proven in [10] and that we mention in our paper in Theorem 4.

Using the notations of the proof of Lemma 3, denoting $ M = {\bf X}\_{\mathcal{V}}{\bf X}\_{\mathcal{V}}^\top $ which is a PSD matrix, the bound therein can be chosen to depend only on the trace of $M$, the square root of the trace of $M^2$, and the spectral norm of $M$. All of these quantities can be bounded by the trace of $M$, which in turn is equal to $\Vert{\bf X}\_{\mathcal{V}}\Vert_F^2$, at most bounded by $t$ since context vectors are assumed to have a norm at most bounded by $1$.

## Additional baselines (wAUZ, 3afv)
We added a comparison to the Trace-Norm bandit [6] and Local Clustering of Bandits [13]. We also added a comparison to OLS bandits with independent task learning.

#  References
[1] Bastani, Hamsa, and Mohsen Bayati. “Online Decision Making with High-Dimensional Covariates.” Operations Research, Nov. 2019.

[2] Kim, Gi-Soo, and Myunghee Cho Paik. “Doubly-Robust Lasso Bandit.” Advances in Neural Information Processing Systems, vol. 32, Curran Associates, Inc., 2019.

[3] Oh, Min-Hwan, et al. “Sparsity-Agnostic Lasso Bandit.” Proceedings of the 38th International Conference on Machine Learning, PMLR, 2021.

[4] Cella, Leonardo, and Massimiliano Pontil. “Multi-Task and Meta-Learning with Sparse Linear Bandits.” Proceedings of the Thirty-Seventh Conference on Uncertainty in Artificial Intelligence, PMLR, 2021.

[5] Ariu, Kaito, et al. “Thresholded Lasso Bandit.” Proceedings of the 39th International Conference on Machine Learning, PMLR, 2022, pp. 878–928.

[6] Cella, Leonardo, et al. “Multi-Task Representation Learning with Stochastic Linear Bandits.” Proceedings of The 26th International Conference on Artificial Intelligence and Statistics, PMLR, 2023.

[7] Gentile, Claudio, Shuai Li, and Giovanni Zappella. "Online clustering of bandits." International conference on machine learning. PMLR, 2014.

[8] Bühlmann, Peter, and Sara Van De Geer. Statistics for high-dimensional data: methods, theory and applications. Springer Science & Business Media, 2011.

[9] Ji, Shuiwang, and Jieping Ye. "An accelerated gradient method for trace norm minimization." Proceedings of the 26th annual international conference on machine learning. 2009.

[10] Hsu, Daniel, Sham Kakade, and Tong Zhang. "A tail inequality for quadratic forms of subgaussian random vectors." (2012): 1-6.

[11] Wainwright, Martin J. High-dimensional statistics: A non-asymptotic viewpoint. Vol. 48. Cambridge university press, 2019.

[12] Jung, Alexander. "Networked exponential families for big data over networks." IEEE Access 8 (2020): 202897-202909.

[13] Ban, Yikun, and Jingrui He. "Local clustering in contextual multi-armed bandits." Proceedings of the Web Conference 2021.

---

### Decision · Program_Chairs · 2024-09-25

**Decision:**

Reject

**Comment:**

The paper proposes a multi-task bandit algorithm which exploits the graph structure between tasks when preference vectors are piecewise constant over the graph. The algorithm solves a new optimization problem with Lasso regularization, and chooses actions greedily based on the resulting estimate which implicitly incoporates exploration. Sublinear regret is guaranteed with high probability, without the need to identify the clusters.

Contribution: A novel oracle inequality (which wasn't established even in the offline setting) is derived for estimation error of the parameter matrix estimate, which can be of independent interest.

Weakness of the paper raised by reviewers were
- iid assumption
- lack of comparison baselines and lack of literature review
- ambiguity of the regret bound in terms of the dependence on dimension
- RE condition not well explained

One point that I do not agree with the reviewers is that the "iid assumption is strong". It is true that many bandit algorithms do not require this assumption, but still, existing algorithms that utilize Lasso are mostly (almost all, I think) built on this assumption. Hence, I think this should not be a decisive factor of acceptance/rejection.

However, even putting aside the comments on iid assumptions, some reviewers commonly point out the ambiguity of the current regret bound in terms of the dependence on the dimension. Authors claim that the relationship between the eigenvalue of the Gram matrix and dimension was not discussed before, but I find a paper (Kim et al., 2021) that discusses such relationship (although the definiton of eigenvalue may not be exactly the same.)
Moreover, intuitions behind many assumptions leading to the regret are not well explained.

Overall, the paper could benefit from improved presentation, including in-depth literature review, additional analysis of the dependence on dimension, and intuitive explanations behind numerous assumptions.

Reference: Kim, W., Kim, G. S., & Paik, M. C. (2021). Doubly robust thompson sampling with linear payoffs. Advances in Neural Information Processing Systems, 34, 15830-15840.